# Extensive cargo identification reveals distinct biological roles of the 12 importin pathways

Makoto Kimura[1]*, Yuriko Morinaka[1], Kenichiro Imai[2,3], Shingo Kose[1], Paul Horton[2,3], Naoko Imamoto[1]*

[1]Cellular Dynamics Laboratory, RIKEN, Wako, Japan; [2]Artificial Intelligence Research Center, National Institute of Advanced Industrial Science and Technology, Tokyo, Japan; [3]Biotechnology Research Institute for Drug Discovery, National Institute of Advanced Industrial Science and Technology, Tokyo, Japan

**Abstract** Vast numbers of proteins are transported into and out of the nuclei by approximately 20 species of importin-$\beta$ family nucleocytoplasmic transport receptors. However, the significance of the multiple parallel transport pathways that the receptors constitute is poorly understood because only limited numbers of cargo proteins have been reported. Here, we identified cargo proteins specific to the 12 species of human import receptors with a high-throughput method that employs stable isotope labeling with amino acids in cell culture, an in vitro reconstituted transport system, and quantitative mass spectrometry. The identified cargoes illuminated the manner of cargo allocation to the receptors. The redundancies of the receptors vary widely depending on the cargo protein. Cargoes of the same receptor are functionally related to one another, and the predominant protein groups in the cargo cohorts differ among the receptors. Thus, the receptors are linked to distinct biological processes by the nature of their cargoes.

*For correspondence: makimura@ riken.jp (MK); nimamoto@riken.jp (NI)

Competing interests: The authors declare that no competing interests exist.

## Introduction

In interphase cells, proteins and RNAs migrate into and out of the nuclei through the central channels of the nuclear pore complexes (NPC) embedded in the nuclear envelope. These nuclear pores are lined with FG-repeat domains that constitute a permeability barrier, and only macromolecules that reversibly interact with the FG-repeats can permeate this barrier (*Schmidt and Görlich, 2016*). One such group of proteins, the importin (Imp)-$\beta$ family proteins, are nucleocytoplasmic transport receptors (NTRs) that primarily carry nuclear proteins and small RNAs as their cargoes through the nuclear pores, although non-importin family NTRs also act depending on the cargo and physiological conditions (*Kose et al., 2012*; *Lu et al., 2014*; *Weberruss et al., 2013*). The human genome encodes 20 species of Imp-$\beta$ family NTRs, of which 10 [Imp-$\beta$, transportin (Trn)-1, -2, -SR (-3), Imp-4, -5 (RanBP5), -7, -8, -9, and -11] are nuclear import receptors, 7 [exportin (Exp)-1 (CRM1), -2 (CAS/CSE1L), -5, -6, -7, -t, and RanBP17] are export receptors, 2 (Imp-13 and Exp-4) are bi-directional receptors, and the function of RanBP6 is undetermined (*Kimura and Imamoto, 2014*). These NTRs constitute multiple parallel transport pathways. The basic mechanism of directional transport was elucidated in the early years (*Görlich and Kutay, 1999*), but even today, the number of NTR-specific cargoes that has been reported is surprisingly small, hindering a biological understanding of the nucleocytoplasmic transport system.

NTRs are thought to transport specific cohorts of cargoes by binding to specific sites on those cargoes (*Chook and Süel, 2011*), but the consensus structures of the NTR-binding sites have been established for only a few NTRs (*Soniat and Chook, 2015*) as follows: the classical nuclear

localization signal (cNLS) for the Imp-α family adapters, which connects Imp-β and cargoes (*Lange et al., 2007*); PY-NLS for Trn-1 and -2 (*Lee et al., 2006*; *Süel et al., 2008*); the nuclear export signal (NES) for Exp-1 (*Hutten and Kehlenbach, 2007*); the SR-rich domain that binds to Trn-SR (*Kataoka et al., 1999*; *Maertens et al., 2014*); and Lys-rich NLS (IK-NLS) for yeast Kap121p (Imp-5 homolog; *Kobayashi and Matsuura, 2013*; *Kobayashi et al., 2015*). The β-like importin-binding (BIB) domain is another NTR-binding site (*Jäkel and Görlich, 1998*), but its consensus sequence and NTR specificity remain obscure. Among the NTRs, Imp-β exclusively uses one of the seven species of the Imp-α family of proteins as an adapter for cargo binding, and many Imp-α/β cargoes have been reported, although Imp-β also directly binds to cargoes (*Goldfarb et al., 2004*). Among the import receptors, Trn-1 and its closest homolog Trn-2 have the second-highest number of cargoes reported thus far, and the PY-NLS motif has been defined, although in some cases the motif is difficult to recognize because of sequence diversity and structural disorder is another requisite (*Soniat and Chook, 2015*, *2016*). For the cargoes of other NTRs, the consensus structures of NTR-binding sites have hardly been derived because only limited numbers of cargoes have been reported, including Imp-β-direct cargoes.

There are many reports on the differential spatiotemporal expression of Imp-β family NTRs, including tissue specificities in humans (*Quan et al., 2008*), developmental or spermatogenic stage specificities in mice (*Major et al., 2011*; *Quan et al., 2008*), and tissue or response specificities in plants (*Huang et al., 2010*). Expression regulation is not only transcriptional but also miRNA-mediated (*Li et al., 2013*; *Szczyrba et al., 2013*) or locally translationally mediated (*Perry and Fainzilber, 2009*). Additionally, the NTRs are functionally regulated by protein modifications (*Wang et al., 2009*), inhibitory factors (*Lieu et al., 2014*), and specific anchorings (*Makhnevych et al., 2003*). These nucleocytoplasmic transport regulations must significantly influence cellular physiology, and their significance may be elucidated if the affected cargoes can be specified. Indeed, in previous studies, NTR regulations have been linked to cellular responses through the functions of specific cargoes. For example, in prostate cancer cells treated with a cinnamaldehyde derivative, the expression of Imp-7 and the transcription factor Egr1 are induced, and the Egr1 imported by Imp-7 activates apoptotic gene transcription (*Kang et al., 2013*). In another example, when the nuclear import of some ribosomal proteins (RPs) is inhibited by the repression of Imp-7 expression, other unassembled RPs restrain the negative regulator of p53 Mdm2 and thereby activate p53 to inhibit cell growth (*Golomb et al., 2012*). Additionally, the inhibition of Trn-2 by the caspase-generated HuR (ELAVL1) fragment is crucial for the cytoplasmic retention of full-length HuR, which induces myogenesis (*Beauchamp et al., 2010*) or staurosporine-induced apoptosis (*von Roretz et al., 2011*). In many other studies, mutations of particular NTR genes in model organisms, including yeast, flies, and plants, have resulted in defects in specific biological processes (*Kimura and Imamoto, 2014*). Thus, each NTR has its own inherent biological significance. However, the details of the molecular processes are largely uncharacterized because the responsible cargoes have not been identified. If we could identify more cargoes, further studies of cellular regulation by nucleocytoplasmic transport would be possible.

We previously established a method for identifying the cargoes of a nuclear import receptor called SILAC-Tp (*Kimura et al., 2013a*, *2013b*, *2014*). SILAC-Tp employs stable isotope labeling with amino acids in cell culture (SILAC) (*Ong et al., 2002*), an in vitro reconstituted nuclear transport system (*Adam et al., 1990*), and quantitative mass spectrometry. A recent advancement of the Orbitrap mass spectrometer drastically increased the identified and quantified protein numbers, and this advancement has been successfully applied to other cargo identification methods (*Kırlı et al., 2015*; *Thakar et al., 2013*). Here, we utilized this advancement for the SILAC-Tp method and identified import cargoes of all 12 NTRs, of which 10 are import and two are bi-directional receptors. Our results illustrate the basic framework and the biological significance of the nucleocytoplasmic transport pathways.

## Results and discussion

### SILAC-Tp effectively identifies cargoes

SILAC-Tp employs an in vitro nuclear transport system, and all 12 NTRs import their reported specific cargoes in this system (*Figure 1—figure supplement 1C*). The transport system consists of

permeabilized HeLa cells labeled with 'heavy' amino acids by SILAC, unlabeled HeLa nuclear extract depleted of Imp-$\beta$ family NTRs and RCC1, unlabeled HeLa cytosolic extract depleted of Imp-$\beta$ family NTRs, one species of recombinant NTR, p10/NTF2, and an ATP regeneration system. Unlabeled 'light' proteins in the nuclear extract are imported into the nuclei of the permeabilized cells. Simultaneously, a control reaction without the NTR is performed. Next, the proteins are extracted from the nuclei and identified and quantified by LC-MS/MS. The recipient nuclei contain both the imported and endogenous proteins, and the ratio of the imported to the endogenous fraction of a protein is calculated as the unlabeled/labeled or light/heavy (L/H) ratio. The quotient of the L/H ratios with the NTR (+NTR) and without it (control), that is, $(L/H_{+NTR})/(L/H_{Ctl})$, of a protein is defined as the +NTR/Ctl value and is used as the index for cargo potentiality.

In one run of SILAC-Tp (control or +NTR), approximately 2500 to 4000 proteins were identified, and the L/H ratios of 1700 to 3100 proteins were quantified. To calculate the +NTR/Ctl value, one protein has to be quantified in both the control and +NTR reactions, and we discarded $L/H_{+NTR}$ values that lacked the counterpart $L/H_{Ctl}$ values. We performed three replicates of SILAC-Tp for each of the 12 NTRs. In the three replicates, 1235 to 1671 proteins were assigned with +NTR/Ctl values three times, and 364 to 502 proteins were assigned only twice (*Supplementary file 1*). We did not consider proteins with single +NTR/Ctl values, although a protein with only a single but high +NTR/Ctl value may still be a cargo (see below). To normalize the index values of the three replicates, the Z-scores of the $\log_2$(+NTR/Ctl) were calculated within each replicate (*Figure 1—figure supplement 2A and B*). Ranking the proteins that have three +NTR/Ctl values by the median of the three Z-scores may reasonably sort the candidate cargoes. However, if the lower Z-score of a protein with only two +NTR/Ctl values is higher than the median Z-scores of those candidate cargoes, the protein may also be a candidate cargo. Thus, we ranked the proteins by the second (the lower of the two or the middle of the three) Z-scores, and termed the result the 2nd-Z-ranking (*Supplementary file 1*).

To define the border that separates candidate cargoes from other proteins in the 2nd-Z-ranking, we first reviewed the distribution of reported Trn-1 cargoes in the Trn-1 2nd-Z-ranking because many Trn-1 cargoes have been reported. For an unbiased evaluation, we employed the lists of cargoes consolidated by other researchers (*Chook and Süel, 2011*). Twenty-seven reported cargoes were included in the 2nd-Z-ranking (totaling 1649 proteins; *Supplementary file 1*, Trn-1 'Report and feature'). We calculated the reported cargo rates (to serve as a proxy for precision), recall, and Fisher's exact test p-values for rank cutoffs in increments of 1%. Computing reported cargo rate requires deciding which candidate cargoes should be considered as false positives. Since a gold standard set of definitely non-cargo proteins is not available, it is not clear which previously unreported cargoes should be counted as false positives, and which, if any, should be discarded as unclear. Therefore, we estimated reported cargo rates in two ways: (i) treating all the 1622 proteins not reported as cargoes as negative examples (*Figure 1—figure supplement 3A* and *Figure 1—source data 1A*); and (ii) discarding proteins with undetermined or nuclear subcellular localization according to Uniprot annotation, and treating the remaining 259 non-nuclear proteins as negative examples (*Figure 1—figure supplement 3A* and *Figure 1—source data 1B*). In the former case, the reported cargo rate corresponds to a lower bound on the precision, and even in the latter case, the reported cargo rates are expected to underestimate precision, because almost certainly some of the proteins that we exclude as unclear are in fact true cargoes.

To select cargoes with high sensitivity, we employed the cutoff of 15% that yields a high recall of 0.741 (*Figure 1—figure supplement 3B* and *Figure 1—source data 1A and 1B*; recall is not affected by the assumptions of negative examples). Among the 27 reported cargoes, 20 cargoes were ranked in the top 15% (247 proteins; p=5.39 $\times$ 10$^{-12}$ by Fisher's exact test), and the others were dispersed in the lower ranks (*Figures 1A* and *2A*; *Figure 1—figure supplement 2C and E*).

We examined the direct binding of Trn-1 to a subset of proteins in the 2nd-Z-ranking using a bead halo assay (*Patel and Rexach, 2008*) (*Supplementary file 2*) in which the binding of GFP-fusion proteins to GST-Trn-1 on glutathione-Sepharose beads was observed by fluorescence microscopy. If RanGTP (a Q69L GTP-fixed mutant) (*Bischoff and Ponstingl, 1995*) inhibits the protein–Trn-1 binding, the functionality of the binding is verified. For all the bead halo assays in this work, we principally selected well-characterized proteins that have not been reported as cargoes from (i) proteins ranked high (within the top 15% in the 2nd-Z-ranking or 4% in the 3rd-Z-ranking, see below), around presumptive cutoffs (within about top 15–25% in the 2nd-Z-ranking), or lower and (ii) highly ranked proteins that are suspected as indirect cargoes or false positives based on their well-known

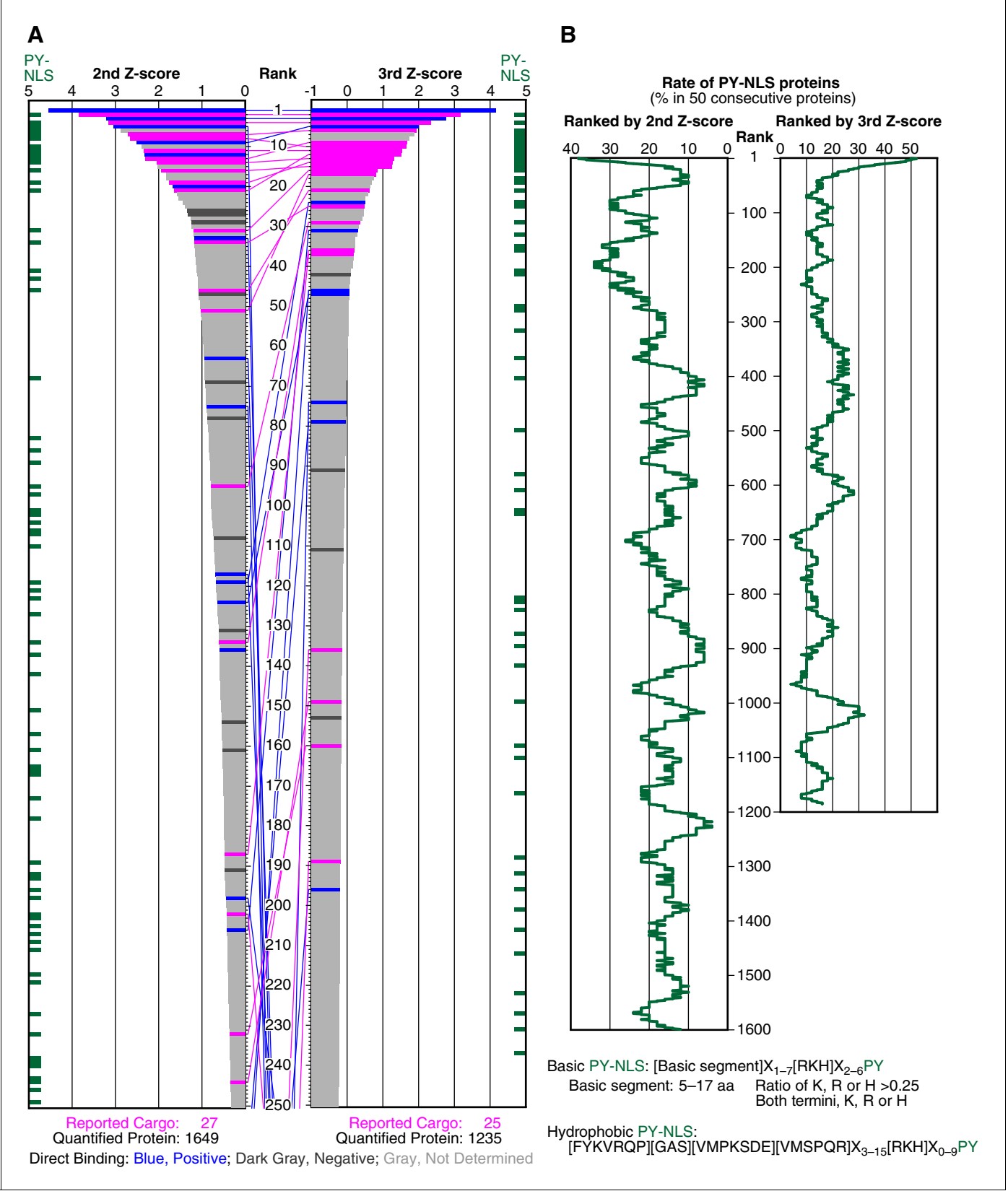

**Figure 1.** SILAC-Tp effectively sorts Trn-1 cargoes. (A) Z-scores in the Trn-1 2nd- and 3rd-Z-rankings. The second (left) and third (right) Z-scores are presented for the top 250 proteins in the Trn-1 2nd- and 3rd-Z-rankings, respectively. The total number of ranked (quantified) proteins and the number

*Figure 1 continued on next page*

*Figure 1 continued*

of previously reported cargoes included in the ranking are indicated at the bottom. The magenta bars represent previously reported cargoes. The blue and dark gray bars represent the proteins that did and did not bind directly to Trn-1, respectively, in the bead halo assays (*Supplementary file 2*). Identical proteins marked by the colors are connected by lines. Proteins that carry PY-NLS motifs are indicated by green bars. (B) Distribution of PY-NLS motif-containing proteins in the rankings. The percentage of the proteins carrying PY-NLS motifs in 50 consecutively aligned proteins is presented along with the 2nd- and 3rd-Z-rankings (left and right, respectively). For example, the top 50 proteins in the 2nd-Z-ranking include 19 (38%) PY-NLS motif-containing proteins, and thus the value at position 1 is 38%. Two types of PY-NLS motifs, basic and hydrophobic, are defined as presented at the bottom.

The following source data and figure supplements are available for figure 1:

**Source data 1.** Statistical analysis of reported cargoes in the Trn-1 2nd- and 3rd-Z-ranking.

**Figure supplement 1.** SILAC-Tp experimental system.

**Figure supplement 2.** Trn-1 cargoes are effectively sorted by the second or third Z-scores in three replicates of SILAC-Tp.

**Figure supplement 3.** Reported cargo rates and recall of the Trn-1 2nd- and 3rd-Z-ranking.

features, for example, S100A6 or EEF1A2 (see the legend of *Supplementary file 2*). The negative rate of the bead halo assays should be higher than the true overall false-positive rate of the SILAC-Tp, because proteins in (ii) are selected preferentially. Seventeen novel candidate cargoes in the top 266 (top 16%) bound to Trn-1, and RanGTP inhibited the binding (*Figure 1A*; *Supplementary file 1*, Trn-1 'Direct binding'; *Supplementary file 2*). Although the assays were not comprehensive, many of the highly ranked proteins are *bona fide* Trn-1 cargoes. The highly ranked proteins that did not bind to Trn-1 in the assays are still candidate indirect cargoes that may form complexes with other proteins that directly bind to Trn-1 (see the case of POLE3 for Imp-13 below). As an example of a protein with only a single but high Z-score (+NTR/Ctl value), DIMT1 bound to Trn-1 (DHRS4 with Imp-$\beta$ is another example), but we did not consider such proteins.

Because many reported Trn-1 cargoes carry PY-NLSs, we examined the distribution of PY-NLS motif-containing proteins in the 2nd-Z-ranking (*Figure 1B*). The percentages of PY-NLS motif-containing proteins within a window width of 50 positions were higher in the range of the top 200, indicating a higher rate of PY-NLS motif-containing proteins within the top 250 (top 15%). The reported Trn-1 cargoes were similarly distributed in the Trn-2 2nd-Z-ranking (*Supplementary file 1*, Trn-2 'Report or feature'). Because Trn-1 and -2 share nearly the same reported cargoes (*Twyffels et al., 2014*), this result demonstrates the reproducibility of the SILAC-Tp method. Based on these evaluations, we assumed that the proteins in the top 15% (247 proteins) of the 2nd-Z-ranking are candidate cargoes with high sensitivity (0.741) and termed them the 2nd-Z-15% cargoes.

Next, we examined whether the cutoff employed for Trn-1 is applicable to Imp-13 and Trn-SR whose 2nd-Z-rankings include several reported cargoes. The Imp-13 2nd-Z-ranking (totaling 2060 proteins) includes eight reported cargoes (*Supplementary file 1*, Imp-13), and seven of these are ranked in the top 244 (top 12%; p=2.83 $\times$ $10^{-7}$; *Figure 2B*; *Figure 2—figure supplements 1A* and *2A*). In bead halo assays for a subset of the ranked proteins, 24 novel candidate cargoes in the top 326 (top 16%) bound directly to Imp-13, and RanGTP inhibited the binding (*Figure 2—figure supplement 2A*; *Supplementary file 1*, Imp-13; *Supplementary file 2*). One component of a reported cargo complex, that is, POLE3, did not bind to Imp-13, but its binding partner CHRAC1 (*Walker et al., 2009*) did. Thus, the binding partners of the direct cargoes are also ranked high. Many reported Trn-SR cargoes are SR-domain proteins (*Chook and Süel, 2011*), and they can be grouped into either SR-rich splicing factors (SFs) or other SR-domain proteins. The Trn-SR 2nd-Z-ranking (totaling 2021 proteins) contains three reported cargoes (*Supplementary file 1*, Trn-SR), and they are ranked in the top 55 (top 3%; p=1.91 $\times$ $10^{-5}$; *Figure 2C*; *Figure 2—figure supplement 2B*). The 2nd-Z-ranking contains seven SR-rich SFs other than the reported SFs, and five of these are ranked in the top 90 (top 4%; p=7.61 $\times$ $10^{-18}$). The 2nd-Z-ranking also contains another four proteins that are annotated with 'RS-domain' in UniProt, and three of these are ranked in the top 202 (top 10%; p=3.65 $\times$ $10^{-3}$). Finally, in bead halo assays for a subset, 11 novel candidate cargoes in

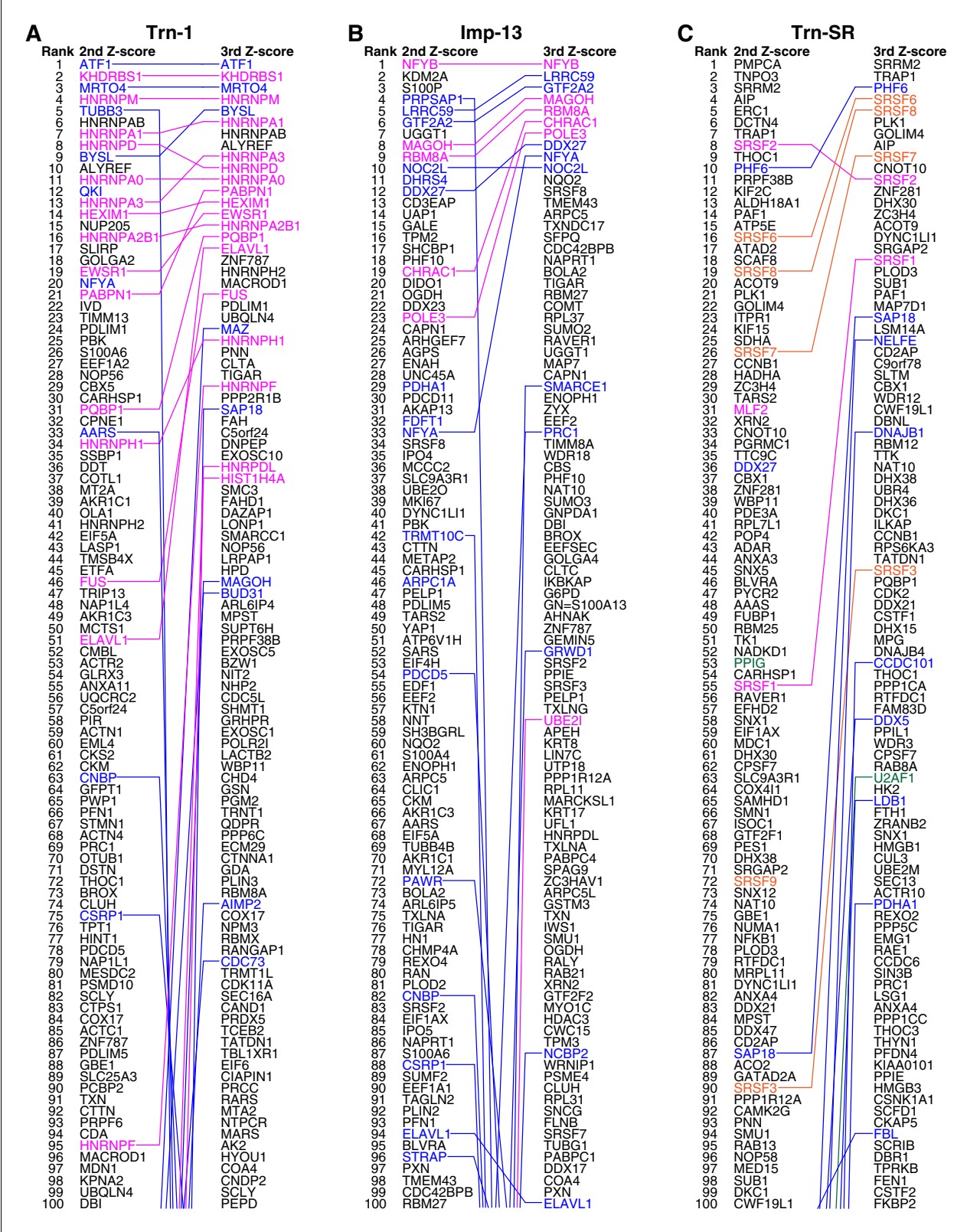

**Figure 2.** Trn-1, Imp-13, and Trn-SR cargo rankings. (A–C) The top 100 proteins in the Trn-1 (A), Imp-13 (B), and Trn-SR (C) 2nd- and 3rd-Z-rankings (left and right, respectively). Magenta, reported cargoes; blue, proteins bound directly to the NTR in the bead halo assays (*Supplementary file 2*); orange in (C), SR-rich SFs that have not been reported; and green in (C), other RS (SR)-domain proteins. Identical proteins marked by the colors are connected by lines.

*Figure 2 continued on next page*

*Figure 2 continued*

The following figure supplements are available for figure 2:

**Figure supplement 1.** Imp-13 cargoes are effectively sorted by the second or third Z-scores in three replicates of SILAC-Tp.

**Figure supplement 2.** SILAC-Tp effectively sorts Imp-13 and Trn-SR cargoes.

**Figure supplement 3.** Imp-*β* cargo ranking and Z-scores in the 2nd- and 3rd-Z-rankings.

the top 237 (top 12%) bound directly to Trn-SR, and RanGTP inhibited the binding (*Figure 2—figure supplement 2B*; *Supplementary file 1*, Trn-SR; *Supplementary file 2*). Hence, the 2nd-Z-15% cargoes could also be defined for Imp-13 (309 proteins) and Trn-SR (302 proteins), and we applied this cutoff to the other NTRs that have few reported cargoes. The 2nd-Z-15% cargoes of the 12 NTRs are presented in *Supplementary file 3*. Some of the 2nd-Z-15% cargoes with low numbers of L/H counts showed deviation in Z-scores or L/H ratios in the three replicates of SILAC-Tp (*Supplementary file 1*), and an example of their quantitation qualities is presented in *Supplementary file 4*.

Exceptionally, Imp-*β* uses Imp-α as an adaptor for cargo binding, and the cytosolic extract used for the transport system contained endogenous Imp-α. Four Imp-αs were found in the Imp-*β* 2nd-Z-ranking (totaling 2027 proteins), and three of these are in the 2nd-Z-15% cargoes (p=1.19 × $10^{-2}$; *Supplementary file 1*, Imp-*β*; *Supplementary file 3*). Thus, the Imp-*β* candidate cargoes must include both Imp-*β*-direct and Imp-α-dependent cargoes. Indeed, 31 proteins in the top 276 (top 14%) bound directly to Imp-α, -*β*, or both in the bead halo assays (*Supplementary file 1*, Imp-*β*; *Figure 2—figure supplement 3*; *Supplementary file 2*). The border for the Imp-*β* candidate cargoes can be relaxed because Imp-*β* imports more cargoes than other NTRs with the help of Imp-α. Indeed, in the bead halo assays, many proteins in the top 35% of the 2nd-Z-ranking bound to Imp-α, although most of the proteins that bound directly to Imp-*β* were ranked in the top 259 (13%). Here, we employed the Imp-*β* 2nd-Z-15% cargoes (303 proteins) to enable equal comparisons with the cargoes of other NTRs.

## Cargo selection with higher specificity

Deviation of the LC-MS/MS quantification within the three replicates complicates cargo selection. However, the Z-scores of the highly ranked reported cargoes were reasonably high in all the three replicates possibly because many of the reported cargoes are abundant proteins that seldom produce outliers in quantification (*Figure 1—figure supplement 2*; *Figure 2—figure supplement 1*). To select proteins that have high Z-scores in all the three replicates, we next ranked the proteins that had three +NTR/Ctl values by the third (lowest) Z-scores (3rd-Z-ranking). The reported cargo rates, recall, and p-values were calculated in 1% rank increments under two assumptions similarly to the case of 2nd-Z-ranking (*Figure 1—figure supplement 3A and B* and *Figure 1—source data 1C and 1D*). The reported cargo rate calculated under the assumption that proteins annotated with non-nuclear localization (178 proteins) are negative examples is as high as 0.85 at the cutoff of top 4% (*Figure 1—figure supplement 3A* and *Figure 1—source data 1D*). The Trn-1 3rd-Z-ranking (totaling 1235 proteins) included 25 reported cargoes, and 17 of these were ranked in the top 37 (top 3%; p=1.67 × $10^{-22}$; *Figures 1A* and *2A*; *Figure 1—figure supplement 2D and F*; *Supplementary file 1*). Seven proteins in the top 47 (top 4%) were novel Trn-1-direct cargoes that were verified in the bead halo assays (*Figures 1A* and *2A*; *Supplementary files 1* and *2*). The percentage of PY-NLS motif-containing proteins within a window width of 50 positions was highest at the first position (*Figure 1B*), indicating that PY-NLS motif-containing proteins are concentrated in the top 50 (top 4%). Thus, most of the proteins that ranked in the top 4% (49 proteins) of the 3rd-Z-ranking are highly reliable cargoes, and we termed these proteins the 3rd-Z-4% cargoes. In a comparison between the Trn-1 2nd-Z-15% and 3rd-Z-4% cargoes, most of the 3rd-Z-4% cargoes were also 2nd-Z-15% cargoes (*Figure 1A*). Some reported or newly identified cargoes in the 2nd-Z-15% cargoes were ranked lower in the 3rd-Z-ranking due to the deviations in the third Z-scores.

In the Imp-13 3rd-Z-ranking (totaling 1671 proteins), seven proteins were reported cargoes, and six of these were ranked in the top 58 (top 3%; p=9.20 × 10$^{-9}$; *Figure 2B*; *Figure 2—figure supplements 1B* and *2A*; *Supplementary file 1*). Additionally, the 3rd-Z-4% cargoes (66 proteins) included eight novel cargoes that directly bound to Imp-13 (*Figure 2B*; *Figure 2—figure supplement 2A*; *Supplementary files 1* and *2*). In the Trn-SR 3rd-Z-ranking (totaling 1591 proteins), both of the two reported cargoes were ranked in the top 18 (top 1%; p=1.21 × 10$^{-4}$), four of the five other SR-rich SFs were in the top 45 (top 3%; p=2.74 × 10$^{-6}$), one of the three SR-domain proteins (other than the SR-rich SFs) was ranked 63rd (top 4%; p=0.11), and six novel cargoes within the top 4% (63 proteins) bound directly to Trn-SR (*Figure 2C*; *Figure 2—figure supplement 2B*; *Supplementary files 1* and *2*). In cases of both Imp-13 and Trn-SR, the proteins were replaced between the 2nd- and 3rd-Z-rankings in a manner similar to the case for Trn-1. We concluded that the 3rd-Z-4% criteria is highly specific including few false positives, albeit at the cost of losing many genuine cargoes. Hence, we employed the 3rd-Z-4% cargoes mainly for the characterization of the identified cargoes, whereas the 2nd-Z-ranking was used for the evaluation of the import efficiencies of the expected cargoes. The 3rd-Z-4% cargoes of the 12 NTRs are presented in *Figure 3*.

## Redundancy in the import pathways

A total of 468 proteins were identified as 3rd-Z-4% cargoes of the 12 NTRs, and 332 of these are unique to one NTR, which clearly reflects the division of roles among the NTRs (*Supplementary file 5B*). Another 136 proteins were shared by two to seven NTRs, and the mean number of shared cargoes between two NTRs was 4.8. In the maximum-likelihood phylogenetic tree of the 12 NTRs (*Figure 4A*), Trn-1 and -2 (84% sequence identity) are paired most closely, and Imp-7 and -8 (65% identity) are the second-most closely paired. These paired NTRs share 28 and 19 cargoes, respectively, and they are paired similarly in a hierarchical clustering based on the cargo profiles (*Figure 4B and C*). The other NTRs that were paired weakly in the phylogenetic tree, namely, Imp-13 and Trn-SR (23% identity), Imp-4 and -5 (22% identity), and Imp-9 and -11 (19% identity), did not form the same pairs when clustering by their cargoes. Thus, the NTR–cargo interactions are conserved only within the highly homologous NTRs. The 2nd-Z-15% cargoes included as many as 1416 proteins in total, 827 of which are shared by two to 12 NTRs, and 589 are unique to one NTR (*Supplementary file 5A and 5D*). Imp-7 and -8 share the largest number (162) among the 2nd-Z-15% cargoes, but Trn-1 and -2 share no more than the other pairs. Of the 247 Trn-1 and 246 Trn-2 2nd-Z-15% cargoes, 69 are shared, and 36 of these are ranked within the top 50 in either ranking. Thus, Trn-1 and -2 still share many highly ranked cargoes but few lower ranked cargoes within the top 15%. The import efficiency of a cargo may differ between Trn-1 and -2, and only one of Trn-1 or -2 may import inefficient cargoes that are ranked lower.

## Division of roles among the NTRs

Because NTR-dependent transport is regulated, a cargo cohort of an NTR must be imported simultaneously and act cooperatively. To explore the roles of the NTR cargoes, the 3rd-Z-4% and 2nd-Z-15% cargoes of each NTR were analyzed for enrichment of Gene Ontology (GO) terms (*Gene Ontology Consortium, 2015*) using g:Profiler (*Reimand et al., 2016*). For all the combinations of a GO term and an NTR, the number of cargoes annotated with the term and the significance (p-values according to g:SCS) of the term enrichment are listed (*Supplementary files 6B, 6C, 7, and 8*). Depending on the hierarchy of the GO terms, the terms are significantly annotated (p<0.05) to the cargo cohorts of none to 12 of the NTRs. Broader terms with smaller term depths are linked to more NTRs, whereas more defined terms with larger term depths are linked to fewer NTRs. Indeed, all 12 of the NTRs are linked to many broad terms, although the cargo numbers and the significances vary widely. Because similar terms were listed redundantly, we selected representative GO terms from those enriched significantly (p<0.05) for the 3rd-Z-4% cargoes of the 12 NTRs and tabulated the correspondences between the cargoes and the annotated terms (*Supplementary file 9*). To compare the GO terms that are specifically linked to each NTR, we listed the terms that are enriched significantly for the cargoes of four or fewer NTRs (*Figures 5* and *6*). Here, again we extracted the representative terms to decrease the size of the list. The selected terms for the 3rd-Z-4% cargoes plainly exhibit the roles of the cargo cohorts. For example, significant numbers of Imp-4, -7 and Exp-4 cargoes are annotated with DNA recombination or DNA conformation (geometric) change

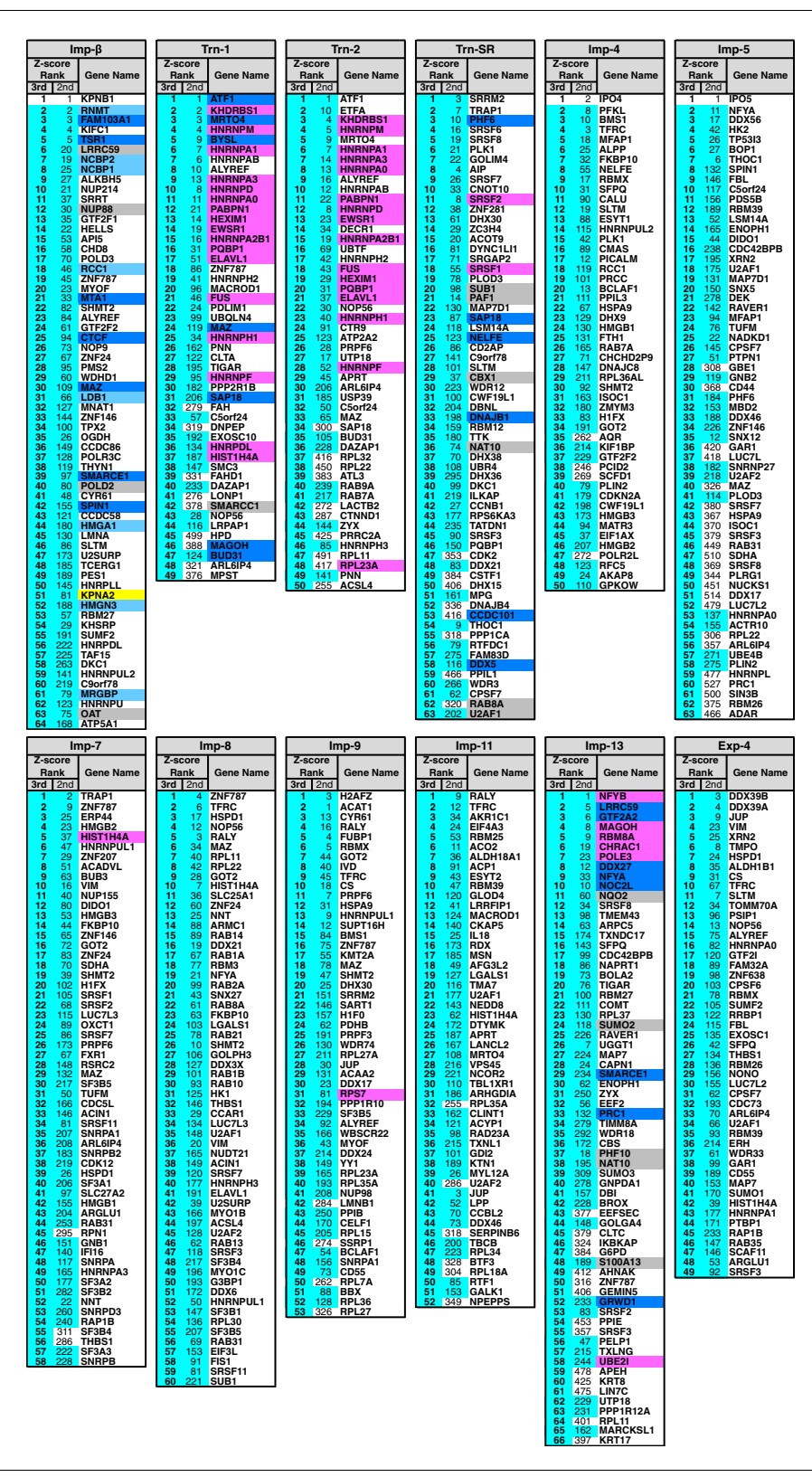

**Figure 3.** 3rd-Z-4% cargoes of the 12 NTRs. The 3rd-Z-4% cargoes of each NTR are listed by the gene names in the 3rd-Z-rank orders. The ranks by the second Z-scores are also shown. The 3rd-Z-4% and 2nd-Z-15% cargoes are indicated by cyan in the rank columns. Colors in the gene name columns: magenta, reported cargoes; blue, cargoes bound directly to the NTR in the bead halo assays (*Supplementary file 2*); light blue, cargoes bond directly to Imp-α but not to Imp-β; gray, proteins that did not bind to the NTRs; and yellow, Imp-α. For the 2nd-Z-15% cargoes, see *Supplementary file 3*.

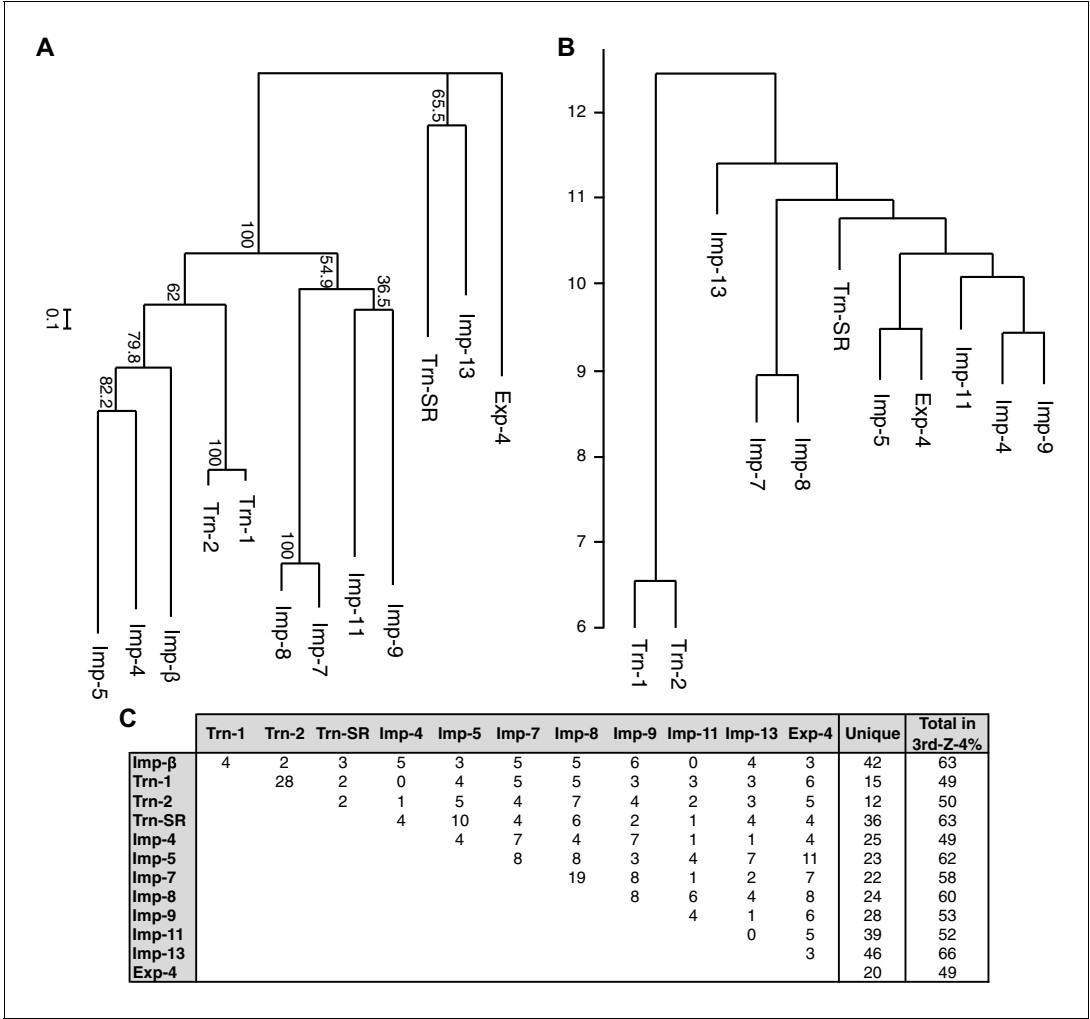

**Figure 4.** Phylogenetic tree and cargo profile hierarchical clustering of the Imp-β family import receptors. (A) Phylogenetic tree of the 12 Imp-β family import receptors with the bootstrap values. Scale bar indicates substitutions per site. (B) A hierarchical clustering dendrogram of the same NTRs (except Imp-β) based on the similarities of their 3rd-Z-4% cargo profiles. Imp-β was excluded because Imp-α connects to Imp-β and many of the identified cargoes. The scale indicates the intercluster distance. (C) The numbers of 3rd-Z-4% cargoes shared by two NTRs. For the 2nd-Z-15% cargoes, see **Supplementary file 5A**.

(*Figure 5*; *Supplementary file 9*), which are terms for biological processes (BPs). These cargoes are also annotated with chromatin, which is a term for cellular component (CC; *Figure 6A*; *Supplementary file 9*), and DNA binding, which is for molecular function (MF; *Figure 6B*; *Supplementary file 9*), all related to DNA recombination and DNA conformation change. For another example, the Trn-SR cargoes are significantly annotated with a range of terms for BPs that are related to cell division or nuclear division and terms for CCs that include condensed chromosome, kinetochore, spindle, and centrosome. Similarly, most of the examined NTRs are linked to terms for BPs via the 3rd-Z-4% cargoes (*Figure 5*) as follows: Imp-β, -4, -7, and Trn-SR are linked to chromatin or chromosome organization; Imp-β, -4, and -13 are linked to DNA repair; Imp-β is linked to mRNA capping; Trn-SR and Exp-4 are linked to mRNA polyadenylation; Trn-1 and -2 are linked to mRNA stabilization; Trn-2, -SR, Imp-5, -9, and Exp-4 are linked to ribosome biogenesis or rRNA processing; Trn-SR is linked to protein folding, modification, ubiquitination, and catabolic process; and Imp-4 and -7 are linked to apoptosis. The NTRs are also consistently linked to the terms for CCs and MFs (*Figure 6*) as follows: Trn-SR, Imp-5, -9, and Exp-4 are linked to Cajal body; Imp-β is linked to cap-binding complex; and Trn-SR is linked to pre-mRNA binding, snoRNA binding, and RNA helicase activity.

| Term ID | Term Name | Imp-β p | Imp-β # | Trn-1 p | Trn-1 # | Trn-2 p | Trn-2 # | Trn-SR p | Trn-SR # | Imp-4 p | Imp-4 # | Imp-5 p | Imp-5 # | Imp-7 p | Imp-7 # | Imp-8 p | Imp-8 # | Imp-9 p | Imp-9 # | Imp-11 p | Imp-11 # | Imp-13 p | Imp-13 # | Exp-4 p | Exp-4 # | Total No. |
|---|---|---|---|---|---|---|---|---|---|---|---|---|---|---|---|---|---|---|---|---|---|---|---|---|---|---|
| GO:0006259 | DNA Metabolic Process | 8E-06 | 11 | 0.199 | 6 | 1 | 2 | 0.146 | 7 | 0.002 | 8 | 1 | 4 | 0.632 | 6 | 1 | 2 | 0.34 | 6 | 1 | 2 | 2E-04 | 10 | 8E-06 | 10 | 978 |
| GO:0006310 | DNA Recombination | 0.503 | 4 | | | | | 1 | 2 | 0.008 | 5 | 1 | 1 | 0.019 | 5 | 1 | 2 | 1 | 1 | 1 | 1 | 1 | 3 | 0.008 | 5 | 298 |
| GO:0006323 | DNA Packaging | 1 | 3 | 1 | 1 | | | 1 | 2 | 0.039 | 4 | | | 0.003 | 5 | 1 | 2 | 1 | 1 | 1 | 1 | 1 | 1 | 1 | 1 | 200 |
| GO:0071103 | DNA Conformation Change | 0.312 | 4 | 1 | 1 | | | 1 | 2 | 1E-04 | 6 | | | 4E-04 | 6 | 0.244 | 4 | 1 | 1 | 1 | 1 | 1 | 1 | 1 | 1 | 263 |
| GO:0032392 | DNA Geometric Change | 1 | 2 | | | | | | | 4E-04 | 4 | | | 0.061 | 3 | 1 | 2 | | | | | | | | | 63 |
| GO:0010216 | Maintenance of DNA Methylation | 0.031 | 2 | | | | | | | | | | | | | | | | | | | | | | | 6 |
| GO:0000737 | DNA Catabolic Process, Endonucleolytic | | | | | | | | | 0.002 | 3 | | | 0.436 | 2 | | | | | 1 | 1 | | | | | 23 |
| GO:0006281 | DNA Repair | 0.002 | 7 | 1 | 3 | | | 1 | 2 | 0.007 | 6 | 1 | 1 | 1 | 4 | 1 | 1 | 1 | 3 | 1 | 2 | 0.043 | 6 | 0.111 | 5 | 519 |
| GO:0006289 | Nucleotide-Excision Repair | 0.475 | 3 | | | | | | | 0.005 | 4 | | | 1 | 1 | | | | | 1 | 1 | 0.017 | 4 | 1 | 1 | 116 |
| GO:0006284 | Base-Excision Repair | 0.043 | 3 | | | | | 1 | 1 | 1 | 2 | | | 1 | 1 | | | | | | | | | | | 52 |
| GO:0006325 | Chromatin Organization | 6E-07 | 11 | 1 | 3 | 1 | 3 | 0.313 | 6 | 0.062 | 6 | 1 | 5 | 1 | 4 | 1 | 1 | 0.943 | 5 | 1 | 4 | 0.374 | 6 | 1 | 4 | 768 |
| GO:0031497 | Chromatin Assembly | 1 | 3 | 1 | 1 | | | 1 | 1 | 1 | 1 | | | 0.035 | 4 | 1 | 1 | 1 | 1 | 1 | 1 | 1 | 1 | 1 | 1 | 162 |
| GO:0016568 | Chromatin Modification | 2E-05 | 9 | 1 | 2 | 1 | 1 | 0.967 | 5 | 1 | 3 | 0.967 | 5 | 1 | 1 | 1 | 1 | 1 | 2 | 1 | 3 | 1 | 5 | 1 | 4 | 614 |
| GO:0006338 | Chromatin Remodeling | 0.037 | 4 | 1 | 2 | | | | | 1 | 1 | 1 | 1 | 1 | 1 | 1 | 1 | | | 1 | 1 | 1 | 2 | 1 | 1 | 152 |
| GO:0070828 | Heterochromatin Organization | 7E-04 | 3 | | | | | | | 1 | 1 | | | | | | | | | | | | | | | 14 |
| GO:0051276 | Chromosome Organization | 4E-11 | 16 | 1 | 5 | 1 | 4 | 3E-06 | 12 | 2E-06 | 11 | 0.348 | 7 | 0.021 | 8 | 1 | 4 | 0.723 | 6 | 1 | 4 | 0.43 | 7 | 0.497 | 6 | 1126 |
| GO:0032200 | Telomere Organization | 0.031 | 4 | 1 | 2 | 1 | 1 | 1 | 3 | 1 | 1 | 1 | 1 | 1 | 1 | 1 | 1 | | | 1 | 1 | 1 | 1 | 0.414 | 3 | 145 |
| GO:0006368 | Transcription Elongation from RNA Polymerase II Promoter | 1E-04 | 5 | | | 1 | 1 | 1 | 2 | 0.118 | 3 | | | | | | | 1 | 2 | 1 | 1 | 1 | 2 | 1 | 1 | 95 |
| GO:0016458 | Gene Silencing | 1E-05 | 7 | 1 | 2 | 1 | 2 | 1 | 2 | 1 | 1 | 1 | 2 | 1 | 2 | 1 | 2 | 1 | 3 | 1 | 1 | | | 1 | 1 | 244 |
| GO:0034660 | ncRNA Metabolic Process | 3E-06 | 9 | 0.986 | 4 | 1 | 4 | 0.312 | 5 | 1 | 2 | 0.023 | 6 | 1 | 3 | 1 | 3 | 0.107 | 5 | 1 | 2 | 0.358 | 5 | 1 | 4 | 480 |
| GO:0034470 | ncRNA Processing | 0.046 | 5 | 0.241 | 4 | 0.307 | 4 | 0.054 | 5 | 1 | 1 | 0.054 | 5 | 1 | 1 | 1 | 3 | 0.018 | 5 | 1 | 2 | 0.954 | 4 | 0.272 | 4 | 331 |
| GO:0006370 | 7-Methylguanosine mRNA Capping | 6E-12 | 7 | | | | | | | 0.616 | 2 | | | | | | | | | | | | | | | 33 |
| GO:0000387 | Spliceosomal snRNP Assembly | 1 | 2 | | | 1 | 1 | | | | | | | 0.017 | 3 | | | 0.012 | 3 | | | 1 | 1 | | | 41 |
| GO:0000389 | mRNA 3'-Splice Site Recognition | | | | | | | | | | | | | 3E-05 | 3 | | | | | | | | | | | 14 |
| GO:0006378 | mRNA Polyadenylation | | | | | 1 | 1 | 0.018 | 3 | 1 | 1 | | | | | 1 | 1 | | | | | | | 0.007 | 3 | 37 |
| GO:0098789 | pre-mRNA Cleavage Required for Polyadenylation | 0.031 | 2 | | | | | | | | | | | | | | | | | | | | | | | 6 |
| GO:0000184 | Nuclear-Transcribed mRNA Catabolic Process, Nonsense-Mediated Decay | 1 | 2 | 1 | 2 | 0.006 | 4 | | | | | 1 | 1 | | | 0.454 | 3 | 3E-10 | 8 | 0.008 | 4 | 0.021 | 4 | | | 123 |
| GO:0048255 | mRNA Stabilization | 1 | 1 | 0.005 | 3 | 0.006 | 3 | | | | | 1 | 1 | | | 1 | 1 | | | | | | | 1 | 1 | 34 |
| GO:0010501 | RNA Secondary Structure Unwinding | | | | | | | 0.032 | 3 | | | 0.032 | 3 | | | 0.023 | 3 | 1 | 2 | 1 | 2 | 1 | 1 | 1 | 1 | 45 |
| GO:0042254 | Ribosome Biogenesis | 0.203 | 4 | 0.064 | 4 | 0.003 | 5 | 4E-04 | 6 | 1 | 2 | 0.236 | 4 | | | 0.158 | 4 | 5E-08 | 8 | 0.1 | 4 | 0.26 | 4 | 0.073 | 4 | 235 |
| GO:0006364 | rRNA Processing | 1 | 2 | 0.456 | 3 | 0.557 | 3 | 0.001 | 5 | 1 | 1 | 0.047 | 4 | | | 0.894 | 3 | 0.019 | 4 | 1 | 2 | 0.052 | 4 | 0.014 | 4 | 155 |
| GO:0042273 | Ribosomal Large Subunit Biogenesis | 1 | 1 | | | 1 | 2 | 1 | 2 | | | 1 | 1 | | | | | 0.014 | 3 | 1 | 1 | 1 | 1 | | | 43 |
| GO:0006412 | Translation | 1 | 3 | 0.333 | 5 | 1E-04 | 8 | 1 | 5 | 1 | 2 | 1 | 4 | 0.895 | 5 | 2E-06 | 10 | 1E-05 | 9 | 1 | 4 | 0.001 | 8 | 1 | 3 | 677 |
| GO:0006413 | Translational Initiation | 1 | 2 | 1 | 1 | 0.006 | 5 | 1 | 1 | 1 | 1 | 1 | 1 | | | 0.014 | 5 | 2E-07 | 8 | 1 | 3 | 1 | 2 | | | 273 |
| GO:0006414 | Translational Elongation | | | | | 0.062 | 4 | | | | | 1 | 2 | 1 | 1 | 1 | 3 | 3E-08 | 8 | 0.076 | 4 | 0.198 | 4 | | | 219 |
| GO:0006415 | Translational Termination | | | | | 0.028 | 4 | | | | | 1 | 1 | | | 1 | 3 | 6E-09 | 8 | 0.973 | 3 | 1 | 3 | | | 179 |
| GO:0006457 | Protein Folding | | | 1 | 1 | | | | | 0.012 | 5 | 1 | 3 | 1 | 1 | 0.162 | 4 | 1 | 2 | 1 | 2 | 1 | 2 | 1 | 1 | 240 |
| GO:0061077 | Chaperone-Mediated Protein Folding | | | | | | | | | 1 | 2 | 1 | 1 | | | 0.032 | 3 | 1 | 2 | 1 | 1 | | | 1 | 1 | 51 |
| GO:0010608 | Posttranscriptional Regulation of Gene Expression | 8E-04 | 7 | 0.003 | 6 | 0.068 | 5 | 0.017 | 6 | 1 | 3 | 1 | 3 | 1 | 3 | 0.01 | 6 | 1 | 1 | 1 | 1 | 0.277 | 5 | 0.902 | 4 | 454 |
| GO:0043412 | Macromolecule Modification | 0.024 | 15 | 1 | 7 | 1 | 5 | 2E-04 | 18 | 0.617 | 11 | 1 | 10 | 0.807 | 12 | 1 | 7 | 1 | 6 | 1 | 9 | 0.832 | 13 | 0.617 | 11 | 4139 |
| GO:0036211 | Protein Modification Process | 1 | 11 | 1 | 6 | 1 | 4 | 0.004 | 16 | 0.42 | 11 | 1 | 9 | 0.538 | 12 | 1 | 6 | 1 | 6 | 1 | 9 | 0.542 | 13 | 1 | 9 | 3959 |
| GO:0016567 | Protein Ubiquitination | 1 | 1 | | | 1 | 1 | 1 | 1 | 0.036 | 7 | 1 | 3 | 1 | 1 | 1 | 2 | | | 1 | 2 | 1 | 1 | 1 | 1 | 784 |
| GO:0018205 | Peptidyl-Lysine Modification | 0.007 | 6 | 1 | 1 | 1 | 1 | | | 1 | 2 | 1 | 2 | 1 | 1 | 1 | 1 | | | 1 | 2 | 0.149 | 5 | 1 | 1 | 398 |
| GO:0031145 | Anaphase-Promoting Complex-Dependent Proteasomal Ubiquitin-Dependent Protein Catabolic Process | | | | | | | | | 4E-04 | 5 | 1 | 2 | 1 | 2 | | | 1 | 2 | | | | | | | 120 |
| GO:0030163 | Protein Catabolic Process | 1 | 1 | 1 | 3 | 1 | 1 | | | 0.005 | 8 | 1 | 4 | 1 | 2 | 1 | 2 | | | 1 | 5 | 1 | 3 | 1 | 1 | 827 |
| GO:0007049 | Cell Cycle | 5E-05 | 13 | 1 | 4 | 1 | 3 | 0.004 | 11 | 0.15 | 8 | 1 | 6 | 1 | 7 | 1 | 7 | 1 | 4 | 1 | 6 | 0.15 | 8 | | | 1796 |
| GO:0000278 | Mitotic Cell Cycle | 2E-04 | 10 | 1 | 3 | 1 | 1 | 0.025 | 8 | 0.033 | 7 | 1 | 3 | 1 | 5 | 1 | 5 | 1 | 1 | 1 | 3 | 1 | 4 | 1 | 2 | 1038 |
| GO:0051301 | Cell Division | 1 | 4 | 1 | 1 | 1 | 1 | 0.001 | 8 | 1 | 4 | 1 | 2 | 1 | 2 | 1 | 1 | | | 1 | 1 | 1 | 2 | 1 | 1 | 672 |
| GO:0000075 | Cell Cycle Checkpoint | | | | | | | 5E-04 | 6 | 1 | 3 | 1 | 4 | 1 | 3 | | | | | | | | | 1 | 1 | 249 |
| GO:0071173 | Spindle Assembly Checkpoint | | | | | | | 4E-04 | 4 | 1 | 2 | 1 | 2 | | | | | | | | | | | | | 46 |
| GO:0007091 | Metaphase/Anaphase Transition of Mitotic Cell Cycle | | | | | | | 1E-03 | 4 | 1 | 2 | 1 | 2 | | | | | | | | | | | | | 59 |
| GO:0007059 | Chromosome Segregation | 1 | 3 | 1 | 1 | | | 0.024 | 5 | 0.141 | 4 | 1 | 1 | 1 | 2 | 1 | 1 | | | | | 1 | 2 | 1 | 1 | 279 |
| GO:0000280 | Nuclear Division | 0.042 | 6 | 1 | 1 | | | 0.004 | 7 | 1 | 4 | 1 | 1 | 1 | 2 | | | | | 1 | 1 | 1 | 3 | | | 550 |
| GO:0007077 | Mitotic Nuclear Envelope Disassembly | 0.026 | 3 | | | | | 1 | 2 | 1 | 1 | 1 | 1 | | | | | 1 | 1 | | | | | | | 44 |
| GO:0048856 | Anatomical Structure Development | 1 | 12 | 0.057 | 14 | 0.102 | 14 | 0.503 | 15 | 1 | 10 | 1 | 11 | 0.009 | 17 | 3E-06 | 22 | 1 | 12 | 1E-05 | 20 | 0.231 | 16 | 0.329 | 13 | 5272 |
| GO:0048731 | System Development | 1 | 11 | 0.224 | 12 | 1 | 11 | 1 | 13 | 1 | 8 | 1 | 10 | 0.107 | 14 | 0.035 | 15 | 1 | 10 | 0.034 | 14 | 1 | 13 | 1 | 10 | 4482 |
| GO:0048869 | Cellular Developmental Process | 1 | 8 | 0.461 | 10 | 0.735 | 11 | 1 | 12 | 0.025 | 13 | 1 | 6 | 1 | 11 | 7E-05 | 18 | 1 | 7 | 0.002 | 15 | 1 | 11 | 0.127 | 12 | 4418 |
| GO:0030154 | Cell Differentiation | 1 | 8 | 1 | 10 | 1 | 10 | 1 | 12 | 0.336 | 11 | 1 | 6 | 1 | 11 | 2E-04 | 17 | 1 | 7 | 0.001 | 15 | 1 | 10 | 0.069 | 12 | 3859 |
| GO:0001649 | Osteoblast Differentiation | 1 | 3 | | | | | | | 1 | 2 | 1 | 2 | 1 | 2 | | | 1 | 1 | 1 | 2 | 1 | 1 | 0.046 | 4 | 209 |
| GO:0003012 | Muscle System Process | 1 | 1 | 1 | 1 | 1 | 2 | 1 | 1 | | | | | 1 | 3 | 1 | 1 | 1 | 1 | 1 | 2 | 1 | 1 | 0.031 | 5 | 398 |
| GO:0012501 | Programmed Cell Death | 1 | 8 | 1 | 3 | 1 | 1 | 0.326 | 9 | 0.034 | 9 | 1 | 8 | 0.003 | 11 | 0.179 | 9 | 0.384 | 8 | 1 | 3 | 1 | 5 | 1 | 7 | 1921 |
| GO:0097194 | Execution Phase of Apoptosis | 1 | 1 | | | | | | | 1 | 1 | 0.122 | 3 | 0.004 | 4 | 1 | 2 | 1 | 2 | | | | | 1 | 1 | 96 |
| GO:0030262 | Apoptotic Nuclear Changes | | | | | | | | | 0.004 | 3 | | | 0.006 | 3 | 1 | 1 | 1 | 1 | | | | | | | 30 |
| GO:0006309 | Apoptotic DNA Fragmentation | | | | | | | | | 7E-04 | 3 | | | 0.264 | 2 | | | 1 | 1 | | | | | | | 18 |
| GO:0009607 | Response to Biotic Stimulus | 1 | 5 | 1 | 4 | 1 | 3 | 1 | 5 | 1 | 5 | 1 | 2 | 1 | 3 | 0.008 | 8 | 0.089 | 7 | 1 | 2 | 1 | 1 | 1 | 3 | 984 |
| GO:0006974 | Cellular Response to DNA Damage Stimulus | 0.037 | 7 | 1 | 4 | | | 1 | 5 | 3E-05 | 9 | 1 | 3 | 1 | 5 | 1 | 1 | 1 | 4 | 1 | 3 | 0.5 | 6 | 0.007 | 7 | 811 |
| GO:0006979 | Response to Oxidative Stress | | | 1 | 3 | 1 | 2 | 1 | 2 | 1 | 1 | | | 1 | 3 | 1 | 2 | | | 1 | 4 | 0.033 | 5 | | | 401 |
| GO:0070887 | Cellular Response to Chemical Stimulus | 1 | 9 | 0.103 | 10 | 0.842 | 9 | 1 | 10 | 1 | 6 | 1 | 9 | 0.124 | 11 | 0.16 | 11 | 1 | 6 | 0.007 | 12 | 2E-06 | 16 | 1 | 9 | 2852 |
| GO:0014070 | Response to Organic Cyclic Compound | 1 | 2 | 1 | 4 | 1 | 4 | 1 | 4 | 1 | 3 | 1 | 2 | 0.382 | 6 | 1 | 5 | 1 | 3 | 1E-04 | 9 | 1 | 4 | 1 | 4 | 890 |
| GO:1901698 | Response to Nitrogen Compound | 1 | 1 | 1 | 3 | 1 | 4 | 1 | 5 | | | 1 | 5 | 1 | 6 | 0.019 | 8 | 1 | 2 | 1 | 3 | 1 | 1 | 1 | 5 | 1087 |
| GO:1901700 | Response to Oxygen-Containing Compound | 1 | 3 | 1 | 5 | 1 | 6 | 1 | 6 | 1 | 4 | 1 | 5 | 0.006 | 10 | 0.058 | 9 | 1 | 6 | 0.021 | 9 | 1 | 4 | 0.086 | 8 | 1660 |
| GO:0010038 | Response to Metal Ion | 1 | 1 | 0.014 | 5 | 1 | 3 | 1 | 2 | 1 | 1 | | | 1 | 2 | 1 | 1 | 1 | 2 | 1 | 2 | 1 | 2 | | | 349 |
| GO:0006952 | Defense Response | 1 | 4 | 1 | 2 | 1 | 3 | 1 | 3 | 1 | 1 | 1 | 7 | 5 | | 0.018 | 10 | 0.144 | 9 | 1 | 3 | 1 | 3 | 1 | 6 | 1867 |
| GO:0002376 | Immune System Process | 1 | 6 | 1 | 3 | 1 | 4 | 1 | 6 | 0.002 | 12 | 1 | 6 | 0.069 | 11 | 0.003 | 13 | 1 | 5 | 1 | 7 | 1 | 6 | 1 | 8 | 2673 |
| GO:0032606 | Type I Interferon Production | 1 | 1 | | | | | 1 | 1 | 0.006 | 4 | | | 0.013 | 4 | 1 | 2 | | | | | | | | | 127 |
| GO:0007165 | Signal Transduction | 1 | 11 | 1 | 10 | 1 | 10 | 0.009 | 19 | 1 | 9 | 0.479 | 16 | 0.036 | 17 | 1E-04 | 21 | 1 | 9 | 0.009 | 17 | 1 | 13 | 0.244 | 14 | 5865 |
| GO:0007154 | Cell Communication | 1 | 12 | 1 | 12 | 1 | 10 | 0.008 | 20 | 1 | 11 | 0.119 | 18 | 0.03 | 18 | 2E-05 | 23 | 1 | 10 | 0.001 | 19 | 0.215 | 18 | 0.165 | 15 | 6420 |
| | No. in Top 4% | | 63 | | 49 | | 50 | | 63 | | 49 | | 62 | | 58 | | 60 | | 53 | | 52 | | 66 | | 49 | |

**Figure 5.** GO term (Biological Process) enrichments of the 3rd-Z-4% cargoes. The 3rd-Z-4% cargoes were analyzed for GO term (term type, Biological Process) enrichment. The significantly enriched terms (p<0.05, cyan) in the 3rd-Z-4% cargoes of four or fewer NTRs were selected, and a representative term for each group of highly similar is presented with their p-values and the numbers (#) of cargoes annotated with them. Total No. denotes the number of proteins annotated with each term in the database. Related terms are bundled in the same color. This table was extracted from

*Figure 5 continued on next page*

*Figure 5 continued*

***Supplementary file 6B***. All the GO terms annotated to the 3rd-Z-4% cargoes are listed in ***Supplementary file 7***. The correspondence between each 3rd-Z-4% cargo and GO term is summarized in ***Supplementary file 9***. For the 2nd-Z-15% cargoes, see ***Supplementary files 6A, 8,*** and ***10***.

Nearly twice as many GO terms were annotated to the 2nd-Z-15% cargoes. The correspondences between the 2nd-Z-15% cargoes of the 12 NTRs and selected representative GO terms enriched significantly for them are tabulated in ***Supplementary file 10***. The excerpted list of terms enriched significantly for the cargoes of four or fewer NTRs contains terms partially different from those in ***Figures 5*** and ***6*** (***Supplementary file 6A***), but many of the NTRs are still linked to terms similar to those of the 3rd-Z-4% cargoes. For example, Imp-$\beta$ and Trn-SR are linked to terms related to cell or nuclear division by the 3rd-Z-4% cargoes (***Figure 5***) and to partially different terms that are still related to cell or nuclear division by the 2nd-Z-15% cargoes (***Supplementary file 6A***). Additionally, Imp-4 is linked to terms related to DNA structure regulation, DNA repair, and apoptosis in both lists. Similarly, most of the examined NTRs are linked in both lists to similar terms that are related to any of the following: chromatin organization, chromosome organization, DNA repair, ribosome biogenesis, protein modification, cell division, nuclear division, and apoptosis. Thus, we regard the 3rd-Z-4% list as a core table of the cargo roles. Naturally, the 2nd-Z-15% cargoes linked the NTRs to additional terms (***Supplementary file 6A***) as follows: Imp-$\beta$, -4, and Trn-SR are linked to DNA-dependent DNA replication; Trn-1 and Imp-7 are linked to gene silencing by RNA; Imp-$\beta$, -7, and -13 are linked to rRNA transcription; Imp-$\beta$ and Trn-SR are linked to protein methylation; Trn-SR is linked to protein peptidyl-prolyl isomerization; Trn-2, Imp-4, and -8 are linked to circadian rhythm; Imp-$\beta$, -4, -13, and Trn-SR are linked to terms for CCs and MFs that are related to RNA polymerase (RNAP) II transcription; and subsets of the NTRs are linked to varying terms that are related to differentiation, development, and response. As an important result, we have illustrated the general framework of the division of roles among the NTRs for the first time, in which one NTR is linked to many BPs and conversely each broadly defined BP is supported by many NTRs, but each closely defined BP is supported by a restricted number of NTRs. One typical example is the allocation of mRNA processing factors (see below).

## Allocation of mRNA processing factors to the NTRs

Some of the GO terms related to mRNA processing were specifically linked to four or fewer NTRs by the 3rd-Z-4% and 2nd-Z-15% cargoes (***Figure 5***; ***Supplementary file 6A***). However, many other terms related to mRNA processing were linked to more NTRs, and conversely, all the NTRs were implicated in mRNA processing. The 2nd- and 3rd-Z-rankings for the 12 NTRs included 275 and 242 proteins, respectively, that were annotated with mRNA processing (***Supplementary file 6B and 6C***). To see the allocation of these proteins to the NTRs, the ranks of these proteins are arranged in a table (***Supplementary file 11A***). The 2nd- and 3rd-Z-rankings revealed similar results. As summarized for the 2nd-Z-ranking (***Figure 7***), particular groups of the mRNA-processing factors are allocated to specific NTRs, showing that each NTR is linked to distinct reactions in mRNA processing: the proteins related to mRNA capping are allocated to Imp-$\beta$ almost exclusively; hnRNP A0 is allocated to Trn-1, -2, Imp-4, -11, and others; hnRNP A1, A2B1, A3, D, F, H1–3, and M are allocated to Trn-1 and -2, and additionally Imp-9 and Exp-4; hnRNP U-like 1 are allocated to Imp-7, -8, and -9; SR-rich SFs are primarily allocated to Trn-SR and secondarily to Imp-7, -8, and -9; SFs 3A and B are allocated to Imp-4, -7, -8, -9, -11, and Exp-4; PQ-rich SF is allocated to Imp-4, -7, -8, and Exp-4; snRNP A–C is allocated to Imp-4, -7, -8, and Exp-4; exon junction complex (EJC) components are exclusively allocated to Imp-11 and -13; cleavage and polyadenylation specificity factor (CPSF) 1 is allocated to Imp-7 and -9; CPSF5 (NUDT21), 6, and 7 are less specifically allocated to other NTRs; general transcription factor IIF is allocated to Imp-$\beta$, -4, and Trn-SR; and RNAP II associating factors are allocated to Trn-SR and separately to other NTRs. Thus, the NTRs import distinctive subsets of mRNA processing factors. In the 2nd-Z-ranking, the SR-rich SFs were not allocated to Imp-5, but they were identified as the Imp-5 3rd-Z-4% cargoes. Thus, subsets of proteins involved in a broadly defined BP, for example, mRNA processing, are allocated to different NTRs, in a manner representative of role division among the NTRs.

**A**

| Term ID | Term Name | Imp-β p | # | Trn-1 p | # | Trn-2 p | # | Trn-SR p | # | Imp-4 p | # | Imp-5 p | # | Imp-7 p | # | Imp-8 p | # | Imp-9 p | # | Imp-11 p | # | Imp-13 p | # | Exp-4 p | # | Total No. |
|---|---|---|---|---|---|---|---|---|---|---|---|---|---|---|---|---|---|---|---|---|---|---|---|---|---|---|
| GO:0008622 | Epsilon DNA Polymerase Complex | | | | | | | | | | | | | | | | | | | | | 0.024 | 2 | | | 5 |
| GO:0044452 | Nucleolar Part | 1 | 2 | 1 | 1 | 0.033 | 3 | 0.076 | 3 | 1 | 1 | 0.076 | 3 | | | 1 | 2 | | | | | 1 | 1 | 3E-04 | 4 | 60 |
| GO:0015030 | Cajal Body | 1 | 1 | | | | | 6E-04 | 4 | | | 6E-04 | 4 | | | 1 | 1 | 0.024 | 3 | 1 | 2 | | | 0.019 | 3 | 52 |
| GO:0042382 | Paraspeckles | | | 1 | 1 | 1 | 1 | | | | | | | 1 | 1 | 1 | 1 | | | | | 1 | 1 | 2E-05 | 3 | 6 |
| GO:0034399 | Nuclear Periphery | 1 | 1 | 1 | 2 | 1 | 1 | 1 | 1 | 0.249 | 3 | 1 | 1 | | | 1 | 1 | 0.31 | 3 | 1 | 1 | 1 | 2 | 0.006 | 4 | 122 |
| GO:0005635 | Nuclear Envelope | 2E-05 | 8 | 1 | 1 | 1 | 1 | 1 | 1 | 1 | 2 | 1 | 1 | 1 | 1 | 1 | 1 | 1 | 3 | 1 | 1 | 0.172 | 5 | 1 | 2 | 410 |
| GO:0000785 | Chromatin | 0.012 | 6 | 0.724 | 4 | 1 | 1 | 1 | 2 | 1E-04 | 7 | 1 | 3 | 1 | 3 | 1 | 2 | 0.073 | 5 | 1 | 2 | 1 | 3 | 0.003 | 6 | 442 |
| GO:0000791 | Euchromatin | | | | | | | | | | | | | | | | | 0.005 | 3 | | | 1 | 1 | 1 | 1 | 31 |
| GO:0098687 | Chromosomal Region | 1 | 3 | 1 | 2 | | | 2E-06 | 8 | 1 | 1 | 1 | 3 | 1 | 3 | 1 | 1 | | | 1 | 1 | 1 | 3 | 1 | 3 | 301 |
| GO:0000793 | Condensed Chromosome | 1 | 3 | 1 | 1 | | | 0.004 | 5 | 0.032 | 4 | 1 | 1 | 0.067 | 4 | | | | | | | 1 | 1 | | | 191 |
| GO:0000776 | Kinetochore | | | | | | | 0.015 | 4 | 1 | 1 | 1 | 1 | 1 | 2 | | | | | | | 1 | 2 | | | 116 |
| GO:0005819 | Spindle | 1 | 3 | 1 | 1 | | | 4E-05 | 7 | 1 | 1 | 0.455 | 4 | 1 | 2 | | | | | 1 | 2 | 1 | 2 | | | 279 |
| GO:0005813 | Centrosome | | | 1 | 1 | 1 | 1 | 0.001 | 7 | 1 | 2 | | | | | 1 | 2 | | | 1 | 1 | 1 | 2 | | | 476 |
| GO:0090575 | RNA Polymerase II Transcription Factor Complex | 0.006 | 4 | 1 | 2 | 1 | 2 | | | | | 1 | 1 | | | 1 | 1 | | | | | 0.008 | 4 | 1 | 2 | 96 |
| GO:0016602 | CCAAT-Binding Factor Complex | | | | | | | | | | | | | 1 | 1 | 1 | 1 | | | | | 0.014 | 2 | | | 4 |
| GO:0005845 | mRNA Cap Binding Complex | 1E-06 | 4 | | | | | | | | | | | | | | | | | | | | | | | 12 |
| GO:0005732 | Small Nucleolar Ribonucleoprotein Complex | | | 1 | 1 | 1 | 1 | 1 | 1 | | | 0.441 | 2 | | | 1 | 1 | | | | | | | 0.001 | 3 | 20 |
| GO:0030532 | Small Nuclear Ribonucleoprotein Complex | | | | | 1 | 1 | | | | | 1 | 2 | 1E-19 | 11 | 0.054 | 3 | 4E-06 | 5 | | | | | 1 | 1 | 60 |
| GO:0097525 | Spliceosomal snRNP Complex | | | | | 1 | 1 | | | | | 1 | 2 | 3E-20 | 11 | 0.037 | 3 | 2E-06 | 5 | | | | | 1 | 1 | 53 |
| GO:0005684 | U2-Type Spliceosomal Complex | | | | | | | | | | | 0.008 | 3 | 5E-10 | 6 | 0.006 | 3 | | | 1 | 1 | | | 1 | 1 | 29 |
| GO:0089701 | U2AF | | | | | | | 1 | 1 | | | 0.002 | 2 | | | 0.002 | 2 | | | 0.001 | 2 | | | 1 | 1 | 2 |
| GO:0005849 | mRNA Cleavage Factor Complex | | | | | | | 1 | 1 | | | 1 | 1 | 1 | 1 | | | | | | | | | 3E-04 | 3 | 14 |
| GO:0044391 | Ribosomal Subunit | | | | | 0.019 | 4 | | | 1 | 1 | 1 | 1 | | | 1 | 3 | 3E-09 | 8 | 0.727 | 3 | 1 | 2 | | | 162 |
| GO:0031428 | Box C/D snoRNP Complex | | | 1 | 1 | 1 | 1 | | | | | 1 | 1 | | | 1 | 1 | | | | | | | 0.018 | 2 | 6 |
| GO:0005856 | Cytoskeleton | 0.293 | 9 | 1 | 5 | 1 | 3 | 2E-04 | 13 | 1 | 5 | 1 | 7 | 1 | 4 | 1 | 6 | 1 | 5 | 0.001 | 11 | 0.014 | 11 | 1 | 4 | 1949 |
| GO:0010494 | Cytoplasmic Stress Granule | 1 | 1 | 1 | 1 | 1 | 1 | | | | | 1 | 1 | 1 | 1 | | | 0.008 | 3 | | | | | | | 32 |
| GO:0044445 | Cytosolic Part | | | | | 0.048 | 4 | | | 1 | 2 | 1 | 1 | | | 1 | 3 | 2E-08 | 8 | 1 | 3 | | | | | 205 |
| GO:0005739 | Mitochondrion | 0.622 | 8 | 1 | 5 | 1 | 4 | 1 | 4 | 0.102 | 8 | 1 | 5 | 9E-04 | 11 | 1E-05 | 13 | 3E-04 | 11 | 0.003 | 10 | 0.966 | 8 | 0.102 | 8 | 1699 |
| GO:0005783 | Endoplasmic Reticulum | 1 | 2 | 1 | 3 | 1 | 4 | 1 | 1 | 1 | 5 | 1 | 3 | 1 | 6 | 8E-04 | 11 | 1 | 1 | 1 | 2 | 1 | 4 | 1 | 5 | 1623 |
| GO:0005768 | Endosome | 1 | 1 | 1 | 1 | 1 | 2 | 1 | 4 | 1 | 3 | 1 | 4 | 1 | 3 | 2E-08 | 12 | 1 | 1 | 1 | 2 | 1 | 1 | 1 | 3 | 779 |
| GO:0005794 | Golgi Apparatus | | | 1 | 2 | 1 | 2 | 1 | 3 | 1 | 5 | 1 | 2 | 1 | 2 | 2E-04 | 11 | 1 | 3 | 1 | 3 | 1 | 5 | 1 | 5 | 1436 |
| GO:0031410 | Cytoplasmic Vesicle | 1 | 1 | 1 | 2 | 1 | 2 | 1 | 4 | 1 | 5 | 1 | 3 | 1 | 4 | 3E-07 | 13 | 1 | 4 | 1 | 2 | 1 | 1 | 1 | 5 | 1232 |
| GO:0005925 | Focal Adhesion | 1 | 2 | 1 | 1 | 1 | 2 | | | | | 1 | 1 | 1 | 4 | 1 | 1 | 0.004 | 6 | 0.002 | 6 | 0.039 | 5 | 0.008 | 6 | 383 |
| GO:0005905 | Coated Pit | | | 1 | 1 | | | | | | | 1 | 2 | 1 | 1 | 1 | 2 | 1 | 1 | 1 | 1 | | | 0.033 | 3 | 62 |
| GO:0043209 | Myelin Sheath | 1 | 1 | | | | | | | | | 1 | 2 | 1 | 3 | 0.047 | 4 | 1 | 2 | 1 | 2 | 0.032 | 4 | 1 | 1 | 175 |
| | No. in Top 4% | | 63 | | 49 | | 50 | | 63 | | 49 | | 62 | | 58 | | 60 | | 53 | | 52 | | 66 | | 49 | |

**B**

| Term ID | Term Name | Imp-β p | # | Trn-1 p | # | Trn-2 p | # | Trn-SR p | # | Imp-4 p | # | Imp-5 p | # | Imp-7 p | # | Imp-8 p | # | Imp-9 p | # | Imp-11 p | # | Imp-13 p | # | Exp-4 p | # | Total No. |
|---|---|---|---|---|---|---|---|---|---|---|---|---|---|---|---|---|---|---|---|---|---|---|---|---|---|---|
| GO:0034061 | DNA Polymerase Activity | 0.008 | 3 | | | | | 1 | 1 | | | | | | | | | | | | | 1 | 2 | | | 30 |
| GO:0003690 | Double-Stranded DNA Binding | 0.023 | 7 | 1 | 4 | 1 | 4 | 1 | 2 | 0.62 | 5 | 1 | 5 | 0.153 | 6 | 1 | 3 | 1 | 4 | | | 1 | 2 | 0.62 | 5 | 752 |
| GO:0003697 | Single-Stranded DNA Binding | 1 | 1 | 0.001 | 4 | 1 | 2 | 1 | 1 | 1 | 2 | | | 0.164 | 3 | 1 | 2 | 1 | 1 | 1 | 2 | | | 1 | 2 | 88 |
| GO:0098847 | Sequence-Specific Single Stranded DNA Binding | | | 0.039 | 2 | 0.046 | 2 | | | | | | | | | | | | | | | | | 1 | 1 | 9 |
| GO:0008301 | DNA Binding, Bending | | | | | | | | | 0.001 | 3 | | | 0.002 | 3 | | | | | | | | | | | 20 |
| GO:0043566 | Structure-Specific DNA Binding | 0.289 | 3 | 1 | 1 | | | 1 | 1 | 0.002 | 4 | | | 0.226 | 3 | | | 0.162 | 3 | | | | | 1 | 1 | 98 |
| GO:0000217 | DNA Secondary Structure Binding | | | | | | | | | 7E-04 | 3 | | | 0.001 | 3 | | | 1 | 1 | | | | | | | 18 |
| GO:0000400 | Four-Way Junction DNA Binding | | | | | | | | | 5E-05 | 3 | | | 9E-05 | 3 | | | 1 | 1 | | | | | | | 8 |
| GO:0097100 | Supercoiled DNA Binding | | | | | | | | | 0.007 | 2 | | | 0.011 | 2 | | | | | | | | | 1 | 1 | 4 |
| GO:0003682 | Chromatin Binding | 8E-11 | 12 | 1 | 3 | 1 | 2 | 1 | 4 | 0.061 | 5 | 1 | 2 | 1 | 2 | 1 | 3 | 1E-05 | 8 | 1 | 1 | 1 | 4 | 0.924 | 4 | 457 |
| GO:0042162 | Telomeric DNA Binding | | | 0.003 | 3 | 0.004 | 3 | | | | | | | | | | | | | | | 1 | 1 | 1 | 1 | 30 |
| GO:0001047 | Core Promoter Binding | 1 | 3 | 1 | 1 | 1 | 2 | 1 | 2 | 1 | 2 | 1 | 2 | 1 | 2 | 0.91 | 3 | 0.019 | 4 | | | 1 | 2 | 0.514 | 3 | 156 |
| GO:0000988 | Transcription Factor Activity, Protein Binding | 3E-04 | 8 | 0.012 | 6 | 0.227 | 5 | 0.006 | 7 | 1 | 1 | 1 | 2 | 1 | 2 | 1 | 2 | 1 | 4 | 1 | 3 | 1 | 4 | 1 | 2 | 587 |
| GO:0003727 | Single-Stranded RNA Binding | 1 | 2 | 2E-08 | 6 | 4E-04 | 4 | | | 1 | 1 | 1 | 1 | 1 | 2 | 1 | 1 | 1 | 2 | 1 | 2 | 1 | 1 | 1 | 2 | 62 |
| GO:0036002 | Pre-mRNA Binding | | | 1 | 1 | 1 | 1 | 0.004 | 3 | | | 1 | 1 | 1 | 1 | 1 | 1 | | | 1 | 1 | 1 | 1 | 1 | 1 | 23 |
| GO:0071208 | Histone Pre-mRNA DCP Binding | | | | | | | | | | | | | 0.011 | 2 | | | | | | | | | | | 4 |
| GO:0003730 | mRNA 3'-UTR Binding | 1 | 1 | 0.015 | 3 | 0.018 | 3 | | | | | | | | | 1 | 1 | | | | | | | | | 49 |
| GO:0017091 | AU-rich Element Binding | | | 0.001 | 3 | 0.002 | 3 | | | | | | | 1 | 1 | | | 0.432 | 2 | | | | | 1 | 1 | 23 |
| GO:0017069 | snRNA Binding | 1 | 1 | 1 | 1 | 1 | 1 | 1 | 1 | | | | | 3E-07 | 5 | 1 | 1 | 1 | 1 | | | 1 | 1 | 1 | 1 | 34 |
| GO:0035614 | snRNA Stem-Loop Binding | | | | | | | | | | | | | 0.002 | 2 | | | | | | | | | | | 2 |
| GO:0004004 | ATP-Dependent RNA Helicase Activity | | | | | | | | | 2E-07 | 6 | 1 | 1 | 0.102 | 3 | | | 0.001 | 4 | 0.05 | 3 | 1 | 2 | 1 | 1 | 66 |
| GO:0030515 | snoRNA Binding | 1 | 1 | 1 | 1 | | | 1 | 1 | 0.005 | 3 | 1 | 1 | 0.471 | 2 | | | | | | | | | 0.323 | 2 | 24 |
| GO:0051082 | Unfolded Protein Binding | | | 1 | 1 | | | | | 0.009 | 4 | 1 | 1 | 1 | 1 | 1 | 2 | 1 | 2 | 1 | 1 | 1 | 1 | 1 | 1 | 102 |
| GO:0019789 | SUMO Transferase Activity | | | | | | | | | | | | | | | | | | | | | 4E-04 | 3 | | | 11 |
| GO:0019901 | Protein Kinase Binding | 1 | 1 | 1 | 2 | 1 | 3 | 0.046 | 6 | 1 | 2 | 1 | 3 | 1 | 2 | 1 | 3 | 1 | 1 | 1 | 3 | 1 | 4 | 1 | 3 | 543 |
| GO:0003988 | Acetyl-CoA C-Acyltransferase Activity | | | | | | | | | | | | | | | | | 0.021 | 2 | | | | | | | 6 |
| GO:0016887 | ATPase Activity | 1 | 4 | 1 | 1 | 1 | 1 | 0.008 | 6 | 1 | 1 | 1 | 1 | 1 | 2 | 1E-05 | 8 | 1 | 3 | 1 | 3 | 1 | 1 | 1 | 3 | 402 |
| GO:0003924 | GTPase Activity | | | | | 1 | 3 | 1 | 1 | 1 | 1 | 1 | 3 | 0.126 | 4 | 2E-09 | 9 | | | | | 1 | 2 | 1 | 2 | 225 |
| | No. in Top 4% | | 63 | | 49 | | 50 | | 63 | | 49 | | 62 | | 58 | | 60 | | 53 | | 52 | | 66 | | 49 | |

**Figure 6.** GO term (Cellular Component and Molecular Function) enrichments of the 3rd-Z-4% cargoes. The 3rd-Z-4% cargoes were analyzed, and the results are presented in a format similar to that of *Figure 5*. (A) Term type, Cellular Component. (B) Term type, Molecular Function. These tables were extracted from *Supplementary file 6B*. All the GO terms annotated to the 3rd-Z-4% cargoes are listed in *Supplementary file 7*. The correspondence between each 3rd-Z-4% cargo and GO term is summarized in *Supplementary file 9*. For the 2nd-Z-15% cargoes, see *Supplementary file 6A*, *8,* and *10*.

| Accession | Major Feature | Gene Name | Rank by 2nd Z-score | | | | | | | | | | | |
|---|---|---|---|---|---|---|---|---|---|---|---|---|---|---|
| | | | Imp-β | Trn-1 | Trn-2 | Trn-SR | Imp-4 | Imp-5 | Imp-7 | Imp-8 | Imp-9 | Imp-11 | Imp-13 | Exp-4 |
| O43148 | mRNA Capping | RNMT | 2 | 1432 | 1420 | 1906 | 358 | 327 | 1487 | 1066 | 687 | 1560 | 598 | 447 |
| Q09161 | Cap-Binding | NCBP1 | 25 | 327 | 1037 | 1373 | 751 | 1466 | 906 | 1220 | 655 | 1248 | 594 | 1075 |
| P52298 | | NCBP2 | 19 | | | 1416 | | 1529 | 865 | | | 1510 | 602 | |
| Q13151 | hnRNP | HNRNPA0 | 247 | 11 | 13 | 145 | 50 | 137 | 92 | 422 | 105 | 7 | 558 | 82 |
| P09651 | | HNRNPA1 | 245 | 7 | 7 | 285 | 364 | 776 | 323 | 438 | 171 | 773 | 641 | 177 |
| P22626 | | HNRNPA2B1 | 301 | 16 | 19 | 382 | 555 | 902 | 336 | 420 | 160 | 711 | 531 | 123 |
| P51991 | | HNRNPA3 | 634 | 13 | 14 | 531 | 260 | 823 | 165 | 163 | 182 | 405 | 987 | 210 |
| P07910 | | HNRNPC | 496 | 857 | 259 | 386 | 324 | 1140 | 337 | 570 | 117 | 563 | 676 | 229 |
| Q14103 | | HNRNPD | 851 | 8 | 8 | 566 | 623 | 1300 | 565 | 938 | 709 | 1237 | 1512 | 633 |
| P52597 | | HNRNPF | 1678 | 95 | 52 | 1702 | 382 | 211 | 268 | 230 | 608 | 1289 | 1251 | 304 |
| P31943 | | HNRNPH1 | 438 | 34 | 40 | 948 | 171 | 723 | 233 | 325 | 243 | 628 | 646 | 202 |
| P55795 | | HNRNPH2 | 741 | 41 | 42 | 1452 | 1126 | 534 | 363 | 280 | 302 | 802 | 573 | 208 |
| P31942 | | HNRNPH3 | 373 | 167 | 85 | 753 | 708 | 686 | 221 | 177 | 177 | 394 | 404 | 184 |
| P61978 | | HNRNPK | 509 | 1130 | 331 | 351 | 611 | 919 | 459 | 569 | 318 | 982 | 580 | 315 |
| P14866 | | HNRNPL | 453 | 491 | 125 | 634 | 154 | 477 | 532 | 621 | 148 | 999 | 921 | 236 |
| Q8WVV9 | | HNRPLL | 145 | 1397 | 1430 | 1331 | 1281 | 774 | 320 | 542 | 479 | 1018 | 1056 | 212 |
| P52272 | | HNRNPM | 403 | 4 | | 294 | 532 | 754 | 618 | 723 | 216 | 333 | 777 | 292 |
| O60506 | | SYNCRIP | 770 | 866 | 483 | 1123 | 503 | 508 | 435 | 529 | 575 | 808 | 628 | 693 |
| O43390 | | HNRNPR | 471 | 1421 | 133 | 374 | 1269 | 598 | 463 | 719 | 293 | 1236 | 674 | 388 |
| Q00839 | | HNRNPU | 123 | 837 | 395 | 415 | 1025 | 759 | 1247 | 475 | | 1133 | 645 | 556 |
| Q9BUJ2 | | HNRNPUL1 | 572 | 1600 | 1586 | 1503 | 354 | 1382 | 47 | 50 | 9 | 1253 | 1323 | 100 |
| Q8IYB3 | SR-Rich | SRRM1 | | 1440 | 1040 | 1977 | 863 | 1842 | | | 276 | 6 | | 723 |
| Q9UQ35 | Splicing Factor | SRRM2 | 202 | 203 | 147 | 3 | 1412 | 1463 | 249 | 54 | 151 | 803 | 1872 | 355 |
| Q07955 | | SRSF1 | 1617 | 328 | 113 | 55 | 984 | 1159 | 105 | 84 | 238 | 490 | 786 | 145 |
| Q01130 | | SRSF2 | 133 | 234 | 143 | 8 | 156 | 628 | 68 | 85 | 152 | 373 | 83 | 164 |
| P84103 | | SRSF3 | 274 | 294 | 254 | 90 | 243 | 379 | 110 | 118 | 49 | 236 | 357 | 92 |
| Q13243 | | SRSF5 | 1002 | 213 | 378 | 524 | 538 | 833 | 457 | 659 | 212 | 1504 | 644 | 379 |
| Q13247 | | SRSF6 | 1964 | 212 | 279 | 16 | 740 | 537 | 288 | 446 | 124 | 1091 | 1728 | 287 |
| Q16629 | | SRSF7 | 377 | 553 | 592 | 26 | 200 | 380 | 86 | 120 | 85 | 224 | 232 | 79 |
| Q9BRL6 | | SRSF8 | 115 | 738 | 1244 | 19 | | 369 | | | | | 34 | |
| Q05519 | | SRSF11 | 1194 | 342 | 162 | 1764 | 221 | 1199 | 81 | 81 | 93 | 338 | 1576 | 185 |
| Q15459 | Splicing Factor 3A, B | SF3A1 | 1621 | 442 | 366 | 1667 | 118 | 982 | 206 | 228 | 190 | 184 | 1313 | 150 |
| Q15428 | | SF3A2 | 1623 | 402 | 320 | 1644 | 126 | 782 | 177 | 209 | 202 | 106 | 1414 | 133 |
| Q12874 | | SF3A3 | 1650 | 375 | 291 | 1674 | 114 | 1021 | 222 | 243 | 231 | 227 | 1367 | 160 |
| O75533 | | SF3B1 | 1660 | 343 | 273 | 1542 | 81 | 862 | 211 | 147 | 188 | 56 | 1326 | 84 |
| Q13435 | | SF3B2 | 1638 | 432 | 276 | 1522 | 143 | 1000 | 282 | 206 | 178 | 214 | 1491 | 151 |
| Q15393 | | SF3B3 | 1604 | 322 | 288 | 1516 | 172 | 999 | 316 | 352 | 192 | 71 | 1191 | 207 |
| Q15427 | | SF3B4 | 1552 | 458 | 214 | 1566 | 215 | 1029 | 311 | 217 | 180 | 126 | 843 | 217 |
| Q9BWJ5 | | SF3B5 | 1634 | 418 | 247 | 1478 | 265 | 1077 | 217 | 207 | 229 | 77 | 1317 | 188 |
| Q9Y3B4 | | SF3B14 | 1663 | 336 | 246 | 1685 | 43 | 912 | 181 | 188 | 330 | 148 | 1483 | 131 |
| Q01081 | Splicing Factor U2AF | U2AF1 | 190 | 154 | 201 | 202 | 329 | 175 | 99 | 148 | 48 | 177 | 1476 | 66 |
| P26368 | | U2AF2 | 261 | 360 | 342 | 1293 | 220 | 218 | 111 | 128 | 76 | 286 | 1107 | 106 |
| P23246 | PQ-Rich SF | SFPQ | 111 | 219 | 1209 | 1890 | 31 | 1831 | 74 | 24 | 547 | 1542 | 143 | 42 |
| P09012 | snRNP A-F | SNRPA | 1030 | 444 | 400 | 218 | 459 | 1063 | 117 | 142 | 218 | 415 | 1887 | 361 |
| P09661 | | SNRPA1 | 1463 | 338 | 316 | 1664 | 112 | 976 | 207 | 231 | 156 | 107 | 1475 | 141 |
| P14678 | | SNRPB | 764 | 333 | 222 | 1540 | 238 | 795 | 228 | 143 | 210 | 226 | 736 | 219 |
| P08579 | | SNRPB2 | 1422 | 311 | 257 | 1603 | 84 | 1050 | 183 | 216 | 144 | 268 | 1510 | 85 |
| P09234 | | SNRPC | 354 | 555 | 393 | 915 | 411 | 825 | 130 | 170 | 272 | 310 | 710 | 91 |
| P62314 | | SNRPD1 | 743 | 300 | 241 | 1271 | 166 | 896 | 359 | 441 | 196 | 481 | 1602 | 364 |
| P62316 | | SNRPD2 | 1199 | 604 | 238 | 1646 | 284 | 1308 | 352 | 493 | 311 | 342 | 1084 | 227 |
| P62318 | | SNRPD3 | 843 | 384 | 226 | 1523 | 549 | 861 | 260 | 162 | 217 | 282 | 631 | 256 |
| P62304 | | SNRPE | 1512 | 357 | 203 | 1399 | 230 | 948 | 285 | 293 | 241 | 396 | 1458 | 431 |
| P62306 | | SNRPF | 584 | 426 | 466 | 893 | 73 | 1191 | 171 | 266 | 223 | 240 | 562 | 329 |
| Q9Y5S9 | Exon Junction Complex | RBM8A | 497 | 517 | 505 | 976 | 783 | 1857 | 1614 | 1052 | 281 | 109 | 9 | 570 |
| P38919 | | EIF4A3 | 629 | 289 | 392 | 256 | 394 | 998 | 535 | 852 | 490 | 24 | 737 | 681 |
| P61326 | | MAGOH | 1643 | 388 | 329 | 1183 | 484 | 1414 | 929 | 975 | 402 | 841 | 8 | 450 |
| Q10570 | Cleavage and Polyadenylation Specificity Factor | CPSF1 | 1029 | 690 | 1105 | 1481 | 231 | 1950 | 48 | 1236 | 56 | 1586 | 1837 | 114 |
| Q9P2I0 | | CPSF2 | 825 | 578 | 351 | 309 | 1434 | 1326 | 917 | 1541 | 530 | 99 | 1856 | 1730 |
| Q9UKF6 | | CPSF3 | 1104 | 1348 | 274 | 1367 | 1270 | 1666 | 1358 | 1664 | 663 | | 1639 | 1359 |
| O43809 | | NUDT21 | 415 | 335 | 99 | 157 | 593 | 253 | 254 | 165 | 154 | 590 | 1478 | 254 |
| Q16630 | | CPSF6 | 281 | 647 | 368 | 125 | 201 | 289 | 125 | 96 | 121 | 738 | 687 | 103 |
| Q8N684 | | CPSF7 | 187 | 521 | 191 | 62 | 421 | 145 | 158 | 208 | 89 | 138 | 1136 | 62 |
| P35269 | General Transcription | GTF2F1 | 35 | 587 | 192 | 68 | 64 | 685 | 224 | 846 | 416 | 466 | 210 | 309 |
| P13984 | | GTF2F2 | 61 | 636 | 249 | 134 | 229 | 964 | 741 | 811 | 236 | 837 | 510 | 472 |
| Q8N7H5 | RNA Pol II Associating Factor | PAF1 | 1901 | 886 | 1304 | 14 | 579 | 1478 | 1224 | 827 | | | 1657 | 33 |
| Q6P1J9 | | CDC73 | 1478 | 266 | 542 | 206 | 992 | 1355 | 484 | 1227 | 329 | 1635 | 1829 | 193 |
| Q6PD62 | | CTR9 | 106 | 1398 | 91 | 189 | | 29 | 159 | 79 | | 760 | 1509 | 324 |
| Q8WVC0 | | LEO1 | 1248 | 1271 | 792 | 272 | 624 | 1570 | 574 | 1601 | 187 | 1546 | 1277 | 404 |
| P18615 | | NELFE | 911 | 1482 | 1117 | 123 | 55 | 737 | 757 | 1317 | 487 | 1199 | 1891 | 882 |

Color Scale: Top ▮ 4%  ▮ 4–8%  ▮ 8–15%

**Figure 7.** mRNA processing factors in the 2nd-Z-rankings. The ranks of the mRNA processing factors in the 2nd-Z-rankings of the 12 NTRs are presented. The color scale is set by percentile rank as indicated. The 2nd-Z-rankings of the 12 NTRs include 275 proteins in total that are annotated with mRNA processing in GO. Of these, 69 were selected and are presented. For other factors and the 3rd-Z-rankings, see *Supplementary file 11A*.

## Allocation of RPs to the NTRs

RPs migrate into the nuclei for ribosome assembly, but the NTRs responsible for import have been determined for only a few of RPs (*Chook and Süel, 2011*). The 3rd-Z-4% cargoes include 15 RPs (*Supplementary files 1*, *5C,* and *11B*). To see the allocations of all the RPs to the NTRs, the ranks of

the RPs in the 2nd-Z-rankings are arranged in a table (*Figure 8*). Because the +NTR/Ctl values were obtained for most of the RPs in the three SILAC-Tp replicates, the second Z-scores are the median Z-scores in most cases, and they should fairly reflect the import efficiencies. Half of the RPs are included in the 2nd-Z-15% cargoes of one to five NTRs, and most of the RPs are ranked in the top 30% in the 2nd-Z-rankings of additional NTRs. Surprisingly, most RPs, especially the 60S subunit proteins, are ranked in the top 50% of most of the 2nd-Z-rankings, and few RPs are ranked lower. These findings imply that most of the RPs are allocated to multiple NTRs, but the import efficiencies vary depending on the NTR. Indeed, several RPs are reported cargoes of multiple NTRs (*Jäkel and Görlich, 1998*; *Jäkel et al., 2002*). Imp-7, -8, and -9 primarily import RPs, Imp-11 and Exp-4 secondarily import RPs, and all other NTRs also contribute to the import of RPs to some extent. Among the highly homologous NTR pairs, Trn-2 and Imp-8 import RPs more efficiently than Trn-1 and Imp-7, respectively, which indicates that RP import is one of the roles shared unequally by similar NTRs. This differentiation is clearer in the 3rd-Z-rankings (*Supplementary file 11B*).

## Allocation of transcription factors to the NTRs

Sequence-specific DNA-binding transcription factors (annotated with 'transcription factor activity, sequence specific DNA binding' in GO) play pivotal roles in many cellular processes, but they are not significantly enriched in the 3rd-Z-4% cargoes of any NTR (*Supplementary file 6B*). Transcription cofactors (annotated with 'transcription factor activity, protein binding'), which may engage in gene-specific transcription, are significantly enriched in the 3rd-Z-4% cargoes of only three NTRs (*Figure 6B*; *Supplementary file 6B*). Nonetheless, transcription factors (sequence-specific DNA binding) are enriched in the Imp-$\beta$ 2nd-Z-15% cargoes, and cofactors (protein binding) are enriched in the 2nd-Z-15% cargoes of 10 NTRs (*Supplementary file 6C*). Additionally, some transcription factors and cofactors are included in the 2nd-Z-15% cargoes, albeit not enriched. Thus, the 2nd-Z-15% cargoes of each NTR include 17 to 36 transcription factors or cofactors as listed at the bottom of *Supplementary file 11D*. We performed GO analyses (term type, BP) for these transcription factors and cofactors (*Supplementary file 11C and 11D*). The annotated terms may reflect both direct transcription regulation activities and indirect effects via transcription. The proteins annotated with histone modification are enriched in the Imp-$\beta$ and -13 cargoes, and the term may reflect their direct functions. The cargoes of several NTRs annotated with varying types of nuclear receptor signaling may act as cofactors in receptor-regulated transcription. In contrast, many of the transcription factors and cofactors identified as cargoes are annotated differently with various terms related to cell proliferation, development, rhythmic processes, or apoptosis and may act on these processes via transcriptional regulation. Thus, the NTRs import transcription factors and cofactors that work in distinct cellular processes.

## Characterizations of the cargoes of individual NTRs

The GO analyses elucidated the characteristics of the NTR-specific cargoes, but the terms are annotated to not only the central players but also many indirect participants in BPs. Here, we primarily discuss the roles of the notable 3rd-Z-4% cargoes of each NTR and supplement this information with references to the 2nd-Z-15% cargoes. To make our points clear, we classified the 3rd-Z-4% and 2nd-Z-15% cargoes by their characteristics and their allocations to each NTR are presented in *Supplementary file 5C and 5E*. We describe the features of the Imp-13 and Trn-SR cargoes first, because it includes the discussion on an export cargo or SR-domains. Biological functions linked to NTRs by the natures of their cargoes need to be verified by further experiments.

## Imp-13 cargoes

Several Imp-13 cargoes have previously been reported, and our SILAC-Tp clearly reproduced the reported import specificities. Nuclear transcription factor Y subunits $\beta$ (NFYB) and $\gamma$ (NFYC) have been reported to be Imp-13 specific cargoes, whereas subunit $\alpha$ (NFYA), which has a BIB-like sequence, has been reported to bind to multiple NTRs (*Kahle et al., 2005*). NFYB is ranked first in both the Imp-13 2nd- and 3rd-Z-rankings, and NFYC is a 2nd-Z-15% cargo (*Figure 3*; *Supplementary files 1* and *3*). Additionally, we identified NFYA as a cargo of multiple NTRs. Interestingly, a subunit of the general transcription factor TFIIA (GTF2A2) that interacts with NFYA (*Rolland et al., 2014*) is also a highly ranked Imp-13 3rd-Z-4% cargo. We could not identify the Imp-

| | | | Rank by 2nd Z-score | | | | | | | | | | | |
|---|---|---|---|---|---|---|---|---|---|---|---|---|---|---|
| | Accession | Gene Name | Imp-β | Trn-1 | Trn-2 | Trn-SR | Imp-4 | Imp-5 | Imp-7 | Imp-8 | Imp-9 | Imp-11 | Imp-13 | Exp-4 |
| 40S Subunit | P08865 | RPSA | 1112 | 1181 | 580 | 1143 | 442 | 599 | 568 | 436 | 520 | 514 | 805 | 640 |
| | P15880 | RPS2 | 1279 | 860 | 840 | 992 | 1095 | 859 | 633 | 488 | 481 | 565 | 740 | 594 |
| | P23396 | RPS3 | 1266 | 1058 | 645 | 1042 | 962 | 657 | 537 | 418 | 512 | 372 | 973 | 494 |
| | P61247 | RPS3A | 1127 | 1047 | 859 | 1096 | 672 | 745 | 548 | 413 | 351 | 455 | 827 | 475 |
| | P62701 | RPS4X | 1197 | 911 | 835 | 982 | 925 | 624 | 474 | 394 | 422 | 558 | 800 | 564 |
| | P46782 | RPS5 | 1094 | 967 | 678 | 1095 | 725 | 769 | 478 | 252 | 338 | 502 | 798 | 458 |
| | P62753 | RPS6 | 673 | 902 | 456 | 1173 | 590 | 829 | 507 | 502 | 395 | 522 | 791 | 369 |
| | P62081 | RPS7 | 1103 | 1031 | 907 | 1127 | 889 | 655 | 511 | 431 | 81 | 588 | 747 | 560 |
| | P62241 | RPS8 | 1161 | 1083 | 871 | 1020 | 927 | 691 | 530 | 464 | 424 | 503 | 691 | 489 |
| | P46781 | RPS9 | 1147 | 1016 | 699 | 963 | 704 | 626 | 392 | 404 | 366 | 523 | 880 | 535 |
| | P46783 | RPS10 | 924 | 1105 | 521 | 675 | 420 | 693 | 420 | 335 | 328 | 253 | 662 | 490 |
| | P62280 | RPS11 | 1170 | 1086 | 743 | 735 | 842 | 555 | 490 | 390 | 369 | 587 | 751 | 448 |
| | P25398 | RPS12 | 1110 | 1166 | 727 | 1033 | 1029 | 752 | 418 | 326 | 411 | 623 | 619 | 482 |
| | P62277 | RPS13 | 1087 | 830 | 888 | 1140 | 812 | 693 | 516 | 429 | 420 | 521 | 916 | 456 |
| | P62263 | RPS14 | 1089 | 1203 | 936 | 1248 | 1075 | 881 | 319 | 382 | 418 | 657 | 689 | 623 |
| | P62841 | RPS15 | 1373 | 908 | 691 | 997 | 1002 | 487 | 510 | 449 | 583 | 953 | 936 | 612 |
| | P62244 | RPS15A | 1037 | 1263 | 862 | 1014 | 878 | 651 | 503 | 432 | 436 | 329 | 1020 | 546 |
| | P62249 | RPS16 | 1045 | 1041 | 763 | 966 | 957 | 588 | 517 | 369 | 371 | 583 | 821 | 653 |
| | P0CW22 | RPS17L | 1210 | 1158 | 709 | 802 | 784 | 512 | 520 | 447 | 361 | 403 | 1018 | 441 |
| | P62269 | RPS18 | 981 | 1089 | 674 | 869 | 911 | 583 | 467 | 391 | 333 | 729 | 977 | 521 |
| | P39019 | RPS19 | 1122 | 1099 | 937 | 993 | 888 | 594 | 461 | 384 | 360 | 285 | 851 | 506 |
| | P60866 | RPS20 | 1123 | 968 | 858 | 736 | 995 | 472 | 424 | 226 | 365 | 293 | 701 | 544 |
| | P63220 | RPS21 | 1215 | 988 | 954 | 1036 | 718 | 653 | 704 | 340 | 444 | 549 | 502 | 614 |
| | P62266 | RPS23 | 1219 | 957 | 776 | 785 | 756 | 475 | 305 | 297 | 348 | 516 | 634 | 345 |
| | P62847 | RPS24 | 1017 | 1056 | 899 | 1073 | 890 | 652 | 496 | 409 | 290 | 484 | 779 | 486 |
| | P62851 | RPS25 | 1111 | 1211 | 1052 | 925 | 563 | 812 | 450 | 313 | 525 | 428 | 1066 | 410 |
| | P62854 | RPS26 | 975 | 1400 | 1428 | 885 | 1000 | 916 | 428 | 250 | | | 371 | 1242 |
| | P42677 | RPS27 | 1051 | 997 | 684 | 811 | 616 | 595 | 405 | 428 | 610 | 446 | 631 |
| | P62979 | RPS27A | | 710 | 451 | 1108 | 966 | 668 | 828 | 350 | 483 | 265 | 489 | 577 |
| | P62857 | RPS28 | 828 | 1106 | 802 | 809 | 917 | 596 | 557 | 367 | 401 | 598 | 726 | 391 |
| | P62273 | RPS29 | 1256 | 1064 | 1068 | 873 | 916 | 722 | 519 | 318 | 536 | 1056 | 702 | 365 |
| | P62861 | FAU(RPS30) | 1265 | 981 | 675 | 1487 | 908 | 990 | 528 | 320 | 403 | 150 | 695 | 407 |
| 60S Subunit | P05388 | RPLP0 | 1152 | 1066 | 662 | 1028 | 1146 | 711 | 690 | 821 | 564 | 731 | 1069 | 520 |
| | P05386 | RPLP1 | 526 | 732 | 650 | 1030 | 373 | 913 | 817 | 678 | 594 | 416 | 323 | 503 |
| | P05387 | RPLP2 | 908 | 733 | 476 | 766 | 929 | 666 | 575 | 481 | 686 | 260 | 302 | 401 |
| | P39023 | RPL3 | 830 | 665 | 524 | 1047 | 455 | 800 | 332 | 160 | 184 | 366 | 824 | 161 |
| | P36578 | RPL4 | 832 | 563 | 410 | 1247 | 633 | 891 | 317 | 302 | 207 | 459 | 758 | 246 |
| | P46777 | RPL5 | 873 | 638 | 566 | 841 | 853 | 1053 | 830 | 843 | 624 | 612 | 889 | 778 |
| | Q02878 | RPL6 | 701 | 562 | 478 | 1199 | 601 | 815 | 389 | 331 | 306 | 689 | 578 | 340 |
| | P18124 | RPL7 | 816 | 618 | 457 | 1057 | 739 | 789 | 354 | 227 | 137 | 157 | 716 | 305 |
| | Q6DKI1 | RPL7L1 | 1842 | | 1483 | 41 | | | | | | | | |
| | P62424 | RPL7A | 929 | 775 | 582 | 1230 | 404 | 639 | 353 | 237 | 262 | 448 | 743 | 180 |
| | P62917 | RPL8 | 728 | 731 | 481 | 1266 | 504 | 773 | 333 | 246 | 153 | 234 | 757 | 303 |
| | P32969 | RPL9 | 1153 | 662 | 347 | 1334 | 368 | 806 | 318 | 195 | 265 | 595 | 870 | 291 |
| | P27635 | RPL10 | 960 | 789 | 668 | 1177 | 974 | 809 | 399 | 311 | 263 | 489 | 712 | 420 |
| | P62906 | RPL10A | 808 | 621 | 360 | 1227 | 461 | 787 | 313 | 241 | 147 | 471 | 769 | 251 |
| | P62913 | RPL11 | 793 | 712 | 491 | 1326 | 1123 | 710 | 497 | 40 | 201 | 375 | 401 | 377 |
| | P30050 | RPL12 | 863 | 763 | 614 | 1208 | 305 | 814 | 338 | 321 | 254 | 314 | 845 | 286 |
| | P26373 | RPL13 | 836 | 530 | 492 | 1197 | 275 | 775 | 244 | 185 | 189 | 323 | 759 | 247 |
| | P40429 | RPL13A | 737 | 940 | 375 | 1203 | 943 | 864 | 239 | 362 | 228 | 305 | 656 | 398 |
| | P50914 | RPL14 | 814 | 1136 | 561 | 1204 | 290 | 749 | 299 | 154 | 240 | 573 | 897 | 362 |
| | P61313 | RPL15 | 801 | 771 | 432 | 1246 | 556 | 820 | 388 | 244 | 205 | 360 | 841 | 353 |
| | P18621 | RPL17 | 799 | 1004 | 706 | 1170 | 610 | 350 | 308 | 315 | 209 | 518 | 597 | 166 |
| | Q07020 | RPL18 | 758 | 760 | 538 | 1229 | 369 | 971 | 248 | 368 | 269 | 422 | 754 | 314 |
| | Q02543 | RPL18A | 847 | 484 | 551 | 1400 | 1285 | 830 | 437 | 328 | 232 | 304 | 605 | 289 |
| | P84098 | RPL19 | 1109 | 1118 | 651 | 1243 | 406 | 695 | 502 | 301 | 384 | 176 | 975 | 367 |
| | P46778 | RPL21 | 818 | 883 | 507 | 1161 | 647 | 836 | 382 | 319 | 247 | 607 | 509 | 139 |
| | P35268 | RPL22 | 729 | 714 | 450 | 474 | 515 | 306 | 52 | 42 | 347 | 1021 | 398 | 376 |
| | P62829 | RPL23 | 817 | 703 | 535 | 897 | 668 | 842 | 536 | 448 | 323 | 718 | 836 | 514 |
| | P62750 | RPL23A | 1092 | 625 | 417 | 1146 | 844 | 807 | 162 | 222 | 165 | 458 | 753 | 358 |
| | P83731 | RPL24 | 1035 | 963 | 718 | 1141 | 643 | 743 | 421 | 307 | 299 | 139 | 1001 | 381 |
| | P61254 | RPL26 | 650 | 821 | 523 | 1160 | 664 | 636 | 192 | 229 | 320 | 547 | 649 | 279 |
| | Q9UNX3 | RPL26L1 | | 1337 | | 1693 | 454 | 567 | | 122 | | | | |
| | P61353 | RPL27 | 705 | 654 | 550 | 1158 | 614 | 925 | 301 | 192 | 326 | 430 | 941 | 216 |
| | P46776 | RPL27A | 792 | 615 | 477 | 1151 | 295 | 901 | 306 | 274 | 211 | 105 | 599 | 182 |
| | P46779 | RPL28 | 666 | 566 | 412 | 1325 | 203 | 603 | 409 | 346 | 342 | 806 | 555 | 198 |
| | P47914 | RPL29 | 982 | 682 | 598 | 1219 | 660 | 660 | 303 | 235 | 346 | 615 | 847 | 294 |
| | P62888 | RPL30 | 621 | 659 | 570 | 1252 | 760 | 570 | 273 | 136 | 173 | 370 | 808 | 220 |
| | P62899 | RPL31 | 543 | 1026 | 590 | 1142 | 1104 | 811 | 567 | 286 | 376 | 379 | 568 | 336 |
| | P62910 | RPL32 | 885 | 583 | 416 | 1324 | 301 | 926 | 381 | 199 | 252 | 1055 | 910 | 242 |
| | P49207 | RPL34 | 1135 | 1410 | 350 | 1285 | 597 | 546 | 278 | 123 | 305 | 223 | 658 | 260 |
| | P42766 | RPL35 | 731 | 508 | 356 | 1058 | 518 | 593 | 175 | 168 | 104 | 67 | 454 | 128 |
| | P18077 | RPL35A | 951 | 614 | 772 | 1366 | 814 | 840 | 385 | 219 | 193 | 255 | 813 | 102 |
| | Q9Y3U8 | RPL36 | 1157 | 845 | 317 | 1231 | 125 | 961 | 414 | 395 | 128 | 239 | 823 | 230 |
| | P83881 | RPL36A | 1238 | | | | | | | | 382 | 295 | 1511 | |
| | Q969Q0 | RPL36AL | | 1347 | 522 | 937 | 211 | 609 | 601 | 489 | | | | 168 |
| | P61927 | RPL37 | 309 | | | | 1012 | | 458 | | | 1638 | 130 | 474 |
| | P61513 | RPL37A | 1142 | 827 | 526 | 940 | 613 | 707 | 551 | 238 | 237 | 562 | 1075 | 162 |
| | P63173 | RPL38 | 780 | 1052 | 409 | 1046 | 387 | 623 | 280 | 210 | 266 | 555 | 586 | 264 |

Color Scale: Top 15% | 15-30% | 30-50% | >50%

**Figure 8.** Ribosomal proteins in the 2nd-Z-rankings. The ranks of the ribosomal proteins in the 2nd-Z-rankings of the 12 NTRs are presented. The color scale is set by the percentile rank as indicated. For the 3rd-Z-rankings, see *Supplementary file 11B*.

13 reported cargo glucocorticoid receptor (*Tao et al., 2006*) in our MS, but proteins that may interact with nuclear receptors, e.g., thyroid hormone receptor-associated protein 3 (THRAP3) (*Ito et al., 1999*), RNA-binding protein 14 (RBM14) (*Iwasaki et al., 2001*), and transcription activator BRG1 (SMARCA4) (*Dai et al., 2008*), were identified as Imp-13 2nd-Z-15% cargoes. THRAP3 interacts with EJC (*Lee et al., 2010*), whose subunits, RNA-binding protein RBM8A and mago nashi homolog MAGOH, are well-characterized Imp-13 cargoes (*Mingot et al., 2001*). RBM8A and MAGOH are highly ranked Imp-13 3rd-Z-4% cargoes, which were not identified as cargoes of the other NTRs with the exception of Trn-1. However, another EJC subunit, that is, translation initiation factor 4A-III (EIF4A3) (*Shibuya et al., 2004*), was identified as an Imp-11 3rd-Z-4% cargo. Thus, the EJC subunits are imported through different pathways. A well-characterized Imp-13 cargo, SUMO-conjugating enzyme UBC9 (UBE2I) (*Mingot et al., 2001*), was identified as a 3rd-Z-4% cargo, and SUMO2 and SUMO3 were also identified as 3rd-Z-4% cargoes. Components of the chromatin accessibility complex CHRAC15 (CHRAC1) and DNA polymerase ε subunit 3 (POLE3) are also Imp-13 reported cargoes (*Walker et al., 2009*), and they are highly ranked 3rd-Z-4% cargoes. In the GO analysis, Imp-13 was linked to chromatin modification by the 2nd-Z-15% cargoes (*Supplementary file 6A*). Nucleolar complex protein 2 homolog (NOC2L) is ranked 10th in both the second and third Z-scores, and lysine-specific demethylase 2A (KDM2A) is ranked second in the second Z-score. The Imp-13 2nd-Z-15% cargoes include many actin-related proteins that are involved in chromatin remodeling and transcription (*Oma and Harata, 2011*; *Yoo et al., 2007*).

Surprisingly, a reported Imp-13 export cargo eIF1A (EIF1AX; *Mingot et al., 2001*) was identified as a 2nd-Z-15% cargo (ranked 84th and 164th by the second and third Z-score, respectively; *Supplementary files 1* and *3*). If a protein endogenous to the permeabilized cell nuclei is exported preferentially in the +NTR in vitro transport reaction, the $(L/H_{+NTR})/(L/H_{Ctl})$ value will be raised and the protein will be ranked high. However, it cannot be generalized because we have only one example. Most of the highly ranked Imp-13 cargoes must be import cargoes, because in all the bead halo assays where the cargoes bound to Imp-13 RanGTP inhibited the binding (*Supplementary file 2*).

## Trn-SR cargoes

The reported Trn-SR cargoes include SR-rich splicing factors (SFs) that coordinate transcription elongation, mRNA splicing, and mRNA export (*Zhong et al., 2009*). Here, we found that proteins engaging in these processes are also Trn-SR cargoes. The Trn-SR 3rd-Z-4% cargoes include the RNA polymerase (RNAP) II elongation factors NELFE and PAF1 (a subunit of the Paf1 complex, PAF1C), DDX and DHX family RNA helicases, and the THO complex subunit THOC1 as well as SR-rich SFs. The 2nd-Z-15% cargoes additionally include PAF1C subunits CTR9, CDC73, and LEO, FACT complex subunits SSRP1 and SPT16, additional DDX and DHX family helicases, and THOC6 and THOC3 (*Supplementary file 5C and 5E*). Trn-SR bound to NELFE, CDC73, DDX5, and DDX27 in the bead halo assays (*Figure 3*, *Supplementary files 1*, *2,* and *3*). Peptidyl-prolyl cis-trans isomerases, which are contained in human spliceosomes (*Wahl et al., 2009*), were also identified as 3rd-Z-4% and 2nd-Z-15% cargoes. DnaJ homologs were also identified as 3rd-Z-4% and 2nd-Z-15% cargoes, although the spliceosome component DNAJC8 (*Zhou et al., 2002*) was not. The 3rd-Z-4% cargoes also include proteins related to nuclear division or chromosome segregation, the Ser/Thr protein kinase PLK1, dual specificity protein kinase TTK, G2/M-specific cyclin-B1 (CCNB1), cyclin-dependent kinase (CDK) 2, protein FAM83D, and dynein 1 light intermediate chain 1 (DYNC1LI1) in addition to proteins related to histone acetylation or deacetylation including histone deacetylase complex subunit SAP18 and SAGA-associated factor 29 homolog CCDC101. Indeed, in the bead halo assays, Trn-SR bound to SAP18 and CCDC101 (*Figure 3*; *Supplementary files 1*, *2,* and *3*). Additionally, the 2nd-Z-15% cargoes include many proteins that are related to nucleosome or chromatin regulation. Thus, the Trn-SR cargoes are involved in chromosome regulation in addition to the coordination of transcription elongation, mRNA splicing, and mRNA export.

Surprisingly, SR-rich SFs, which have been assumed to be Trn-SR-specific cargoes, were also identified as cargoes of other NTRs (*Figure 7*). To determine the allocation of the other SR-domain proteins to the NTRs, we here analyzed the distribution of SRSRSR hexa-peptide sequences in the 3rd-Z-4% cargoes (*Supplementary file 11E*). Imp-5, -7, -8, and Exp-4 as well as Trn-SR may be the specific NTRs for proteins with the hexa-peptide, most of which are nuclear proteins. The hexa-peptide-containing proteins other than the SR-rich SFs are primarily included in the Imp-5 and Exp-4 cargoes.

## Imp-β cargoes

The Imp-$\beta$ cargoes play roles in DNA synthesis and repair and chromatin regulation. The Imp-$\beta$ 3rd-Z-4% cargoes include DNA polymerase δ subunits (POLD2 and 3) and mismatch repair endonuclease PMS2 (*Supplementary file 5C*). Additionally, the Imp-$\beta$ 2nd-Z-15% cargoes include PCNA-associated factor KIAA0101 and DNA-(apurinic or apyrimidinic site) lyase (APEX1) (*Supplementary file 5E*). These proteins act in DNA synthesis or repair. The notable 3rd-Z-4% cargoes related to chromatin regulation include high-mobility group (HMG) proteins, histone acetyltransferase complex NuA4 subunit MRGBP, SWI/SNF-related regulator of chromatin SMARCE1, Spindlin-1 (SPIN1), chromodomain-helicase CHD8, and lymphoid-specific helicase HELLS. Additionally, the 2nd-Z-15% cargoes include the NuA4 subunit MORF4L2, SWI/SNF complex subunit SMARCC2, chromatin assembly factor 1 subunit CHAF1B, polycomb protein EED, and sister chromatid cohesion protein PDS5B. Chromatin remodeling by some of these factors is closely related to transcription. The 3rd-Z-4% cargoes include general transcription factor TFIIF (GTF2F1 and 2), TFIIH subunit MAT1, and TBP-associating factor TAF15, and the 2nd-Z-15% cargoes include TFIIH subunit cyclin-H (CCNH) and mediator complex subunit MED15. The sequence-specific transcription factors and cofactors are described above. mRNA capping factors are Imp-$\beta$ cargoes as described. Thus, many Imp-$\beta$ cargoes are related to the initial stage of gene expression.

## Trn-1 and -2 cargoes

The transcription factor ATF1 was ranked first in both the 2nd- and 3rd-Z-rankings of the Trn-1 and -2 but was ranked low for the other NTRs (*Figure 3*; *Supplementary files 3* and *5*). As described, many of the cargoes that ranked higher in the Trn-1 and -2 2nd- and 3rd-Z-rankings (e.g. hnRNPs) are shared by Trn-1 and -2, but RPs are included only in the Trn-2 3rd-Z-4% cargoes. Additional divergences can be observed between their 2nd-Z-15% cargoes. As expected, their cargoes include many mRNA processing factors, but among them snRNPs are preferentially included in the Trn-2 2nd-Z-15% cargoes (*Figure 7*). Actin and actin-related proteins (ARPs), which play roles in chromatin remodeling and transcription (*Visa and Percipalle, 2010*; *Yoo et al., 2007*), proteins related to nuclear division, and tRNA ligases are preferentially Trn-1 cargoes, whereas proteins related to DNA repair and HMG proteins are preferentially Trn-2 cargoes (*Supplementary file 5E*).

## Imp-4 cargoes

In the GO analysis, Imp-4 was linked to DNA metabolic processes, chromosome organization, and related terms (*Figures 5* and *6*). Consistently, replication factor C subunit 5 (RFC5) and HMG proteins are Imp-4 3rd-Z-4% cargoes, and the 2nd-Z-15% cargoes include DNA polymerase α subunit POLA1, DNA ligase I (LIG1), DNA topoisomerase I (TOP1), SWI/SNF complex subunit SMARCC2, the SWI/SNF-related chromatin regulator SMARCA5, nucleosome remodeling factor subunit BPTF, and FACT complex subunit SPT16 (*Supplementary file 5C and 5E*). The participation of the Imp-4 cargoes in chromatin organization is supported by a report that Imp-4 binds to the histone chaperon complex (*Tagami et al., 2004*), although the subunits were not identified in our MS. Imp-4 was also linked to cell cycle in the GO analysis. The Imp-4 3rd-Z-4% cargoes include the regulator of chromosome condensation RCC1 and the Ser/Thr protein kinase PLK1, and the 2nd-Z-15% cargoes include the sister chromatid cohesion protein PDS5 homolog PDS5B. Imp-4 was also linked to programed cell death or apoptosis in the GO analysis. The representative related 3rd-Z-4% cargoes are the death-promoting transcriptional repressor BCLAF1 and the tumor suppressor ARF (CDKN2A), and the 2nd-Z-15% cargoes are ribosomal L1 domain-containing protein 1 (RSL1D1) and apoptosis-inducing factor 1 (AIFM1).

## Imp-5 cargoes

Few characteristics are unique to the Imp-5 3rd-Z-4% cargo cohort. However, this cohort includes proteins related to ribosome biogenesis, such as rRNA 2'-O-methyltransferase fibrillarin (FBL), H/ACA ribonucleoprotein complex subunit 1 (GAR1), and the ribosome biogenesis protein BOP1. This group also includes proteins related to nucleosome or chromatin organization, including spindlin-1 (SPIN1), protein DEK, the methyl-CpG-binding domain protein MBD2, and the paired amphipathic helix protein SIN3B (*Supplementary file 5C*). SR-rich SFs are also included as described. The Imp-5 2nd-Z-15% cargoes include many ARPs, proteins related to spindle organization or microtubule-

based processes, and several CDKs (*Supplementary file 5E*). Thus, a portion of the Imp-5 cargoes may be involved in cytokinesis. A number of translation initiation factors (eIFs) and elongation factors, many of which are annotated with nuclear localization (*Supplementary file 1*), are also among the Imp-5 2nd-Z-15% cargoes.

## Imp-7 and -8 cargoes

The cognate NTRs Imp-7 and -8 share many 3rd-Z-4% and 2nd-Z-15% cargoes (*Figure 4*; *Supplementary file 5A*). The major cargoes of these NTRs are a range of mRNA SFs, but by the third Z-scores, snRNPs were identified only as Imp-7 and not Im-8 cargoes (*Supplementary file 5C*). Additional divergences can be observed between the Imp-7 and -8 cargoes (*Supplementary file 5C and 5E*). HMG proteins were identified only as Imp-7 3rd-Z-4% and 2nd-Z-15% cargoes, whereas more RPs were identified as Imp-8 cargoes. Proteins related to cell cycle regulation, the mitotic checkpoint protein BUB3, cell division cycle 5-like protein (CDC5L), and CDK12, are included in the Imp-7 3rd-Z-4% cargoes, and the Ser/Thr protein kinase PLK1 is a 2nd-Z-15% cargo, but these proteins are not Imp-8 cargoes. Many eIFs are Imp-8 but not Imp-7 2nd-Z-15% cargoes.

## Imp-9 cargoes

The Imp-9 cargoes include many RPs and mRNA SFs. Proteins that are important for DNA packaging or nucleosome organization were also identified as Imp-9 cargoes (*Supplementary file 5C and 5E*). Histone H2A.Z, which is located in specific regions on chromosome (*Weber and Henikoff, 2014*), is ranked first and third in the third and second Z-scores, respectively. Additionally, the linker histone H1 (H1F0), histone-lysine N-methyltransferase 2A (KMT2A), and the SPT16 and SSRP1 subunits of the FACT complex, which regulates histone H2A.Z (*Jeronimo et al., 2015*), were also identified as 3rd-Z-4% cargoes. Among the Imp-9 2nd-Z-15% cargoes, other histones, DNA topoisomerase I (TOP1) and IIα (TOP2A), HMG proteins, SWI/SNF-related matrix-associated actin-dependent regulator of chromatin subfamily E member 1 (SMARCE1), and scaffold attachment factor B1 (SAFB) are included.

## Imp-11 cargoes

Imp-11 was linked to developmental processes in the GO analysis (*Figure 5*; *Supplementary file 6A*), and few proteins with typical nuclear functions, such as DNA replication, nucleosome organization, and transcription, were found among the Imp-11 3rd-Z-4% cargoes (*Supplementary file 5C*). The Imp-11 2nd-Z-15% cargoes include several proteins related to nuclear division, such as Pogo transposable element with ZNF domain (POGZ), α-endosulfine (ENSA), CDK regulatory subunit 2 (CKS2), and the Ser/Thr protein kinase NEK7 (*Supplementary file 5E*). Many ARPs, tubulins and their related factors, tRNA ligases, and mRNA SFs are also in the Imp-11 2nd-Z-15% cargoes.

## Exp-4 cargoes

The subunits of RNAP II elongation factors and mRNA processing factors are the representative Exp-4 cargoes, although they are also cargoes of several other NTRs (*Supplementary file 5C and 5E*). PAF1C subunit parafibromin (CDC73) is an Exp-4 3rd-Z-4% cargo, and other PAF1C subunits, that is, PAF1 and RTF1, FACT complex subunits, i.e., SSRP1 and SPT16 (SUPT16H), and elongation complex protein 2 (ELP2) are 2nd-Z-15% cargoes. A variety of mRNA processing factors, including 3'-end processing factors and THO complex subunits, are also Exp-4 3rd-Z-4% and 2nd-Z-15% cargoes. Thus, the factors that act in processes from transcription elongation to mRNA export are included in the Exp-4 cargoes. As discussed for another bi-directional NTR Imp-13, the possibility cannot be denied that the identified Exp-4 candidate cargoes include export cargoes.

## Seemingly non-nuclear proteins

A number of nucleoporins (NUPs), which are the components of the NPC, were identified as cargoes. Increasing evidence demonstrates that the import of NUPs through NPCs is important for gene expression (*Burns and Wente, 2014*). Moreover, many mitochondrial proteins are highly ranked. These proteins preferentially localize to the mitochondria due to chaperon-regulated or cotranslational mechanisms in vivo and might interact with NTRs in the in vitro transport system. The transport system contains cytosolic extract and unlabeled (light) mitochondrial proteins in it could be

imported if they interact with NTRs. The $(L/H_{+NTR})/(L/H_{Ctl})$ values of them can be calculated, because LC-MS/MS can quantify low levels of labeled (heavy) proteins whether they are endogenous to the recipient nuclei or residual after washing. Thus, mitochondrial proteins with high $(L/H_{+NTR})/(L/H_{Ctl})$ values are imported proteins even if the import is fortuitous. Nuclear localization is annotated to many mitochondrial proteins (*Supplementary file 1*), and actual nuclear localization is possible as in the cases of AIFM1 and ATFS-1 (*Nargund et al., 2012*; *Susin et al., 1999*). As was the case with the high-throughput cargo identification of the export receptor Exp-1 (CRM1) (*Kırlı et al., 2015*), our method identified other seemingly cytoplasmic proteins as cargoes. We did not detect direct binding between the NTRs and some of these cytoplasmic proteins, for example, Ras-related Rab family proteins and S100 proteins, in the bead halo assays (*Supplementary file 2*), but nuclear import by indirect binding is still possible.

## Additional remarks

Here, we have presented the first complete picture of nuclear import via the 12 importin pathways. The 12 pathways must serve distinct roles because the NTRs are linked to different cellular processes by their cargoes. However, the cargoes are intricately allocated to the NTRs, and each NTR is linked to multiple cellular processes. The biological functions of NTRs designated in this work should be further clarified in future experiments.

We used HeLa nuclear extract as the cargo source, but it might not reconstitute all NTR–cargo interactions precisely because proteins in the nuclear extract might have different modifications or binding partners from those in cytoplasm where NTRs bind to cargoes in vivo. Some reported cargoes were ranked lower in the 2nd- and 3rd-Z-ranking, and it might be attributable to these differences of protein states. Alternatively, the transport capacity of our in vitro transport system might not be enough to identify all the cargoes, especially those with low transport efficiency. To reach a definitive conclusion, experiments in vivo might be needed.

We could not find any novel motifs that may serve as NTR-binding sites on the identified cargoes using the ungapped motif search method of MEME (*Bailey and Elkan, 1994*). A more extensive search for such motifs and higher order structures using alternative methods is currently underway.

# Materials and methods

## SILAC-Tp

SILAC-Tp has previously been described in detail (*Kimura et al., 2014*), but we provide a brief description here. HeLa-S3 cytosolic and nuclear extracts were depleted of Imp-$\beta$ family NTRs with phenyl-Sepharose (GE healthcare), and the nuclear extract was subsequently depleted of RCC1 with a Ran-affinity method and concentrated. The extracts were dialyzed against transport buffer (TB, 20 mM HEPES–KOH (pH 7.3), 110 mM KOAc, 2 mM MgOAc, 5 mM NaOAc, 0.5 mM EGTA, 2 mM DTT, and 1 µg/mL each of aprotinin, pepstatin A, and leupeptin). Adherent HeLa-S3 cells were labeled with u-$^{13}C_6$ Lys and u-$^{13}C_6$ Arg by SILAC (*Ong et al., 2002*) and seeded onto a glass plate. After rinsing in ice cold TB, the cells were permeabilized with 40 µg/mL digitonin in TB for 5 min on ice and then rinsed again. The permeabilized cells were pretreated with 4 µM RanGDP and an ATP regeneration system in TB for 20 min at 30°C to remove the residual Imp-$\beta$ family NTRs and then rinsed. The cells were incubated in transport mixture (50% cytosolic extract, 10% nuclear extract, 1 µM p10/NTF2, and ATP regeneration system in TB) with (+NTR) or without (Ctl) 0.3–0.7 µM of one NTR for 20 min at 30°C for the import reaction. (The NTR concentrations were optimized using the recombinant cargoes presented in *Figure 1—figure supplement 1C*.) After rinsing, the cells were incubated in extract mixture (50% cytosolic extract and ATP regeneration system in TB) for 20 min at 30°C and rinsed with NaCl-TB (TB containing 110 mM NaCl instead of KOAc) to remove the nonspecifically binding proteins. To extract the proteins, the cells were suspended in nuclear buffer (20 mM Tris–HCl, pH 8.0, 420 mM NaCl, 1.5 mM MgCl$_2$, 0.2 mM EDTA, 2 mM DTT, and 1 µg/mL each of aprotinin, pepstatin A, and leupeptin), sonicated, and centrifuged.

Actually, the transport reactions for two NTRs were simultaneously performed with one control reaction and triplicated. The simultaneously processed NTRs were Imp-$\beta$ and Imp-13, Trn-1 and -2, Imp-7 and -8, Imp-9 and -11, and Imp-5 and Trn-SR, and the reactions for Imp-4 and Exp-4 were performed individually with controls.

## Peptide analysis by LC-MS/MS

After the in vitro transport reaction, 25 μg each of the extracted proteins was concentrated by acetone precipitation, reduced with DTT, and alkylated with iodoacetamide. The proteins were digested with trypsin and Lys-C endopeptidase (enzyme/substrate ≈ 1/50) for 16 hr at 37°C. The peptides were evaporated to dryness, dissolved in Solvent-1 (0.1% TFA and 15% $CH_3CN$), and fractionated on Empore Cation Exchange-SR (3M, Maplewood, Minnesota). For the fractionation, the support was stacked manually inside the tapered end of a micropipette tip, the tip was fixed into the punched lid of a microtube, and the liquids were run by centrifugation (*Wiśniewski et al., 2009*). The resin was sequentially washed by ethanol and Solvent-1 containing 500 mM ammonium acetate and equilibrated with Solvent-1, and the peptides were then applied. After washing in Solvent-1, the peptides were eluted stepwise by Solvent-1 containing 125, 250, and 500 mM ammonium acetate and Solvent-2 (5% $NH_4OH$, 30% methanol, and 15% $CH_3CN$). The eluates were evaporated to dryness, and the peptides were dissolved in 0.1% TFA and 2% $CH_3CN$.

The peptides were applied to a liquid chromatograph (LC) (EASY-nLC 1000; Thermo Fisher Scientific, Waltham, Massachusetts) coupled to a Q Exactive hybrid quadrupole-Orbitrap mass spectrometer (Thermo Fisher Scientific) with a nanospray ion source in positive mode. The LC was performed on a NANO-HPLC capillary column C18 (75 μm x 150 mm, 3 μm particle size, Nikkyo Technos, Tokyo) at 45°C. The peptides were eluted with a 100-min 0–30% $CH_3CN$ gradient and a subsequent 20-min 30–65% gradient in the presence of 0.1% formic acid at a flow rate of 300 nL/min. The Q Exactive-MS was operated in the top-10 data-dependent scan mode. The parameters for the Q Exactive operation were as follows: spray voltage, 2.3 kV; capillary temperature, 275°C; mass range (m/z), 350–1800; and normalized collision energy, 28%. The raw data were acquired with Xcalibur (RRID:SCR_014593; ver. 2.2 SP1).

## Protein identification and quantitation

The MS and MS/MS data were searched against the Swiss-Prot database (2014_07–2016_01) using Proteome Discoverer (RRID:SCR_014477; ver. 1.4, Thermo Fisher Scientific) with the MASCOT search engine software (RRID:SCR_014322; ver. 2.4.1, Matrix Science, London). The search parameters were as follows: taxonomy, *Homo sapiens*; enzyme, trypsin; static modifications, carbamidomethyl (Cys); dynamic modifications, oxidation (Met); precursor mass tolerance, ±6 ppm; fragment mass tolerance, ±20 mDa; maximum missed cleavages, 1; and quantitation, SILAC (R6, K6). The proteins were considered identified when their false discovery rates were less than 5%. The SILAC L/H ratios were also calculated by Proteome Discoverer (ver. 1.4) with the default setting: show the raw quan values, false; minimum quan value threshold, 0; replace missing quan values with minimum intensity, false; use single-peak quan channels, false; apply quan value corrections, true; reject all quan values if not all quan channels are present, false; fold change threshold for up-/down-regulation, 1.5; maximum allowed fold change, 100; use ratios above maximum allowed fold change for quantification, false; percent co-isolation excluding peptides from quantification, 100; protein quantification, use only unique peptides; experimental bias, none. Proteins with L/H count ≥1 were included in further analysis. The L/H counts are shown in *Supplementary file 1*. To access the mass spectrometry data, see below.

From the SILAC quantitation values of the control and +NTR reactions, the +NTR/Ctl = $(L/H_{+NTR})/(L/H_{Ctl})$ ratio of each protein was calculated, and the Z-score of the $log_2$(+NTR/Ctl) of each protein was calculated within each replicate.

$$Z - score = (X - \mu)/\sigma$$

where $X$ is $log_2(+NTR/Ctl) = log_2[(L/H_{+NTR})/(L/H_{Ctl})]$ of each protein, $\mu$ is the mean of $X$, and $\sigma$ is the standard deviation of $X$.

## Reported cargo rate and recall

To calculate reported cargo rate (a lower bound on precision) and recall (sensitivity), we used the 27 and 25 reported cargoes of Trn-1 as the positive examples of the 2nd- and 3rd-Z-rankings, respectively. We do not have explicit labeling of negative examples. Most likely some portion of the proteins not reported as cargoes are genuine cargoes, but it is difficult to estimate that portion. Therefore, as a rough guide we tallied statistics under two simple assumptions: (i) that all proteins

not reported as cargoes should be treated as negative examples and (ii) that in the proteins not reported as cargoes, proteins annotated in Uniprot (RRID:SCR_002380) as having non-nuclear sub-cellular localization should be treated as negative examples and the other proteins excluded from the analysis (treated as neither positive nor negative). The first definition yielded 1622 and 1210 negative examples in the 2nd- and 3rd-Z-ranking, respectively, and the second definition 259 and 178 in the 2nd- and 3rd-Z-ranking, respectively. Since the first definition is maximally pessimistic, it allows estimation of an upper bound on the rate of false positives, while the second definition is more optimistic.

$$Reported\,cargo\,rate(i) = p(i)/[p(i)+n(i)]$$

$$Recall(i) = p(i)/P$$

where $p(i)$ denotes the number of previously reported cargoes (a lower bound on the number of positive examples) and $n(i)$ denotes the number of negative examples in the top i%; while $P$ denotes the total number of previously reported cargoes.

## Gene ontology analysis

GO (RRID:SCR_002811) analyses were performed using g:Profiler (RRID:SCR_006809; r1488-1536_e83_eg30) (*Reimand et al., 2016*). The search parameters were the following: organism, *Homo sapiens*; significance threshold, g:SCS; statistical domain size, all known genes; GO version, GO direct 2015-12-09 to 2016-01-21, releases/2015-12-08.

## Phylogenetic analysis of the 12 Imp-β family NTRs

The phylogeny was inferred by maximum likelihood using RAxML (RRID:SCR_006086; ver. 8.1.17) (*Stamatakis, 2006*) with 1000 bootstrap replicates and the LG model with gamma-distributed rate variation. The amino acid sequences were aligned using Clustal Omega (RRID:SCR_001591; ver. 1.2.0) (*Sievers et al., 2011*) with the default parameters, and the resulting multiple alignments were trimmed using trimAl (ver. 1.2) (*Capella-Gutiérrez et al., 2009*) in gappyout mode.

## Hierarchical clustering of the 11 Imp-β family NTRs based on the degree of overlap of the 3rd-Z-4% cargoes

We performed a hierarchical clustering of the Imp-β family NTRs based on their cargo profile similarities using Ward's method with Euclidean distance as implemented in the software R (RRID:SCR_001905; *R Development Core Team, 2012*). Here, we omitted Imp-β because its cargoes include many Imp-α-dependent indirect cargoes. To define a cargo profile for each NTR, we first defined a set of cargoes by merging the 3rd-Z-4% cargoes of the 11 NTRs other than Imp-β, which yielded a total of 426 cargoes. We then defined length 426 binary vectors for each NTR with a 1 for each cargo in the top 4% list and a 0 otherwise and input these 11 vectors into R to perform the clustering.

## Bead halo assay

The proteins and *Escherichia coli* extracts were prepared as described (*Kimura et al., 2013a*). The bead halo assays (*Supplementary file 2*) were performed as described (*Patel and Rexach, 2008*). Briefly, GST or GST-NTR was immobilized on glutathione-Sepharose (GE healthcare), and mixed with an extract of *E. coli* expressing a GFP-fusion protein in EHBN buffer (10 mM EDTA, 0.5% 1,6-hexanediol, 10 mg/mL bovine serum albumin, and 125 mM NaCl), and the binding was observed by fluorescent microscopy. The GTP-fixed mutant of Ran Q69L-Ran, which inhibits specific NTR–cargo interactions, was added to determine the specificity of the binding. The expression and degradation levels of the GFP-fusion proteins were analyzed, and the concentrations of GFP-moieties were quantified by triplicate quantitative Western blotting of the extracts with an anti-GFP antibody. Because the GFP-moiety weakly bound to GST-Trn-1, GST-Trn-2, and GST-Trn-SR in the bead halo assay, the concentrations of the GFP-fusion proteins and GFP (control) were equalized, and images were acquired and processed under identical condition. In contrast, because the GFP-moiety does not bind to GST-Imp-13, GST-Imp-β, or GST-Imp-α, the control reaction mixture for these NTRs contained higher concentration of GFP than any other GFP-fusion proteins. Three images (GST, GST-

NTR, and GST-NTR + Q69L-Ran) for each GFP-fusion protein were acquired under identical conditions, and the background intensities and dynamic ranges were equalized.

## Cell line
HeLa-S3 (RRID:CVCL_0058; mycoplasma, not detected) was obtained from Dr. Fumio Hanaoka (RIKEN).

## Antibodies
See *Supplementary file 12B*.

## GFP-fusion proteins used for in vitro transport
The GFP-fusion proteins used in *Figure 1—figure supplement 1C* were prepared as described (*Kimura et al., 2013a*). For accessions and references, see *Supplementary file 12B*. The SOX2 cDNA (pF1KB9652) was from Kazusa DNA Res. Inst. (Kisarazu, Japan), and the others were cloned from a HeLa cDNA library (SuperScript, Life Technology) by PCR.

## Database deposition
The mass spectrometry proteomics data have been deposited to the ProteomeXchange Consortium (RRID:SCR_004055; http://www.proteomexchange.org/) via the PRIDE (RRID:SCR_003411; *Vizcaíno et al., 2016*) partner repository with the dataset identifier PXD004655.

The .msf and .raw data files of each experiment summarized in *Supplementary file 1* are listed in *Supplementary file 12A*. The protein and peptide quantitation results can be seen by opening .msf files by Proteome Discoverer software. To see spectra and chromatograms, .msf files and corresponding .raw files must be in the same local directory. A demo version of Proteome Discoverer can be downloaded at the Thermo Scientific omics software portal site (https://portal.thermo-brims.com/).

## Acknowledgements
We thank Martin Beck, Alessandro Ori, and Kentaro Tomii for helpful discussion, Masahiro Oka for a supporting experiment, Shoko Motohashi and Ai Watanabe for technical assistance, and Hiroshi Mamada for a plasmid. We also thank the Support Unit for Bio-Material Analysis, RIKEN BSI Research Resource Center, especially Kaori Otsuki, Masaya Usui, and Aya Abe for MS analysis. This work was supported by JSPS KAKENHI Grant Number 15K07064 and a RIKEN FY2014 Incentive Research Project to MK, KAKENHI Grant Numbers 26251021, 26116526, and 15H05929 to NI, and the Platform Project for Supporting in Drug Discovery and Life Science Research (Platform for Drug Discovery, Informatics, and Structural Life Science) from the Japan Agency for Medical Research and Development (AMED) to KI.

## Additional information

### Funding

| Funder | Grant reference number | Author |
|---|---|---|
| Japan Society for the Promotion of Science | KAKENHI,15K07064 | Makoto Kimura |
| RIKEN | FY2014 Incentive Research Project | Makoto Kimura |
| Japan Agency for Medical Research and Development | Platform Project for Supporting in Drug Discovery and Life Science Research | Kenichiro Imai |
| Japan Society for the Promotion of Science | KAKENHI,26251021 | Naoko Imamoto |
| Japan Society for the Promotion of Science | KAKENHI,26116526 | Naoko Imamoto |

| Japan Society for the Promotion of Science | KAKENHI,15H05929 | Naoko Imamoto |
|---|---|---|

The funders had no role in study design, data collection and interpretation, or the decision to submit the work for publication.

### Author contributions

MK, Conceptualization, Formal analysis, Funding acquisition, Investigation, Visualization, Methodology, Writing—original draft, Project administration, Writing—review and editing; YM, Formal analysis, Investigation, Visualization, Writing—review and editing; KI, Formal analysis, Funding acquisition, Investigation, Visualization, Methodology, Writing—review and editing; SK, Conceptualization, Resources, Methodology, Writing—review and editing; PH, Supervision, Validation, Investigation, Methodology, Writing—review and editing; NI, Conceptualization, Supervision, Funding acquisition, Validation, Project administration, Writing—review and editing

### Author ORCIDs

Makoto Kimura, http://orcid.org/0000-0003-0868-5334
Naoko Imamoto, http://orcid.org/0000-0002-2886-3022

## Additional files

### Supplementary files

• Supplementary file 1. Results of SILAC-Tp. Each sheet contains the result of SILAC-Tp with one of the 12 NTRs. The proteins that exhibited $+NTR/Ctl = (L/H_{+NTR})/(L/H_{Ctl})$ values at least once in the three replicates (Experiments 1–3) are listed with the $\log_2[(L/H_{+NTR})/(L/H_{Ctl})]$ values. The second and third Z-scores and the ranks according to those scores are also presented. The 2nd-Z-15% and 3rd-Z-4% cargoes are indicated in cyan. The LC-MS/MS quantitation data for each replicate (Experiments 1–3) are also included. Light/Heavy, the median of quantified L/H values; Light/Heavy count, the number of quantified values; Light/Heavy variability, coefficient-of-variation for log-normal distributed data. [a]Report: The proteins listed by *Chook and Süel (2011)* are regarded as reported cargoes, and the references are provided in the Legend sheet. For Imp-$\beta$, only direct cargoes are listed. For Trn-2, the reported Trn-1 cargoes are listed. [b]Direct Binding: The results of the bead halo assays (*Supplementary file 2*) are summarized. ++ or +, positive; ± or −, negative. [c]GO Nucleus: Annotated with 'nucleus' in Gene Ontology (term type, cellular component). The rows can be sorted into preferable orders with Excel. To access the mass spectra, chromatograms, or raw data, see *Supplementary file 12A*. Statistics: For each experiment, the number of proteins assigned with $\log_2[(L/H_{+NTR})/(L/H_{Ctl})]$ values (proteins assigned with Z-scores), the mean and standard deviation (S.D.) of $\log_2[(L/H_{+NTR})/(L/H_{Ctl})]$ are listed. $Z - score = (X - \mu)/\sigma$ where $X = \log_2[(L/H_{+NTR})/(L/H_{Ctl})]$ of each protein, $\mu$ is the mean of X in one experiment, and $\sigma$ is the S.D. of $X$.

• Supplementary file 2. NTR–cargo direct binding. The direct binding of the candidate cargoes to the NTRs was analyzed by bead halo assay. From well-characterized proteins that have not been reported as cargoes, (i) proteins ranked high (within the top 15% in the 2nd-Z-rankings or 4% in the 3rd-Z-rankings), around presumptive cutoffs (within about top 15–25% in the 2nd-Z-rankings), or lower and (ii) highly ranked proteins that are suspected as indirect cargoes or false positives based on their well-known features, e.g., PMPCA, GALE, UAP1, NQO2, EEF1A2, RAB2A, RAB8A, S100A4, S100A6, S100A13, and S100P, were selected and analyzed. Proteins in (i) verify the cargo identification and cutoff setting, and proteins in (ii) serve for finding indirect cargoes and false positives. The negative rate of these bead halo assays should be higher than the true overall false positive rate of the SILAC-Tp, because proteins in (ii) were selected preferentially. GST or GST-NTR was attached to glutathione-Sepharose beads, mixed with an extract of *E. coli* expressing GFP or a GFP-fusion protein, and observed by fluorescence microscopy. Q69L-Ran, which inhibits the NTR–cargo functional binding, was added as appropriate. The contrast of the bead fluorescence between the GST and GST-NTR indicates the binding, and the inhibition of this binding by Q69L-Ran certifies the specificity of the binding; ++ or +, positive; ± or −, negative. Summary of the results (p2–5): The results are

summarized in both 2nd- and 3rd-Z-rank order. The 2nd-Z-15% and 3rd-Z-4% cargoes are indicated by cyan, and positive binding (++ or +) is indicated by blue. Trn-1 (p6–8): The GFP-fusion proteins were divided into five groups (A–E) according to the expression levels. Because GFP binds weakly to Trn-1, the concentrations of GFP (control) and GFP-fusion proteins were equalized within each group, and the binding was observed in the same conditions. The images are comparable within a group. Trn-2 (p9): GFP weakly binds to Trn-2, and the concentrations of GFP and GFP-fusion proteins were equalized. The images are comparable. Proteins whose ranks differed substantially between the Trn-1 and Trn-2 Z-ranking were assayed. Imp-13 (p10–13): GFP does not bind to Imp-13, and GFP was added to the control mixture at the highest concentration. Three images (GST, GST-Imp-13, and GST-Imp-13 + Q69L-Ran) for each GFP-fusion protein were acquired under identical conditions, and the background intensities and dynamic ranges were equalized. Trn-SR (p14–16): GFP weakly binds to Trn-SR, and the procedures were similar to those used for Trn-1. The GFP-fusion proteins were divided into four groups (A–D), and the images are comparable within a group. Imp-$\alpha/\beta$ (p17–20): GFP does not bind to Imp-$\alpha$ or -$\beta$, and the procedures were similar to those used for Imp-13. GST-Imp-$\alpha_2$ lacks the N-terminal Imp-$\beta$-binding domain. Western blotting (p21–22): The GFP-fusion proteins in the *E. coli* extracts were relatively quantified by Western blotting using an anti-GFP antibody (Roche). The extracts containing the amounts of protein (ng) indicated at the bottoms were loaded. The arrowheads indicate the expected full-length products. The GFP-moieties including those of the partial products were quantified by chemiluminescence. GFP was used as the standard. The Western blots were replicated more than three times. Accessions and sequences (p23–27): The cDNAs were cloned from a HeLa cDNA library by PCR. The accession numbers of the proteins are listed. If the sequence of a used protein is different from that in the database, the deleted, substituting, or inserted amino acids are indicated by the colors. The sequences that matched perfectly are not presented.

• Supplementary file 3. The 2nd-Z-15% cargoes of the 12 NTRs. The 2nd-Z-15% cargoes of each NTR are listed by the gene names in the 2nd-Z-rank orders. The ranks by the third Z-scores are also shown. Cyan in the rank columns indicates the 2nd-Z-15% and 3rd-Z-4% cargoes. Colors in the gene name columns: magenta, reported cargoes; blue, cargoes bound directly to the NTR in the bead halo assays (*Supplementary file 2*); light blue, cargoes bound directly to Imp-$\alpha$ but not Imp-$\beta$; gray, proteins that did not bind to the NTRs; yellow, Imp-$\alpha$; and green, reported export cargoes. For the 3rd-Z-4% cargoes, see *Figure 3*.

• Supplementary file 4. Example of extracted ion chromatograms (EICs) of peptides. EICs of GLUD1 peptides in the SILAC-Tp with Trn-1: As a general problem of high-throughput LC-MS/MS quantitation, quantitative values of proteins with fewer quantified peptides deviate among the replicates. Referring to EICs of the quantified peptides is useful to avoid misidentification of cargoes. For example, the L/H ratios of a Trn-1 2nd-Z-15% cargo GLUD1 (P00367, ranked 110th and 342nd by the second and third Z-score, respectively) deviated largely in the three replicates of SILAC-Tp with Trn-1 (*Supplementary file 1*). Panels (A–H) show EICs of the indicated peptide (trypsin targets, K and R, are written in lower cases) in the three (three Ctl and there +Trn) experiments (some peptides were not identified in all the experiments). Magenta letters indicate the quantified peptides and L/H rations. In panel (A), the elution time of the peptide TAMkYNLGLDLr differs largely between the expriment-1 Ctl and experiment-2 +Trn-1, the peak shape of the experiment-2 +Trn-1 is irregular, and the L/H ratio of it is much higher than those of other peptides in +Trn-1 experiments (B and E). Thus, there is concern about misidentification. Because the L/H count of GLUD1 in the experiment-2 +Trn-1 is two (TAMkYNLGLDLr and NLNHVSYGr, A and B) and the L/H ratio of a protein is defined as the median, the L/H ratio of GLUD1 in the expriment-2 +Trn-1 is affected by the L/H ratio of TAMkYNLGLDLr. Exclusion of the L/H ratio of TAMkYNLGLDLr in the experiment-2 +Trn-1 lowers the Z-score rank of GLUD1 significantly. In panel (C), the chromatogram of the peptide HGGTIPIVP-TAEFQDr in the experiment-1 Ctl has an irregular peak, and the L/H ratio of it is much higher than those of other peptides in the Ctl experiments (A–H). Thus, overlap with other peptide or other failures may be possible. However, the L/H count of GLUD1 in the experiment-1 Ctl is four (TAM-kYNLGLDLr, HGGTIPIVPTAEFQDr, ALASLMTYk, and GASIVEDkLVEDLr) and the value of HGGTIPIVPTAEFQDr does not affect the median. (The L/H ratio of TAMkYNLGLDLr, whose EIC

differ between the experiment-1 Ctl and expriment-2 +Trn in (A) as mentioned above, may affect the L/H ratio of GLUD1 in the experiment-1 Ctl, but we assumed that it is reliable.) As above, the L/H ratios of proteins with low L/H counts (*Supplementary file 1*) may be affected by LC-MS/MS artifacts, and misidentification can be avoided by referring to the EICs. All the EICs and MS spectra in this work can be accessed by downloading the mass spectrometry data and Proteome Discoverer software (see the Materials and methods and *Supplementary file 12A*).

• Supplementary file 5. Redundancy of NTRs: Cargoes shared by NTRs. (A) The numbers of the 2nd-Z-15% cargoes shared by two NTRs. For the 3rd-Z-4% cargoes, see *Figure 4*. (B) Redundancy of the 3rd-Z-4% cargoes. Many proteins are included in the 3rd-Z-4% cargoes of multiple NTRs. The ranks of these cargoes in the 3rd-Z-rankings for all 12 NTRs are presented. (C) Relationships between the NTRs and the characteristics of their 3rd-Z-4% cargo proteins. The 3rd-Z-4% cargoes are grouped according to their characteristics (functions or biological processes that the proteins act in), and their ranks are presented as in (B). To make our points clear, typical terms for the protein characteristics and typical proteins related to the terms have been selected with reference to Gene Ontology (GO) and UniProt. Thus, the terms in this sheet are slightly different from those in the databases, fewer proteins than annotated in the databases are grouped, and the list is redundant. For the complete linkages between the GO terms and the 3rd-Z-4% cargoes, see *Supplementary file 7*. (D) Redundancy of the 2nd-Z-15% cargoes. The ranks of the 2nd-Z-15% cargoes are presented as in (B). (E) Relationships between the NTRs and the characteristics of their 2rd-Z-15% cargo proteins. The 2nd-Z-15% cargoes are grouped, and their ranks are presented in a manner similar to that in (C). For the complete linkages between the GO terms and the 2nd-Z-15% cargoes, see *Supplementary file 8*.

• Supplementary file 6. GO term enrichments of the identified cargoes. (A) Extraction of the GO term enrichments of the 2nd-Z-15% cargoes. The 2nd-Z-15% cargoes were analyzed for GO term enrichment in (C). The terms that were significantly enriched (p<0.05, cyan) in the 2nd-Z-15% cargoes of four or fewer NTRs were selected, and terms that represent many similar terms are presented. With the p-values, the numbers (#) of cargoes annotated with each of the terms are presented. Total No. represents the number of proteins annotated with each term in the database. Related terms are bundled in the same color. For the 3rd-Z-4% cargoes, see *Figures 5* and *6*. The correspondences between each 2nd-Z-15% cargo and GO term are summarized in *Supplementary file 10*. All the GO terms annotated to the 2nd-Z-15% cargoes are listed in *Supplementary file 8*. (B) Full table of the GO term enrichments of the 3rd-Z-4% cargoes. The 3rd-Z-4% cargoes were analyzed for GO term enrichment. For all combinations of GO terms and NTRs, the p-values for the term enrichments in the 3rd-Z-4% cargoes and the numbers (#) of cargoes annotated with the terms are presented. The numbers following '# in' are the total numbers of 3rd-Z-4% cargoes. Total No. represents the number of proteins annotated with each term in the database. Cyan, p<0.05. *Figures 5* and *6* were extracted from this table. This table was derived from *Supplementary file 7*, and see *Supplementary file 7* to retrieve the protein accessions. (C) Full table of the GO term enrichments of the 2nd-Z-15% cargoes. The 2nd-Z-15% cargoes were analyzed and are presented in a manner similar to that in (B). (A) was extracted from this table. This table was derived from *Supplementary file 8*, and see *Supplementary file 8* to retrieve the protein accessions.

• Supplementary file 7. The 3rd-Z-4% cargoes annotated with GO terms. With respect to each NTR, the accessions of the 3rd-Z-4% cargoes annotated with each GO term are listed. Cyan, significant term enrichment (p<0.05) in the 3rd-Z-4% cargoes of the NTR. Total No. represents the number of proteins annotated with each term in the database. *Supplementary files 6B* and *9* were derived from this table. For the 2nd-Z-15% cargoes, see *Supplementary file 8*.

• Supplementary file 8. The 2nd-Z-15% cargoes annotated with GO terms. With respect to each NTR, the accessions of the 2nd-Z-15% cargoes annotated with each GO term are listed. Cyan, significant term enrichment (p<0.05) in the 2nd-Z-15% cargoes of the NTR. Total No. represents the number of proteins annotated with each term in the database. *Supplementary files 6C* and *10* were derived from this table. For the 3nd-Z-4% cargoes, see *Supplementary file 7*.

• Supplementary file 9. Correspondences between the 3rd-Z-4% cargoes and GO terms. Each sheet shows the correspondences between the 3rd-Z-4% cargoes of one NTR and selected GO terms. A term annotation to a cargo is indicated by '1' in the corresponding cell. Reported cargoes are indicated by magenta in the gene name cells, and the results of the bead halo assays (*Supplementary file 2*) are also indicated by colors in the gene name cells: blue, cargoes directly bound to the NTR; light blue, cargoes directly bound to Imp-$\alpha$ but not Imp-$\beta$; gray, proteins that did not bind to the NTR. GO terms that represent many similar terms were selected from the terms enriched significantly ($p<0.05$) for the 3rd-Z-4% cargoes of each NTR, and broadly defined terms were deselected. Magenta and orange in the term ID cells indicate terms that are significantly enriched for the cargoes of four or fewer NTRs, and of them magenta indicates the terms presented in *Figures 5* and *6*. Related GO terms are bundled in the same color, and different colors are used to distinguish the columns easily. The NTRs added in the transport reactions (white in the rank cells) were not analyzed. This table was derived from *Supplementary file 7*. For 2nd-Z-15% cargoes, see *Supplementary file 10*.

• Supplementary file 10. Correspondences between the 2nd-Z-15% cargoes and GO terms. Each sheet shows the correspondences between the 2nd-Z-15% cargoes of one NTR and selected GO terms. A term annotation to a cargo is indicated by '1' in the corresponding cell. Reported cargoes are indicated by magenta in the gene name cells, and the results of the bead halo assays (*Supplementary file 2*) are also indicated by colors in the gene name cells: blue, cargoes directly bound to the NTR; light blue, cargoes directly bond to Imp-$\alpha$ but not Imp-$\beta$; gray, proteins that did not bind to the NTR. GO terms that represent many similar terms were selected from the terms enriched significantly ($p<0.05$) for the 2nd-Z-15% cargoes of each NTR, and broadly defined terms were deselected. Magenta and orange in the term ID cells indicate terms that are significantly enriched for the cargoes of four or fewer NTRs, and of them magenta indicates the terms presented in *Supplementary file 6A*. Related GO terms are bundled in the same color, and different colors are used to distinguish the columns easily. The NTRs added in the transport reactions (white in the rank cells) were not analyzed. This table was derived from *Supplementary file 8*. For 3rd-Z-4% cargoes, see *Supplementary file 9*.

• Supplementary file 11. mRNA processing factors, ribosomal proteins, and transcription factors. (A) Ranks of mRNA processing factors. All of the 2nd-Z-15% and 3rd-Z-4% cargoes that are annotated with mRNA processing in Gene Ontology (GO) are listed with the ranks in the 2nd- and 3rd-Z-rankings. The color scale is set by percentile rank as indicated. *Figure 7* was extracted from this table. (B) The 3rd-Z-rankings of ribosomal proteins. The ranks of the ribosomal proteins in the 3rd-Z-rankings of the 12 NTRs are presented. The color scale is set by percentile rank as indicated. For the 2nd-Z-rankings, see *Figure 8*. (C) Extracts of the GO term enrichments of the transcription factors found in the 2nd-Z-15% cargoes. Seventeen to 36 proteins in the 2nd-Z-15% cargoes of each NTR are annotated with 'transcription factor activity, sequence-specific DNA binding' or 'transcription factor activity, protein binding' in GO. The factors were analyzed for GO term enrichment (term type, biological process, BP) in (D). Typical terms that represent similar terms were extracted from (D). The p-value for the term enrichment and the number (#) of factors annotated with the term are presented. Total No. represents the number of proteins annotated with each term in the database. Cyan, $p<0.05$. Related terms are bundled in the same color. (D) Full table of the GO term enrichments of the transcription factors found in the 2nd-Z-15% cargoes. The 2nd-Z-15% cargoes that are annotated with 'transcription factor activity, sequence-specific DNA binding" or 'transcription factor activity, protein binding' in GO were analyzed for GO term enrichment (term type, BP). The analyzed transcription factors are listed at the bottom. For all of the combinations of GO terms and NTRs, the p-value for the term enrichment and the number (#) of transcription factors annotated with the term are presented. The numbers following '# in' are the total numbers of transcription factors in the 2nd-Z-15% cargoes. Total No. represents the number of proteins annotated with each term in the database. Cyan, $p<0.05$. (C) was extracted from this table. (E) SRSRSR motif in the 3rd-Z-4% cargoes. The 3rd-Z-4% cargoes that contain an 'SRSRSR' hexa-peptide sequence were counted.

• Supplementary file 12. MS data files, recombinant cargoes, and antibodies. (A) MS data files. The mass spectrometry proteomics data (.msf and .raw files) have been deposited to the ProteomeXchange Consortium (http://www.proteomexchange.org/) with the dataset identifier PXD004655. The results of protein and peptide identification and quantitation are summarized in Supplementary file 1, and the .msf and .raw data files corresponding to each experiment in Supplementary file 1 are listed in this table. The quantitation results can be seen by opening .msf files by Proteome Discoverer software. To see spectra and chromatograms, .msf files and corresponding .raw files must be in the same local directory. A demo version of Proteome Discoverer can be downloaded at the Thermo Scientific omics software portal site (https://portal.thermo-brims.com/). (B) GFP-fusion proteins used for in vitro transport and antibodies.

## Major datasets

The following dataset was generated:

| Author(s) | Year | Dataset title | Dataset URL | Database, license, and accessibility information |
|---|---|---|---|---|
| Kimura M, Imamoto N | 2016 | SILAC-Tp (12 importins) | http://www.ebi.ac.uk/pride/archive/projects/PXD004655 | Publicly available at the Pride Archive (accession no: PXD004655) |

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
