## [Decision Letter]

Thank you for submitting your article "Extensive cargo identification reveals distinct biological roles of the 12 importin pathways" for consideration by *eLife*. Your article has been favorably evaluated by James Manley (Senior Editor) and three reviewers, one of whom, Karsten Weis (Reviewer #1), is a member of our Board of Reviewing Editors. The following individuals involved in review of your submission have agreed to reveal their identity: Yuh Min Chook (Reviewer #2); John D Aitchison (Reviewer #3).

The reviewers have discussed the reviews with one another and the Reviewing Editor has drafted this decision to help you prepare a revised submission.

Summary:

In this manuscript, Kimura and colleagues employ a functional in vitro transport assay combined with SILAC-based mass spectrometry experiments to identify in a comprehensive manner cargoes for 12 nuclear transport receptors (NTRs; 10 human Importins and 2 bidirectional Karyopherins). In general, the work is of high quality and carefully analyzed. The results should be of interest to a very broad audience and provide an excellent resource for researchers studying a variety of nuclear processes. However, there are some concerns regarding the data presentation and interpretation that need to be addressed prior to publication.

Required revisions:

1) Certain aspects of the manuscript are not well presented. This is particularly problematic in the second part of the paper where the authors present a GO term analysis for the various NTRs. These results could simply and more effectively presented in tabular form (including a list of the actual cargoes that fall into each GO term). Then the highlights of this analyses should be mentioned and discussed. Claims of collective activity of individual or a few NTRs in designated cellular processes from this GO term analysis should be avoided in the absence of experimental support.

2) Lists like the ones shown in Figure 2 and lists of 3rd-Z-4% cargoes for each of the 12 NTRs should be easily accessible in the main or supplemental figures.

3) Additional supportive information needs to be added to increase the confidence in the presented results. For example, information on the results of halo binding assays is not easily accessible. It was not clearly shown how many of their "highly reliable" (3rd-Z-4%) cargoes were tested for direct NTR binding, what fraction of those tested bound the NTRs and which 3rd-Z-4% cargo did not bind directly to its NTR. Also, it is unclear what criteria were applied to select proteins for the bead halo assay.

4) There is a concern about the 'context' of cargoes used in the assays. The authors added nuclear extract as the source of cargoes to digitonin-permeabilized cells. This is presumably because nuclear extract is much more enriched than cytoplasmic extract in import cargoes as most ultimately reside in the nucleus. However, many cargoes may assume different 'states' (binding partners, PTM, etc.) that are quite different from their cytoplasmic states. Nuclear import cargoes are recognized by their NTRs in the cytoplasm, hence in their cytoplasmic states and not in their nuclear states. Some proteins in nuclear extract may respond to different NTRs than they would in their cytoplasmic states. Have the authors attempted to address this? At least this should be discussed as a potential caveat.

5) How do the authors explain well-established functions of Imp-9 such as import of actin into the nucleus? Imp-9 and Exp-6 (which exports actin) are used routinely to control levels of nuclear actin, but it is curious that the authors did not find actin or actin-binding proteins in their list of potential Imp-9 cargoes.

6) How does the approach consider endogenous NTRs that are abundant in the nucleoplasm at steady state? Presumably these proteins are still present and able to cycle during the import reaction?

7) Why are there such high numbers of proteins detected as imported in the absence of recombinant NTRs? Is this due to residual NTRs, either in the nucleus or cytosolic fraction?

8) Depletion of β-1 from the cytoplasm presumably depletes α. Is this quantified and incorporated into the analysis?

9) 0.3 to 0.7 μM of NTR was used in the experiment. How was this concentration decided for each NTR? How does this compare to the endogenous concentration of each NTR?

10) Subsection “SILAC-Tp”, end of first paragraph – How was the nuclear extract prepared after import? The authors only mention that the cells were suspended in nuclear buffer, sonicated and centrifuged.

11) Based on the SILAC-tp experimental design, a positive Z-score value of a particular protein would mean that the given protein was imported into the heavy-nuclei. A large number of proteins are listed with negative 2nd Z-scores. Does this mean that these proteins were 'exported' out of the recipient nuclei in the presence of the various NTRs?

12) [Supplementary-material SD2-data], Sheet: 'Trn-1' contains three columns of L/H ratios for each identified protein. For some top ranked proteins, (e.g. mitochondrial glutamate dehydrogenase), a large variation between SILAC ratios is observed between the three replicates. For some proteins, no ratios were calculated in some technical replicates. Was this because of low quality LCMS data or due to low abundance of these proteins? For such cases, the authors need to include an extracted ion chromatogram (EIC) of a respective peptide from all replicates to exclude the possibility of SILAC quantification artifacts. (By the way, why is this protein, among other cytoplasmic proteins, present in the nuclear fraction? The authors should comment and edit as necessary).

13) The authors don't specify whether or not they considered 2 or more peptides/protein for a SILAC ratio. Quantification based on a single peptide with sub-optimal signal quality may affect the L/H ratios. A detailed description should be included in the Methods section on the criteria and software used to calculate SILAC ratios.

14) The statistical language (e.g. definition of statistical parameters and their use to justify designated cargoes) would benefit from better visual support via figures/tables.

15) In general, the Z-ranking is cumbersome and difficult to interpret. On the whole, the addition of some descriptive statistics would be helpful for the interpretation; however, they are unlikely to significantly change the conclusions.

16) The cutoffs of 15% (and others) are arbitrary. To determine if the 'reported cargo' proteins are lumped at one end of the ranked list, the authors could use something like a Kolmogorov-Smirnov (KS) or Mann-Whitney (MW) test.

17) It should also be possible to estimate the sensitivity and a lower bound on the specificity at different index cutoffs using those receptors with high numbers of reported cargoes (i.e. TRN-1 and 2). The sensitivity at a cutoff will be (# reported cargoes selected) / (# reported cargoes selected + # reported cargoes *not* selected). The specificity is harder to quantify since the 'not carriers' is not known. However, one could consider all of the 'grey bar' proteins in Figure 1 as negatives, then one could estimate the upper bound on the number of false positives. An estimated sensitivity and specificity like this would help ground otherwise arbitrary cutoffs (e.g. top 100 proteins) as something interpretable. It would also help explain what it means for a cargo to pass the 2nd-Z-15% and 3rd-Z-4% criteria.

18) The enrichments of reported cargoes in highly ranked proteins should have a p-Value, for example as calculated from the hypergeometric distribution.

19) 'To calculate the +NTR/Ctl value, one protein has to be quantified in both the control and +NTR reactions, and we discarded L/H_+NTR_ values that lacked the counterpart L/H_Ctl_ values.' Wouldn't this remove the most interesting hits? That is to say, proteins that can only get into the nucleus through the added receptor?

20) In the second paragraph of the Introduction, the authors mentioned that NTR binding sites for cargoes have been established for Importins α, β, Trn-1/2 and Trn-SR. They missed the Imp-5 and its homologous Kap121p systems, where the Matsuura group very nicely elucidated the binding site for IK-NLSs found in Imp5/Kap121p cargoes.

21) In the second paragraph of the Introduction, the last part of the sentence "…the PY-NLS motif has been defined, although the motif is not an absolute determinant of transport (Soniat and Chook, 2015)." is a bit confusing. Do the authors mean that presence of a PY-NLS does not necessarily determine transport, or do they mean that there are other motifs that may direct Trn-1/2 mediated transport?

---

## [Author Response]

*Required revisions:*

*1) Certain aspects of the manuscript are not well presented. This is particularly problematic in the second part of the paper where the authors present a GO term analysis for the various NTRs. These results could simply and more effectively presented in tabular form (including a list of the actual cargoes that fall into each GO term). Then the highlights of this analyses should be mentioned and discussed. Claims of collective activity of individual or a few NTRs in designated cellular processes from this GO term analysis should be avoided in the absence of experimental support.*

We added lists of the 3rd-Z-4% cargoes (Figure 3) and the 2nd-Z-15% cargoes ([Supplementary-material SD4-data]) in the revised manuscript. We also added tabular diagrams that show the correspondences between the cargoes and GO terms (3rd-Z-4% cargoes, [Supplementary-material SD10-data]; 2nd-Z-15% cargoes, [Supplementary-material SD11-data]). We deleted two sentences that claim specific biological functions of NTRs: “Thus, Imp-13 may regulate gene-specific transcription.” and “Thus, the Imp-13 pathway is related to protein SUMOylation.” Also, we rewrote other sentences claiming NTR functions in cellular processes (subsections “Imp-β cargoes”, “Trn-SR cargoes”, first paragraph and “Exp-4 cargoes”), and now only the functions annotated to the cargoes are described. Additionally, we inserted the sentences “Biological functions linked to NTRs by the natures of their cargoes need to be verified by further experiments.” (subsection “Characterizations of the cargoes of individual NTRs”) and “The biological functions of NTRs designated in this work should be further clarified in future experiments” (subsection "Additional remarks"). In other parts we only state the direct results of database search, such as that an NTR is linked to a term of biological process by its cargoes.

*2) Lists like the ones shown in Figure 2 and lists of 3rd-Z-4% cargoes for each of the 12 NTRs should be easily accessible in the main or supplemental figures.*

We added Figure 3 and [Supplementary-material SD4-data] (see the reply to point 1). These figures present the 3rd-Z-4% (Figure 3) and 2nd-Z-15% ([Supplementary-material SD4-data]) cargoes of the 12 NTRs and the relations between their 2nd- and 3rd-Z-ranks.

*3) Additional supportive information needs to be added to increase the confidence in the presented results. For example, information on the results of halo binding assays is not easily accessible. It was not clearly shown how many of their "highly reliable" (3rd-Z-4%) cargoes were tested for direct NTR binding, what fraction of those tested bound the NTRs and which 3rd-Z-4% cargo did not bind directly to its NTR. Also, it is unclear what criteria were applied to select proteins for the bead halo assay.*

We added summary tables in [Supplementary-material SD3-data]. These tables present the results of the bead halo assays in both the 2nd- and 3rd-Z-rank order, and the 2nd-Z-15% and 3rd-Z-4% cargoes are marked. We also indicated the results of the bead halo assays (the proteins that did not bind to their NTRs as well as those bound) in Figure 1 and Figure 3, Figure 2—figure supplement 2 and Figure 2—figure supplement 3, and [Supplementary-material SD4-data], [Supplementary-material SD10-data] and [Supplementary-material SD11-data] (cargoes that did and did not bind directly to their NTRs are indicated in blue or dark gray, respectively). We added descriptions about the protein selection for bead halo assay (subsection “SILAC-Tp effectively identifies cargoes”, fifth paragraph and in [Supplementary-material SD3-data]). As described in those parts, we principally selected well-characterized proteins that have not been reported as cargoes from (i) proteins ranked high (3rd-Z-4% or 2nd-Z-15% cargoes), around presumptive cutoffs (within about top 15%–25% in the 2nd-Z-ranking), or lower and (ii) highly ranked proteins that are suspected as indirect cargoes or false positives based on their well-known features; S100 proteins, Rab family proteins, EF1α, etc. Because we selected candidate cargoes to test in this manner, the confirmation rate of the bead halo assays is expected to be substantially lower than it would be for randomly selected cargo candidates.

*4) There is a concern about the 'context' of cargoes used in the assays. The authors added nuclear extract as the source of cargoes to digitonin-permeabilized cells. This is presumably because nuclear extract is much more enriched than cytoplasmic extract in import cargoes as most ultimately reside in the nucleus. However, many cargoes may assume different 'states' (binding partners, PTM, etc.) that are quite different from their cytoplasmic states. Nuclear import cargoes are recognized by their NTRs in the cytoplasm, hence in their cytoplasmic states and not in their nuclear states. Some proteins in nuclear extract may respond to different NTRs than they would in their cytoplasmic states. Have the authors attempted to address this? At least this should be discussed as a potential caveat.*

We added the following sentence in the last part of the Results and discussion; “We used HeLa nuclear extract as the cargo source, but it might not reconstitute all NTR–cargo interactions precisely because proteins in the nuclear extract might have different modifications or binding partners from those in cytoplasm where NTRs bind to cargoes in vivo”.

*5) How do the authors explain well-established functions of Imp-9 such as import of actin into the nucleus? Imp-9 and Exp-6 (which exports actin) are used routinely to control levels of nuclear actin, but it is curious that the authors did not find actin or actin-binding proteins in their list of potential Imp-9 cargoes.*

As a general shortcoming of our results, some reported cargoes including actin as an Imp-9 cargo were ranked lower. We added possible explanations for it in the last part of the Results and discussion. One possible explanation is that the protein context noted by the reviewer in point 4 is different between our transport system and the actual state in vivo. Alternatively, the transport capacity of our in vitro transport system might not be enough to identify all the cargoes, and especially cargoes that bind weakly to the NTR may be competed out by those bind strongly.

*6) How does the approach consider endogenous NTRs that are abundant in the nucleoplasm at steady state? Presumably these proteins are still present and able to cycle during the import reaction?*

As mentioned in the Materials and methods, the permeabilized cells were incubated (pretreated) with Ran and ATP regeneration system before the transport reaction (subsection “SILAC-Tp”, first paragraph), and this pretreatment removed NTRs endogenous to the permeabilized cell nuclei as revealed by Western-blotting (see Figure 9). The removing efficiency was not so high as compared with the depletion of NTRs from nuclear and cytoplasmic extracts by phenyl-Sepharose (Figure 1—figure supplement 1). However, the residual NTRs in the permeabilized cell nuclei must be negligible. They must once move out of the nuclei to cycle the import reaction, and they will be diluted extensively in the transport mixture outside the nuclei. We applied as much as 0.25 mL of transport mixture to only 3.5 × 10^5^ permeabilized cells in one reaction, and this mixture contained the cytosolic and nuclear extracts prepared from more than hundred times as many cells. Thus, the effect of the residual NTRs in the permeabilized cells must be negligible, if any, because of the extensive dilution.

Author response image 1.NTRs in the nuclei of permeabilized cells (Western).**DOI:**
http://dx.doi.org/10.7554/eLife.21184.029

More importantly, proteins transported by the residual NTRs must not be ranked higher in our system. The residual NTRs transport proteins at the same levels, if any, in the control and +NTR reactions, or the transport by the residual NTRs in the +NTR reaction might be repressed by the competition with the recombinant NTR. We calculated the index value as the ratio of the L/H values in the presence and absence of a recombinant NTR, i.e., (L/H_+NTR_)/(L/H_Ctl_), and thus the (L/H_+NTR_)/(L/H_Ctl_) value is not raised by the residual NTRs (see also the reply to point 7). For a few proteins, a hetero-dimer of NTRs is reported to act as an import receptor, and in such cases also the import must depend on the recombinant NTR added, even if it forms a dimer with a residual NTR.

*7) Why are there such high numbers of proteins detected as imported in the absence of recombinant NTRs? Is this due to residual NTRs, either in the nucleus or cytosolic fraction?*

Those must include (i) proteins that migrated into the nuclei by diffusion [small proteins (<50 kDa) permeate through the nuclear pores] independently of NTRs, (ii) proteins that were imported by the residual NTRs (if not negligible; see the reply to point 6), and (iii) proteins nonspecifically bound to the nuclear envelope or cytoplasmic structures. However, light proteins contained more in the +NTR sample than in the Ctl sample are imported proteins, and other proteins including (i-iii) should be detected at the same level in the Ctl and +NTR samples. We have addressed this point in our previous paper (Kimura et al., Mol. Cell. Proteomics 12, 145-157, 2013; Supplemental Data Figure S2). In the experiment presented there, we labeled total proteins in the nuclear extract by biotin and added the biotin-labeled extract to the in vitro transport system. The labeled proteins were visualized with Cy3-streptavidin in microscopy, or after protein extraction and SDS-PAGE, visualized with HRP-avidin in membrane-blotting. Consistent with the present result of SILAC-Tp, the microscopic observation indicated that some proteins migrated into the nuclei and others bound nonspecifically to the nuclear envelopes or cytoplasmic structures in the absence of recombinant NTRs. The membrane-blotting indicated that the labeled proteins detected in the Ctl sample (without a recombinant NTR) did not increase in the +NTR sample, and these are the proteins (i-iii) mentioned above. Concomitantly, in the membrane-blotting, many proteins that were not detected in the Ctl sample were detected in the +NTR sample. Meanwhile, the microscopic observation clearly indicated that more proteins were imported into the nuclei in the presence of a recombinant NTR. Thus, the proteins that increased in the +NTR sample were imported proteins and we use the ratio of +NTR/Ctl=(L/H_+NTR_)/(L/H_Ctl_) for the index.

In some actual cases, however, the overall level of the proteins (i-iii) differs between the Ctl and +NTR reactions, and thus the overall (L/H_+NTR_)/(L/H_Ctl_) values also vary among the three replicates (Figure 1—figure supplement 2). Therefore, we used Z-score for the normalization.

In our experimental system, the type (iii) proteins were reduced effectively. After a transport reaction, we washed the permeabilized cells by cytosolic extract with ATP and NaCl-transport buffer successively (subsection “SILAC-Tp”, first paragraph). As shown in Supplemental Data Figure S3 in Kimura et al., Mol. Cell. Proteomics 12, 145-157, 2013, the type (iii) proteins were removed effectively by the washing. However, the extract inevitably contains traces of type (iii) proteins that were not removed.

*8) Depletion of β-1 from the cytoplasm presumably depletes α. Is this quantified and incorporated into the analysis?*

We depleted the cytoplasmic and nuclear extracts of NTRs by four rounds of phenyl-Sepharose treatments. We have analyzed the levels of Imp-α1 in the cytosolic extract before and after one and four rounds of depletion by Western-blotting (see Figure 10; the total protein concentrations and the loaded volumes are indicated). About half of Imp-α1 was still present in the extract after four rounds of depletion. In this work we added 0.7 μM of Imp-β, which is optimum for the Imp-β direct cargo RPL23 (Figure 1—figure supplement 1). The optimum concentrations for Imp-α dependent recombinant cargoes are 2 and 1 μM for Imp-α and Imp-β, respectively, and we have applied these concentrations in Kimura et al., J. Biol. Chem. 288, 24540-24549, 2013. Although we did not add recombinant Imp-α in this work, many Imp-α dependent as well as Imp-β direct cargoes were actually identified as verified by the bead halo assays ([Supplementary-material SD3-data]). Thus, the Imp-β cargoes identified in this work fully represent cargoes that Imp-β imports in the presence of Imp-α.

Author response image 2.Imp-α1 in depleted cytosolic extract (Western).**DOI:**
http://dx.doi.org/10.7554/eLife.21184.030

*9) 0.3 to 0.7 μM of NTR was used in the experiment. How was this concentration decided for each NTR? How does this compare to the endogenous concentration of each NTR?*

We determined the optimum concentration of each NTR using the GFP-fusion proteins of reported cargoes shown in Figure 1—figure supplement 1. The figure supplement shows that the reported cargoes are imported effectively by each specific NTR. We added the description about it in the Materials and methods (subsection “SILAC-Tp”, first paragraph). In our previous paper (Kimura et al., Mol. Cell. Proteomics 12, 145-157, 2013), we have confirmed that proteins in the nuclear extract are also effectively imported by the optimum NTR concentrations determined for the recombinant cargoes, using the biotin-labeled nuclear extract in the same experimental system as described in the reply to point 7.

At this moment, we do not know the in vivo concentrations of the NTRs. In our estimation, however, the concentrations of the recombinant NTRs that we used are not much different from those in our total (undepleted) cytosolic extract, which is competent to transport without additional NTRs. In Figure 11 (the total protein concentrations and volumes of the loaded extracts are indicated), the levels of NTRs in the cytosolic extracts are compared with each recombinant NTR by Western-blotting. 0.3–0.7 μM of NTR (97–130 kDa) corresponds to 110–350 and 55–170 ng in 3.8 and 1.9 μL of the cytosolic extract, respectively.

Author response image 3.NTR concentrations in total and depleted cytosolic extracts (Western).(Left panel) Reproduced from Kimura et al., 2013, Mol. Cell. Proteomics 12:145-157.**DOI:**
http://dx.doi.org/10.7554/eLife.21184.031

*10) Subsection “SILAC-Tp”, end of first paragraph – How was the nuclear extract prepared after import? The authors only mention that the cells were suspended in nuclear buffer, sonicated and centrifuged.*

As we described in the reply to point 7, after the transport reaction we washed the permeabilized cells by depleted cytosolic extract with ATP and NaCl-transport buffer successively to remove the nonspecifically binding proteins. At this point, the permeabilized cells lost most of the cell membranes and cytosols. Then, we just suspended the cells in the nuclear buffer, and sonicated and centrifuged. The nuclear buffer used here is the same buffer as we used for the preparation of the nuclear extract that was added in the transport reaction, and we did not use denaturant or detergent for the protein extraction. Since the cargo source was the nuclear extract, we needed to quantify the same proteins that are soluble in the same buffer. This buffer selection served to increase the preferable proteins to be quantified in the LC-MS samples by discarding proteins insoluble in this buffer. However, the total protein species in the nuclear extract and those in the final samples may be different. As the reviewer is concerned, the final sample still contains many non-nuclear proteins, but as we described in the reply to point 7, the levels of the nonspecifically binding and non-nuclear proteins should be all the same between the Ctl and +NTR samples, whereas the imported proteins should be contained more in the +NTR samples than the Ctl samples.

As we noted in the reply to point 7, however, the overall level of nonspecifically binding proteins sometimes differs between the Ctl and +NTR samples in the actual cases, and consequently the overall (L/H_+NTR_)/(L/H_Ctl_) values also vary among the three replicates (Figure 1—figure supplement 2). We normalized this deviation among the replicates using Z-score.

*11) Based on the SILAC-tp experimental design, a positive Z-score value of a particular protein would mean that the given protein was imported into the heavy-nuclei. A large number of proteins are listed with negative 2nd Z-scores. Does this mean that these proteins were 'exported' out of the recipient nuclei in the presence of the various NTRs?*

At the start point of a transport reaction, the heavy proteins reside in the nuclei and light proteins are out of the nuclei. If one protein is exported in one direction, the heavy molecules in the nuclei is decreased and the (L/H_+NTR_)/(L/H_Ctl_) value rises. Thus, export cargoes may also exhibit high Z-scores. We found one such example, a well-characterized export cargo eIF1A, in the Imp-13 2nd-Z-15% cargoes (ranked 84th and 164th by the 2nd and 3rd Z-score, respectively). We do not generalize this observation because we have only one example. All the highly-ranked proteins that bound to Imp-13 in the bead halo assays dissociated in the presence of RanGTP, and therefore most of the identified Imp-13 cargoes are import cargoes. We discussed these in the Results and discussion (subsection “Imp-13 cargoes”, last paragraph). We also added a statement on the similar possibility for another bi-directional NTR Exp-4 in the Results and discussion (subsection “Exp-4 cargoes”).

We explained the definition and interpretation of Z-score in the reply to point 15. If the log_2_[(L/H_+NTR_)/(L/H_Ctl_)] of a protein equals to the mean of it in the replicate, the Z-score of the protein is 0. If the log_2_[(L/H_+NTR_)/(L/H_Ctl_)] values are higher or lower than the mean, the Z-scores are positive or negative, respectively. Thus, a negative Z-score does not mean that the L/H_+NTR_ is lower than the L/H_Ctl_. (Even if all the proteins are imported, the Z-scores of the proteins imported less than the mean level are negative.)

*12) [Supplementary-material SD2-data], Sheet: 'Trn-1' contains three columns of L/H ratios for each identified protein. For some top ranked proteins, (e.g. mitochondrial glutamate dehydrogenase), a large variation between SILAC ratios is observed between the three replicates. For some proteins, no ratios were calculated in some technical replicates. Was this because of low quality LCMS data or due to low abundance of these proteins? For such cases, the authors need to include an extracted ion chromatogram (EIC) of a respective peptide from all replicates to exclude the possibility of SILAC quantification artifacts. (By the way, why is this protein, among other cytoplasmic proteins, present in the nuclear fraction? The authors should comment and edit as necessary).*

The 3rd-Z-4% cargoes have relatively stable Z-scores or L/H ratios in the three replicates because of the selection method, but as the reviewer pointed, some of the 2nd-Z-15% cargoes have deviated Z-scores or L/H ratios and some of them have only two Z-scores. This is because they are selected by the 2nd Z-scores from proteins that have two or three Z-scores, and therefore a protein with the 3rd Z-score deviating downward can be selected as a cargo if the 2nd Z-score is high as described (subsection “SILAC-Tp effectively identifies cargoes”, second paragraph). The causes of these deviations can be either low quality LC-MS/MS data and/or low L/H counts (low numbers of quantified peptides) of low-abundance proteins. We added a statement on this deviation in the Results and discussion (subsection “SILAC-Tp effectively identifies cargoes”, end of seventh paragraph). As an example of such proteins, we presented EICs of all the identified peptides of mitochondrial glutamate dehydrogenase (GLUD1; the example pointed by the reviewer) from all the three replicates with Trn-1 in [Supplementary-material SD5-data]. In the legend of [Supplementary-material SD5-data], we discussed about the qualities of EICs and the effect of them on the quantitation values of proteins with low L/H counts.

All the EICs can be accessed by downloading the mass spectrometry data from ProteomeXchange database (http://www.proteomexchange.org/), and opening them by Proteome Discoverer software. A free demo version of Proteome Discoverer can be downloaded at the Thermo Scientific omics software portal site (https://portal.thermo-brims.com/). We have renamed the raw data files in the ProteomeXchange and now EICs of interest can be accessed more easily. We also added [Supplementary-material SD13-data], in which the correspondences between the experiments summarized in [Supplementary-material SD2-data] and the data files in ProteomeXchange are tabulated. The dataset identifier of ProteomeXchange is PXD004655, which the reviewers can access using the reviewer account. The data will be made publicly available upon publication of this paper.

Some mitochondrial proteins other than GLUD1 were also highly ranked for other NTRs. We have stated the possible reason why these proteins are identified as highly ranked cargoes in the Results and discussion (subsection “Seemingly non-nuclear proteins”, first paragraph), and now we added more explanation below it. In short, some proteins localize both in the mitochondria and the nuclei, or alternatively mitochondrial proteins could interact with NTRs in the in vitro transport system if it is fortuitous, and a highly sensitive LC-MS/MS system can quantify it.

*13) The authors don't specify whether or not they considered 2 or more peptides/protein for a SILAC ratio. Quantification based on a single peptide with sub-optimal signal quality may affect the L/H ratios. A detailed description should be included in the Methods section on the criteria and software used to calculate SILAC ratios.*

We accepted proteins quantified by a single peptide and added the statement in the Materials and methods (subsection “Protein identification and quantitation”, end of first paragraph). We also added the software name (Proteome Discoverer) and the parameters (default) used for the SILAC L/H ratio calculation in the Materials and methods (in the aforementioned paragraph). The L/H count of each quantified protein in each experiment is presented in [Supplementary-material SD2-data]. The EIC of each quantified peptide and MS spectra can be accessed as mentioned in the reply to point 12. Our method ensures a certain reliability if it includes quantitation by a single peptide, because we replicated it three times.

*14) The statistical language (e.g. definition of statistical parameters and their use to justify designated cargoes) would benefit from better visual support via figures/tables.*

Our responses to points 15, 16, and 17 also relate to this point. We added the definition of Z-score in the Materials and methods (subsection “Protein identification and quantitation”, last paragraph), the legend for Figure 1—figure supplement 2, and [Supplementary-material SD2-data], and the statistical values used for the calculation of Z-scores, i.e., the means and standard deviations, were also added in the legend sheet of [Supplementary-material SD2-data] (see the reply to point 15). We also calculated the reported cargo rates (precision), recall (sensitivities), and Fisher’s exact test p-values in 1% rank increments of cutoffs for the Trn-1 2nd- and 3rd-Z-rankings (Figure 1—figure supplement 3 and Figure 1—source data file 1), and we described their definitions and assumptions in the Result and discussion (subsections “SILAC-Tp effectively identifies cargoes”, third paragraph and “Cargo selection with higher specificity”, first paragraph) and the Materials and methods (subsection “Reported cargo rate and recall”) (see the reply to points 16 and 17).

*15) In general, the Z-ranking is cumbersome and difficult to interpret. On the whole, the addition of some descriptive statistics would be helpful for the interpretation; however, they are unlikely to significantly change the conclusions.*

We have tried some other methods that do not use Z-scores, but as the reviewer’s anticipation the conclusions are not significantly changed. We concluded that ranking by Z-scores is appropriate for data processing in this work. We added the definition of Z-score in the Materials and methods (subsection “Protein identification and quantitation”, last paragraph), the legend for Figure 1—figure supplement 2, and [Supplementary-material SD2-data]. We also added the descriptive statistic values used for the Z-score calculation, the mean and standard deviation of the log_2_[(L/H_+NTR_)/(L/H_Ctl_)] values in each experiment, in the legend sheet of [Supplementary-material SD2-data].

In this work, the +NTR/Ctl=(L/H_+NTR_)/(L/H_Ctl_) value is the primary index for nuclear import. As this value deviates spanning several orders of magnitude, we took the logarithm of the value. As the result, the log_2_[(L/H_+NTR_)/(L/H_Ctl_)] values distributed symmetrically (Figure 1—figure supplement 2), but the median and width of the distribution varied depending on the experiment. Thus, we normalized it using one of the simplest deviation values, Z-score (Figure 1—figure supplement 2). Defined as Z=(X-μ)/σ, where X is the value of the element, μ is the population mean, and σ is the standard deviation. The assumption is simple: the mean and the standard deviation should be same among the populations (among the three replicates). A simple interpretation is: if Z=0 for a protein, then the log_2_[(L/H_+NTR_)/(L/H_Ctl_)] value of the protein equals to the mean; if Z=+1 then the log_2_[(L/H_+NTR_)/(L/H_Ctl_)] value is deviated upward by 1 × σ; if Z=+2, the value is deviated 2 × σ; and if Z=-1, the value is deviated downward by 1 × σ.

*16) The cutoffs of 15% (and others) are arbitrary. To determine if the 'reported cargo' proteins are lumped at one end of the ranked list, the authors could use something like a Kolmogorov-Smirnov (KS) or Mann-Whitney (MW) test.*

We calculated Fisher’s exact test p-value, which is also commonly used. We calculated the p-values in 1% rank increments for the Trn-1 2nd- and 3rd-Z-rankings, and presented them with reported cargo rates (precision) and recall (sensitivity) (see the reply to point 17) as tables in [Supplementary-material SD1-data]. We also added the description about it in the Results and discussion (subsections “SILAC-Tp effectively identifies cargoes”, third paragraph and “Cargo selection with higher specificity”, first paragraph).

In addition to the distribution of reported cargoes, the 2nd-Z-15% and 3rd-Z-4% cutoffs are based on the distribution of PY-NLS motif-containing proteins in the Trn-1 2nd- and 3rd-Z-rankings (Figure 1).

*17) It should also be possible to estimate the sensitivity and a lower bound on the specificity at different index cutoffs using those receptors with high numbers of reported cargoes (i.e. TRN-1 and 2). The sensitivity at a cutoff will be (# reported cargoes selected) / (# reported cargoes selected + # reported cargoes not selected). The specificity is harder to quantify since the 'not carriers' is not known. However, one could consider all of the 'grey bar' proteins in Figure 1 as negatives, then one could estimate the upper bound on the number of false positives. An estimated sensitivity and specificity like this would help ground otherwise arbitrary cutoffs (e.g. top 100 proteins) as something interpretable. It would also help explain what it means for a cargo to pass the 2nd-Z-15% and 3rd-Z-4% criteria.*

Here we use a term, reported cargo rate (precision), because the definition of specificity is difficult in this work as the reviewer pointed out. We calculated the reported cargo rate (precision) and recall (sensitivity) for the Trn-1 2nd- and 3rd-Z-rankings in 1% rank increments and presented them as graphs and tables in Figure 1—figure supplement 3 and [Supplementary-material SD1-data].

Reported cargo rate(*i*)=*p(i)/[p(i*)+*n(i*)]

Recall (*i*)=*p(i)/P*

where *p(i*) denotes the number of reported cargoes and *n(i*) denotes the number of negative examples in the top i%; while *P* denotes the total number of reported cargoes. Since we do not know true non-cargo proteins as the reviewer noted, we calculated the reported cargo rates under two types of assumptions: (i) all proteins that have not been reported as cargoes should be regarded as non-cargo proteins, and (ii) only proteins annotated with non-nuclear localizations in UniProt (with experimental evidence) should be treated as non-cargo proteins. As the reviewer’s comment, in the case of (i) the reported cargo rate is the lower bound on the rate of true positives. We presented them in 1% rank increments of cutoffs for the top 1–20% rank orders. It will help to interpret the cutoffs that we employed here, and readers may select other cutoffs for their purpose. The Trn-1 2nd-Z-15% cutoff has the recall of 0.741, and the 3rd-Z-4% cutoff has the reported cargo rate of 0.85 under the assumption (ii). We added descriptions about these in the Results and discussion (subsections “SILAC-Tp effectively identifies cargoes”, third paragraph and “Cargo selection with higher specificity”, first paragraph) and the Material and methods (subsection “Reported cargo rate and recall”).

*18) The enrichments of reported cargoes in highly ranked proteins should have a p-Value, for example as calculated from the hypergeometric distribution.*

We inserted p-values by Fisher’s exact test at 11 places (Results and discussion).

*19) 'To calculate the +NTR/Ctl value, one protein has to be quantified in both the control and +NTR reactions, and we discarded L/H_+NTR_ values that lacked the counterpart L/H_Ctl_ values.' Wouldn't this remove the most interesting hits? That is to say, proteins that can only get into the nucleus through the added receptor?*

In our data set, such proteins are much fewer than expected and their quantitation values are less reliable. In our experimental system, the absolute quantities of a heavy (H) protein in the Ctl and +NTR samples should ideally be same, and those of the corresponding light (L) protein reflect the transport efficiency. Thus, a protein that has the L/H_+NTR_ value but lacks the L/H_Ctl_ value is a candidate cargo if the H_Ctl_ is quantified and the L/H_Ctl_ value is missing owing to the absence or zero quantity of L_Ctl_, as the reviewer pointed. However, we have discarded such proteins. We now recalculated the L/H ratios with the “replace missing quan values with minimum intensity” option by the Proteome Discoverer software (the results were not added in the manuscript as follows). By this option, missing quantitation values of either L or H peptide (not protein) are replaced with the minimum value quantified in the same LC-MS/MS analysis, and the presumably lowest L/H_Ctl_ values are given to the peptides lacking the L_Ctl_ but having the H_Ctl_ values. If a protein that was assigned with no Z-score in the previous calculation gets two or three Z-scores under this option, or if a protein with one Z-score gets one or two additional Z-scores, the protein will be included in the 2nd-Z-ranking. However, such proteins were unexpectedly rare, and the numbers of proteins added by this option to each of the 2nd-Z-rankings (totaling 1639 to 2060 proteins for each NTR, including both proteins selected and not selected as the 2nd-Z-15% cargoes) were ≤7. Among such proteins only seven listed below were ranked in the top 15%. (Many proteins changed their ranks slightly under this option.) Naturally, the L/H counts (the numbers of quantified peptides) of these proteins in one experiment were low (≤3). Because the quantitation values with this option are less reliable than those obtained without it, we just listed those proteins here and will not add them to the main lists. For the 3rd-Z-rankings, a few proteins that previously had two or less Z-scores got 3rd Z-scores under this option, but none of them were ranked in the top 4%. Some other proteins lack the L/H_Ctl_ ratios presumably because they lack the H_Ctl_ values, and for such proteins we have no reason to estimate that the missing L/H_Ctl_ ratios are lower than the L/H_+NTR_ ratios. Another point is that low protein abundance is not the sole reason for missing quantitation values.

NTRRankAccessionGene NameDescriptionTrn-SR1Q9H583HEATR1HEAT repeat-containing protein 1Imp-41P11177PDHBPyruvate dehydrogenase E1 component subunit β, mitochondriaImp-729P08574CYC1Cytochrome c1, heme protein, mitochondrialImp-830P08574CYC1Cytochrome c1, heme protein, mitochondrialImp-833Q92621NUP205Nuclear pore complex protein Nup205Imp-944Q14974KPNB1Importin subunit β-1Exp-47Q02218OGDH2-oxoglutarate dehydrogenase, mitochondrial

*20) In the second paragraph of the Introduction, the authors mentioned that NTR binding sites for cargoes have been established for Importins α, β, Trn-1/2 and Trn-SR. They missed the Imp-5 and its homologous Kap121p systems, where the Matsuura group very nicely elucidated the binding site for IK-NLSs found in Imp5/Kap121p cargoes.*

We added a mention on the Lys-rich NLS (IK-NLS) for Kap121p and cited Kobayashi and Matsuura 2013 and Kobayashi et al., 2015 in the Introduction (second paragraph).

*21) In the second paragraph of the Introduction, the last part of the sentence "…the PY-NLS motif has been defined, although the motif is not an absolute determinant of transport (Soniat and Chook, 2015)." is a bit confusing. Do the authors mean that presence of a PY-NLS does not necessarily determine transport, or do they mean that there are other motifs that may direct Trn-1/2 mediated transport?*

We modified the description as “in some cases the motif is difficult to recognize because of sequence diversity and structural disorder is another requisite”, and cited Soniat and Chook 2016 also in the Introduction (second paragraph).